# Cancer cells impair monocyte-mediated T cell stimulation to evade immunity

Anais Elewaut[1,2,14], Guillem Estivill[1,2,14], Felix Bayerl[3], Leticia Castillon[4], Maria Novatchkova[1], Elisabeth Pottendorfer[1,2], Lisa Hoffmann-Haas[1], Martin Schönlein[1], Trung Viet Nguyen[1], Martin Lauss[5], Francesco Andreatta[1], Milica Vulin[1], Izabela Krecioch[1], Jonas Bayerl[1,2], Anna-Marie Pedde[3], Naomi Fabre[1], Felix Holstein[1,2], Shona M. Cronin[1,2], Sarah Rieser[1], Denarda Dangaj Laniti[6,7,8], David Barras[6,7,8], George Coukos[6,7,8], Camelia Quek[9,10,11], Xinyu Bai[9,10,11], Miquel Muñoz i Ordoño[1], Thomas Wiesner[12], Johannes Zuber[1], Göran Jönsson[5], Jan P. Böttcher[3], Sakari Vanharanta[4,13] & Anna C. Obenauf[1✉]

The tumour microenvironment is programmed by cancer cells and substantially influences anti-tumour immune responses[1,2]. Within the tumour microenvironment, CD8[+] T cells undergo full effector differentiation and acquire cytotoxic anti-tumour functions in specialized niches[3–7]. Although interactions with type 1 conventional dendritic cells have been implicated in this process[3–5,8–10], the underlying cellular players and molecular mechanisms remain incompletely understood. Here we show that inflammatory monocytes can adopt a pivotal role in intratumoral T cell stimulation. These cells express *Cxcl9*, *Cxcl10* and *Il15*, but in contrast to type 1 conventional dendritic cells, which cross-present antigens, inflammatory monocytes obtain and present peptide–major histocompatibility complex class I complexes from tumour cells through 'cross-dressing'. Hyperactivation of MAPK signalling in cancer cells hampers this process by coordinately blunting the production of type I interferon (IFN-I) cytokines and inducing the secretion of prostaglandin $E_2$ ($PGE_2$), which impairs the inflammatory monocyte state and intratumoral T cell stimulation. Enhancing IFN-I cytokine production and blocking $PGE_2$ secretion restores this process and re-sensitizes tumours to T cell-mediated immunity. Together, our work uncovers a central role of inflammatory monocytes in intratumoral T cell stimulation, elucidates how oncogenic signalling disrupts T cell responses through counter-regulation of $PGE_2$ and IFN-I, and proposes rational combination therapies to enhance immunotherapies.

Although T cell responses are often suppressed in the tumour microenvironment (TME) by inhibitory signals, emerging data suggest that the TME also has a pivotal role in supporting T cell function[1,11]. The activation and differentiation of CD8[+] T cells was thought to occur primarily in the lymph node. However, recent findings indicate that after reaching the tumour, antigen-committed memory or effector T cells require further restimulation to expand, differentiate and effectively control tumour growth[3,4,8,11,12]. This process is thought to take place in discrete niches within the TME, where T cells spatially organize with myeloid cells[3–7], in particular activated CCR7[−] type 1 conventional dendritic cells (cDC1s) and conventional dendritic cells (cDCs) in a stimulatory CCR7[+] state[3,8–10,13,14]. Such multicellular hubs have increasingly been linked to positive outcomes in immunotherapy, thereby underscoring their

therapeutic relevance[4,5]. The growing recognition of T cell restimulation within the TME has prompted the addition of 'the TME subcycle' as a new step in the cancer immunity cycle[11]. However, the processes that facilitate restimulation of primed CD8[+] T cells within the TME remain incompletely understood.

To identify mechanisms of T cell restimulation in the tumour, we capitalized on matched pairs of tumour models that were derived by exposing targeted therapy-naive (N[TT]) *Braf*[V600E]-driven melanoma models to MAPK pathway inhibitors (BRAF inhibitor (BRAFi), and BRAFi and MEK inhibitor (BRAFi/MEKi)) until they acquired resistance (R[TT]). We previously demonstrated that although N[TT] tumours are susceptible to eradication through immune checkpoint blockade (ICB) and adoptive T cell transfer (ACT), R[TT] tumours harbour an immune-evasive

[1]Research Institute of Molecular Pathology (IMP), Vienna BioCenter (VBC), Vienna, Austria. [2]Vienna BioCenter, Doctoral School of the University of Vienna and Medical University of Vienna, Vienna, Austria. [3]Institute of Molecular Immunology, School of Medicine and Health, Technical University of Munich (TUM), Munich, Germany. [4]Translational Cancer Medicine Program, Faculty of Medicine, University of Helsinki, Helsinki, Finland. [5]Lund University Cancer Center, Division of Oncology, Lund University, Lund, Sweden. [6]Ludwig Institute for Cancer Research, Lausanne Branch, University of Lausanne (UNIL), Lausanne, Switzerland. [7]Department of Oncology, University Hospital of Lausanne (CHUV) and University of Lausanne (UNIL), Lausanne, Switzerland. [8]Agora Research Center, Lausanne, Switzerland. [9]Melanoma Institute Australia, The University of Sydney, Sydney, New South Wales, Australia. [10]Charles Perkins Centre, The University of Sydney, Sydney, New South Wales, Australia. [11]Faculty of Medicine and Health, The University of Sydney, Sydney, New South Wales, Australia. [12]Department of Dermatology, Medical University of Vienna, Vienna, Austria. [13]Department of Biochemistry and Developmental Biology, Faculty of Medicine, University of Helsinki, Helsinki, Finland. [14]These authors contributed equally: Anais Elewaut, Guillem Estivill. ✉e-mail: anna.obenauf@imp.ac.at

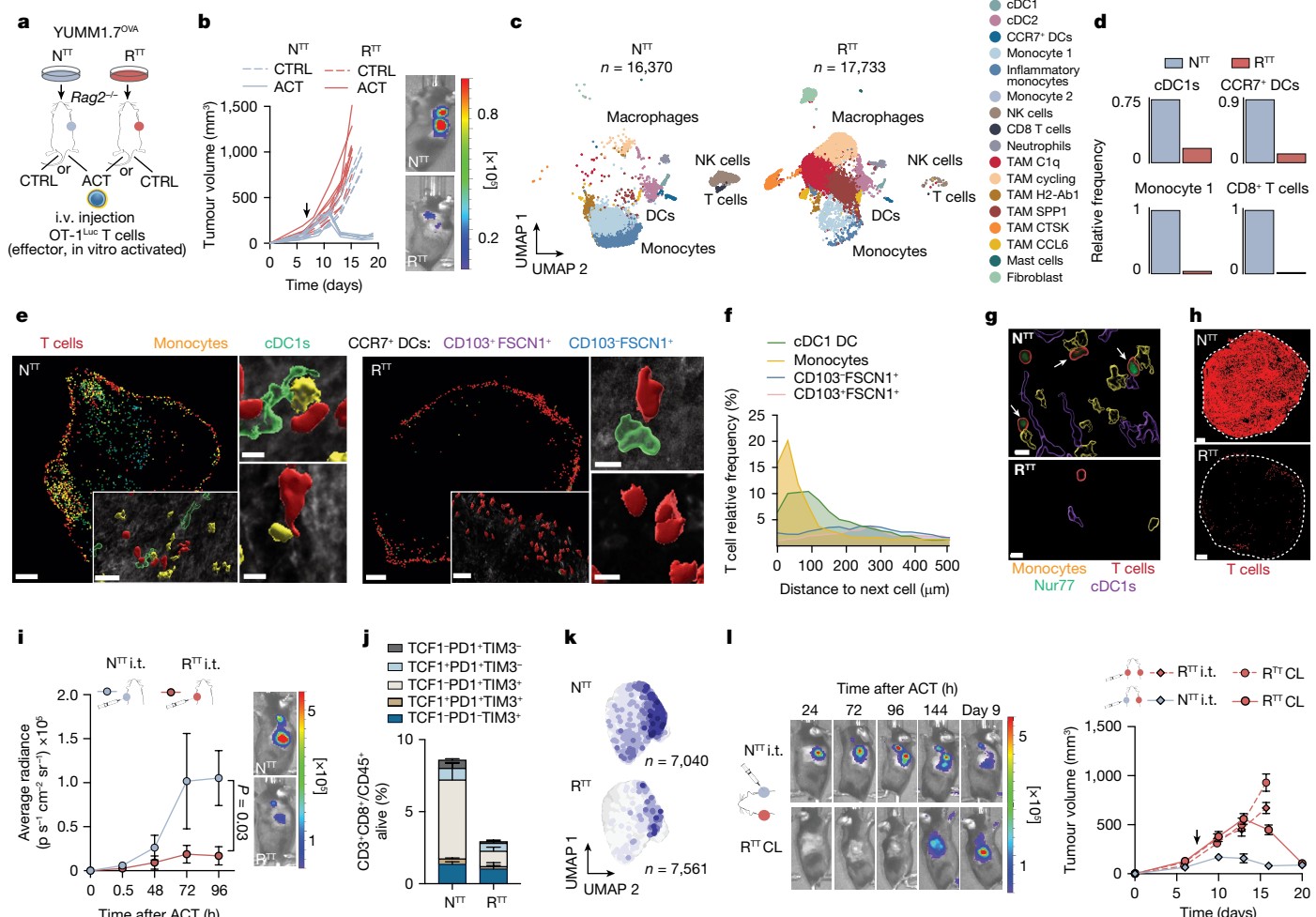

**Fig. 1 | Tumour-infiltrating T cells are restimulated in permissive TMEs.**
**a**, Schematic of subcutaneous injection of YUMM1.7[OVA] N[TT] cells or R[TT] cells in *Rag2*[−/−] mice and OT-1[Luc] ACT through intravenous (i.v.) injection. **b**, Left, ACT responses of N[TT] and R[TT] tumours (*n* = 5 mice per group). Right, representative bioluminescence imaging (BLI) pictures of T cells 96 h after ACT; key indicates radiance (photons per second that leave a square centimeter of tissue and radiate into a solid angle of one steradian). **c**, Uniform manifold approximation and projection (UMAP) maps of scRNA-seq of CD45[+] cells in N[TT] and R[TT] tumours 72 h after ACT (*n* = 4 tumours pooled per group). **d**, Relative cell frequencies from scRNA-seq. **e**, Representative multiparameter IF images of YUMM1.7[OVA] N[TT] and R[TT] tumours 48 h after ACT (*n* = 3 tumours per group). Scale bar, 400 μm (zoom-ins, 50 μm and 10 μm). **f**, Relative T cell frequency and distance to the next immune cell in N[TT] tumours (*n* = 3 tumours per group). **g**, Representative IF image of Nur77–GFP OT-1 cells in YUMM1.7[OVA] N[TT] and R[TT] tumours 48 h after ACT (*n* = 3 tumours per group). Scale bar, 10 μm. Arrows indicate Nur77[+] OT-1 cells. **h**, Representative IF image of YUMM1.7[OVA] N[TT] and R[TT] tumours 72 h after ACT. Dashed lines depict the tumour border (*n* = 2 tumours per group). Scale bar, 500 μm. **i**, Left, quantification of T cells by BLI over time after intratumoral (i.t.) ACT (mean ± s.e.m., *n* = 4 mice per group). Two-way analysis of variance (ANOVA) with Sidak's multiple comparisons test. Right, representative BLI image 96 h after ACT (key as in **b**). **j**, T cell states by flow cytometry 120 h after i.t. ACT (mean ± s.e.m., *n* = 5 N[TT], *n* = 4 R[TT] tumours). **k**, Effector memory T cell signature[29] on scRNA-seq of T cells (*n* = 14 N[TT], *n* = 18 R[TT] pooled tumours). **l**, Left, representative BLI images of T cells in N[TT] and contralateral (CL) R[TT] tumours after i.t. ACT (key as in **b**). Right, growth curves (mean ± s.e.m., *n* = 6 mice per group). Arrows in **b** and **l** indicate day of ACT.

TME that renders them cross-resistant to immunotherapies[15]. Prospective clinical studies[16–18] have confirmed that resistance to targeted therapy jeopardizes subsequent responses to immunotherapy. Here using these models, we show that inflammatory monocytes facilitate the intratumoral expansion of primed T cells even in the absence of cDC1s. We provide mechanistic insights into their mode of action, identify cancer-cell-derived cues that disrupt this process and propose mechanism-based therapies that can reinstate anti-tumour immunity.

## Myeloid polarization underlies immune escape

We established N[TT] tumours and R[TT] tumours of the YUMM1.7 mouse melanoma model (*Braf*[V600E], *Pten*[−/−] *Cdkn2a*[−/−]) that express the model antigen ovalbumin (OVA) in *Rag2*[−/−] mice and performed ACT by intravenously injecting activated tumour antigen-specific CD8[+] T cells (OT-1[Luc]) (Fig. 1a). As previously shown[15], in N[TT] tumours, T cells infiltrated and

controlled tumour growth, whereas R[TT] tumours were resistant (Fig. 1b and Extended Data Fig. 1a). Notably, OT-1[Luc] T cells effectively killed N[TT] and R[TT] tumour cells in vitro (Extended Data Fig. 1b), which implicated a role for the TME in mediating resistance to T cell killing in vivo. To define the composition and transcriptional states of the immune cells within the TME of N[TT] tumours and R[TT] tumours, we performed single-cell RNA sequencing (scRNA-seq) of CD45[+] immune cells before ACT and 72 h after ACT. The immune landscape was markedly different between N[TT] and R[TT] YUMM1.7[OVA] tumours. Specifically, there was a reduction in total CD45[+] abundance in R[TT] tumours and prominent differences within the myeloid cell compartment (Fig. 1c,d and Extended Data Fig. 1c–e). Monocytes, the most abundant immune population in the TME of N[TT] tumours, were strongly reduced in R[TT] tumours (Fig. 1c,d and Extended Data Fig. 1d). This population expressed typical monocyte markers (*Ly6c2*, *C5ar1* and *Fcgr1*), low levels of macrophage markers (*Adgre* and *Apoe*) and lacked classical cDC markers (*Cd24a*, *Flt3*,

*Dpp4* and *Zbtb46*) (Extended Data Fig. 1e and Supplementary Table 1). Moreover, in $R^{TT}$ tumours, cDCs (*Zbtb46*[+], *Flt3*[+] and *Cd24*[+]), including cDC1s and activated CCR7[+] DCs (often referred to as mregDCs)[19–22], were severely reduced and in a dysfunctional state (Fig. 1c,d and Extended Data Fig. 1d–f). Furthermore, immunosuppressive tumour-associated macrophages (TAMs)[21,23,24] (for example, *Spp1*[+] and *Ctsk*[+]) and cycling TAMs were increased in $R^{TT}$ tumours compared with $N^{TT}$ tumours. We observed similar repolarization of the myeloid compartment in the YUMM3.3 (*Braf*[V600E], *Cdkn2a*[−/−]) model, in which $R^{TT}$ tumours failed to respond to ICB (that is, anti-PD-1 and anti-CTLA-4 agents)[15] (Extended Data Fig. 1g–j). These results highlight the conserved regulation and functional importance of the myeloid TME across models and immunotherapies.

## T cells are restimulated in permissive TMEs

Recent studies have shown that activated CD8[+] T cells require additional stimuli from intratumoral myeloid cells to acquire full effector functions and to sustain a T cell response[3,6,10,12]. To examine this idea further, we used multiparameter immunofluorescence (IF) microscopy for discriminatory myeloid cell markers to determine the main interaction partner (or partners) of tumour-infiltrating T cells (Fig. 1e and Extended Data Fig. 2a–c). In $N^{TT}$ tumours, tumour-specific CD8[+] T cells were in close proximity to cDC1s and monocytes and were often organized in multicellular clusters at the tumour margin (Fig. 1f and Extended Data Fig. 2a–c). Notably, tumour-infiltrating T cells interacted with monocytes more commonly than with cDC1s, which is possibly due to the high abundance of monocytes in the TME (Fig. 1f and Extended Data Fig. 2a,b). Within these hubs, T cells stained positive for Nur77, a marker indicative of recent T cell receptor (TCR) signalling, which indicated not only proximity but also direct antigen-specific stimulation by the interacting myeloid cells (Fig. 1g and Extended Data Fig. 2d,e). Within 72 h after ACT, T cells in $N^{TT}$ tumours permeated the entire tumour parenchyma, which was not abolished by blocking T cell egress from the lymph node with FTY720. This result suggested the occurrence of local expansion (Fig. 1h and Extended Data Fig. 2f). By contrast, $R^{TT}$ tumours did not contain such hubs, and T cells remained confined to the periphery. Moreover, they rarely interacted with the few cDCs and monocytes present, and Nur77[+] T cells were strongly reduced (Fig. 1g and Extended Data Fig. 2d,e).

To investigate T cell functionality within TMEs, we injected the same amount of activated OT-1[Luc] T cells intratumorally into $N^{TT}$ tumours and $R^{TT}$ tumours (Fig. 1i). In contrast to $R^{TT}$ tumours, CD8[+] T cells rapidly underwent a proliferative burst and expanded in $N^{TT}$ tumours (Fig. 1i and Extended Data Fig. 2g). Five days after intratumoral injection, we observed a significant reduction in differentiated CD8[+] T cells (TCF1[−] PD1[+]TIM3[+]) in $R^{TT}$ tumours compared with $N^{TT}$ tumours. By contrast, in both conditions, a small fraction (about 5%) of all T cells remained in a stem-cell-like state (TCF1[+]PD1[+]TIM3[−]), which is required for expansion of the T cell effector pool[12,25–28] (Fig. 1j and Extended Data Fig. 2h,i). In $N^{TT}$ tumours, but not in $R^{TT}$ tumours, T cells acquired features of effector memory T cells[29], and the majority displayed upregulated expression of the T cell effector marker CXCR6 (ref. 30) (Fig. 1k, Extended Data Fig. 2j,k and Supplementary Table 2). Altogether, these findings suggest that multicellular hubs that contain cDCs and, unexpectedly, a substantial fraction of monocytes, are associated with T cell expansion. Moreover, T cell proliferation and effector differentiation is facilitated only within the TME of $N^{TT}$ tumours but not of $R^{TT}$ tumours.

## Local licensing of systemic immunity

To examine whether an immune-permissive TME can act as a reservoir for T cell restimulation, we established $N^{TT}$ tumours and $R^{TT}$ tumours in opposite flanks within the same mouse and performed intravenous ACT. T cells expanded in $N^{TT}$ tumours and, at later time points, infiltrated

contralateral $R^{TT}$ tumours and transiently controlled their growth (Extended Data Fig. 2l–o). Of note, we did not observe differences in the myeloid composition of $R^{TT}$ tumours in the presence of a contralateral $N^{TT}$ tumour (Extended Data Fig. 2p). Treatment with FTY720 only marginally reduced T cell infiltration into contralateral $R^{TT}$ tumours, but still resulted in tumour control. This result suggests that T cells may traffic partially through the lymph node and directly through the circulation (Extended Data Fig. 2q,r). To further confirm that restimulated T cells traffic between tumours, we directly injected activated OT-1[Luc] T cells into one tumour and evaluated T cell infiltration in contralateral tumours. T cells introduced into a $N^{TT}$ tumour demonstrated the capacity to expand locally and infiltrate contralateral tumours, irrespective of whether these were $N^{TT}$ tumours or $R^{TT}$ tumours. Contralateral $N^{TT}$ tumours fully regressed, and even contralateral $R^{TT}$ tumours were temporarily controlled (Fig. 1l and Extended Data Fig. 2s–u). Notably, T cells directly injected into $R^{TT}$ tumours initially failed to expand but trafficked to contralateral $N^{TT}$ tumours, where they expanded and controlled tumour growth and, eventually, re-infiltrated $R^{TT}$ tumours (Extended Data Fig. 2t,u). Collectively, our data indicate that after initial priming of T cells, restimulation and subsequent effector functions are strongly dictated by the characteristics of the TME. We conclude that T cells are capable of trafficking between tumours and that T cell restimulation in an immune-permissive TME can facilitate the control of distant, resistant tumours.

## Immunostimulatory role of monocytes

Activated cDC1s and CCR7[+] cDCs have long been implicated in antitumour immunity[13,31–35] and, recently, in intratumoral CD8[+] T cell restimulation[3,4,8,9]. Consistent with that, cDC1 vaccination in $R^{TT}$ tumours restored T cell infiltration and led to transient tumour control (Extended Data Fig. 3a–c). Notably, when $N^{TT}$ cells were injected into *Rag2*[−/−] *Batf3*[−/−] mice, which lack functional cDC1s, and ACT was performed, T cells still infiltrated, expanded and controlled $N^{TT}$ tumours (Fig. 2a). In the absence of functional cDC1s, multiparameter IF showed retained immune hubs, with monocytes clustering together with T cells in $N^{TT}$ tumours but not in $R^{TT}$ tumours (Extended Data Fig. 3d–f). Furthermore, in $N^{TT}$ tumours grown in *Zbtb46*–DTR bone marrow chimeras, in which all cDC subsets are depleted, T cells were effectively restimulated and expanded (Extended Data Fig. 3g–i).

Next, we analysed monocytes for expression of genes linked to T cell stimulation. Monocytes expressed increased levels of the antigen presentation machinery (*Psmb8*, *Psmb9*, major histocompatibility complex class I (MHCI) and MHCII) and *Il15*, which mediates T cell survival and effector differentiation[3]. *Cxcl9* and *Cxcl10*, which are essential for T cell recruitment and linked to positive immunotherapy responses[33], were also expressed at high levels (Fig. 2b). These monocytes did not score positively for an established monocyte-derived DC signature[21], and a large subset of these monocytes also expressed high levels of interferon-stimulated genes (ISGs), which identified them as inflammatory monocytes (Figs. 1c and 2b and Extended Data Fig. 3j). These inflammatory monocytes expressed the interferon-induced surface marker Ly6A (Extended Data Fig. 1e), a result consistent with analyses of human and mouse tumours and viral infection models, which showed that myeloid cells transition to an inflammatory state after IFN-I signalling[23,36–40]. As expected, previously defined ISG signatures (Supplementary Table 3) scored the highest in the monocyte and inflammatory monocyte cluster in $N^{TT}$ tumours (Fig. 2c and Extended Data Fig. 3k). Single-cell regulatory network inference and clustering (SCENIC) analyses predicted transcriptional activity of *Irf9*, *Irf7*, *Irf2* and *Stat2*, major effectors of interferon signalling, specifically in inflammatory monocytes (Fig. 2d). Altogether, these data suggest that even in the absence of cDCs, monocytes are capable of mediating the expansion of tumour-specific CD8[+] T cells and promoting anti-tumour immunity.

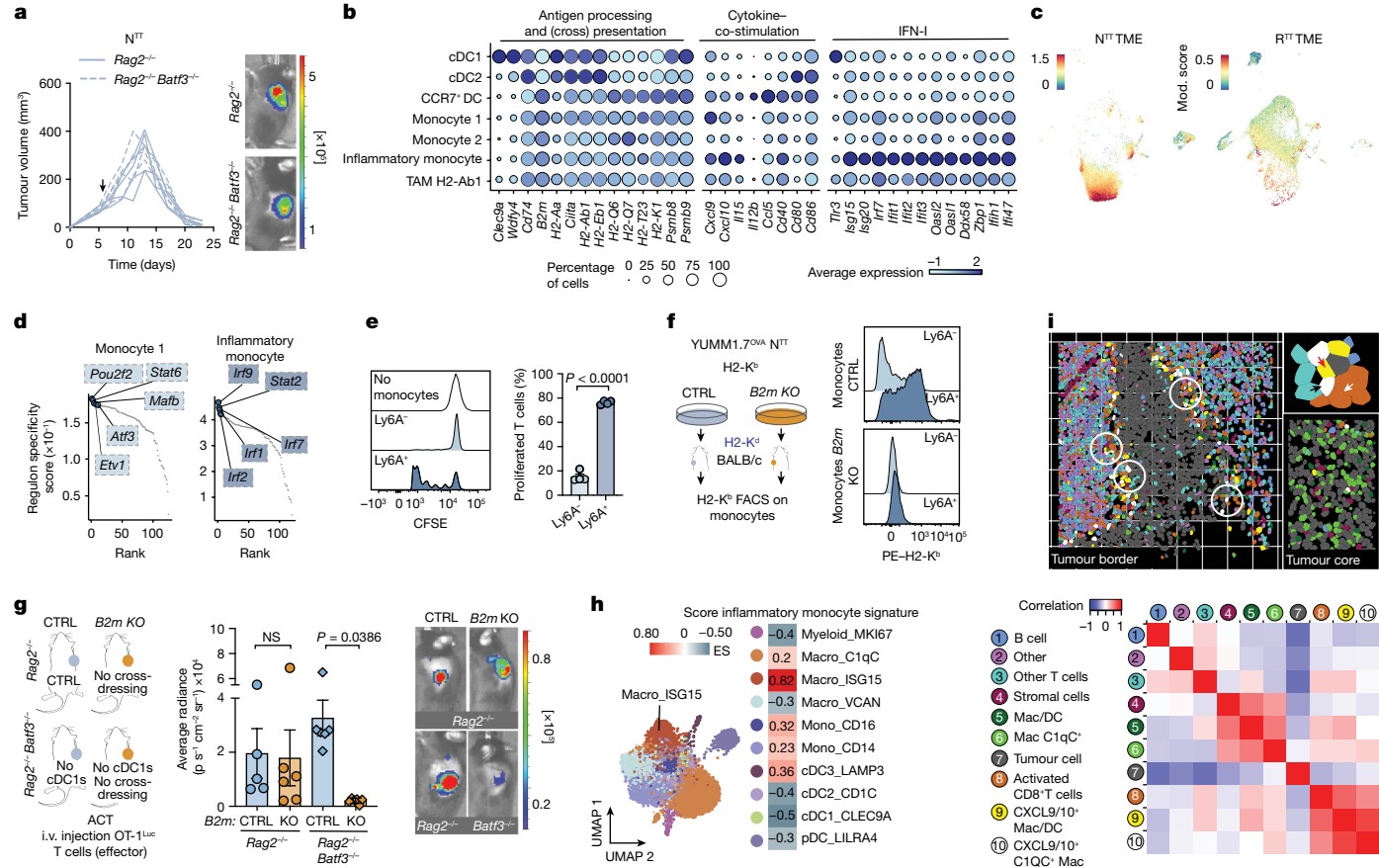

**Fig. 2 | Inflammatory monocytes restimulate T cells within the TME. a**, Left, ACT response of YUMM1.7[OVA] N[TT] tumours in $Rag2^{-/-}$ mice ($n$ = 4 mice) and $Batf3^{-/-}Rag2^{-/-}$ ($n$ = 5 mice). Right, representative BLI image of T cells 96 h after ACT (key as in Fig. 1b). **b**, Gene expression in individual clusters from scRNA-seq from Fig. 1c. **c**, Scoring of an ISG signature[37] on our scRNA-seq data from Fig. 1c. Mod., module. **d**, Regulon specificity score of monocytes calculated using SCENIC in N[TT] tumours. **e**, Left, representative histograms depicting CFSE intensity. Right, quantification of T cell proliferation after 72 h of co-culture of naive CFSE-labelled T cells and inflammatory (Ly6A⁺) or non-inflammatory monocytes (Ly6A⁻) isolated from N[TT] tumours in $Rag2^{-/-}$ mice ($n$ = 4 tumours). Two-tailed unpaired Student's $t$-test. **f**, Left, schematic of injection of YUMM1.7[OVA] N[TT] CTRL or $B2m$ KO cells in BALB/c mice. Right, representative histograms depicting H2-K[b] levels. **g**, Left, schematic of injection of N[TT] CTRL or N[TT] $B2m$ KO tumours in $Rag2^{-/-}$ mice and $Batf3^{-/-}Rag2^{-/-}$ mice. Middle,

quantification of T cell infiltration by BLI. Right, representative BLI image of T cell infiltration 96 h after ACT ($n$ = 5 N[TT] in $Rag2^{-/-}$ and $n$ = 6 mice for other groups) (key as in Fig. 1b). One-way ANOVA with Tukey's multiple comparisons test. **h**, Left, UMAP of human inflammatory macrophages from a melanoma scRNA-seq myeloid dataset[23]. Right, enrichment score (ES) values of the inflammatory monocyte gene signature for each cell cluster. **i**, Top, representative field of view (FOV) of human metastatic melanoma ($n$ = 2 of 72 FOVs) analysed by CosMx spatial transcriptomics profiling. Bottom, Pearson's correlation values between cell types across FOVs ($n$ = 72) were determined and displayed as a heatmap ($n$ = 34 melanoma samples). The white arrow depicts activated CD8⁺ T cells, the black arrow depicts $CXCL9^{+}CXCL10^{+}C1QC^{+}$ macrophages, and the red arrow $CXCL9^{+}CXCL10^{+}$ macrophages and DCs. Mac, macrophage. Bar graphs depict the mean ± s.e.m.

## MHCI-dressed monocytes stimulate T cells

Given that T cells close to monocytes within immune hubs expressed Nur77, indicative of TCR stimulation (Fig. 1g and Extended Data Fig. 3l), we assessed whether monocytes are able to present tumour antigens. H2-K[b]-SIINFEKL staining revealed that inflammatory monocytes, but not their non-inflammatory counterparts, displayed tumour-derived antigens on MHCI (Extended Data Fig. 3m). Moreover, they induced naive T cell activation and proliferation ex vivo, as measured by CFSE dilution (Fig. 2e). However, inflammatory monocytes do not express genes involved in cross-presentation (*Clec9a* and *Wdfy4*)[14] (Fig. 2b). This prompted us to investigate whether inflammatory monocytes can acquire and display antigens through the direct transfer of intact peptide–MHCI (pMHCI) complexes from adjacent cells, a process called MHCI cross-dressing[37,41,42].

To investigate whether monocytes are able to cross-dress, we established N[TT] tumours, which are derived from C57BL/6 mice and therefore express H2-K[b], into MHCI haplotype-mismatched (H2-K[d])

BALB/c mice and examined cancer-cell-derived H2-K[b] expression on myeloid subsets. Only inflammatory monocytes (Ly6A⁺) harboured cancer-cell-derived H2-K[b] and were capable of activating naive T cells ex vivo (Fig. 2f and Extended Data Fig. 3n,o). Notably, in N[TT] tumours harbouring a knockout (KO) of β₂-microglobulin (N[TT] $B2m$ KO), no H2-K[b] signal was detected on BALB/c inflammatory monocytes. This result demonstrated that pMHCI complexes on inflammatory monocytes are sourced from cancer cells (Fig. 2f and Extended Data Fig. 3n). To test the relative contribution of cross-presenting cDC1s and cross-dressed inflammatory monocytes to T cell restimulation, we injected N[TT] and N[TT] $B2m$ KO tumours, which abolishes the ability of inflammatory monocytes to present tumour antigens through MHCI cross-dressing, into $Rag2^{-/-}$ mice and $Rag2^{-/-}Batf3^{-/-}$ mice and performed ACT (Fig. 2g). When pMHCI cross-dressing on monocytes was intact, T cells expanded in N[TT] tumours, even in the absence of cDC1s. Conversely, when pMHCI cross-dressing on inflammatory monocytes was abolished ($B2m$ KO tumours) but cDC1s were present, T cells were also able to expand, which reflected the established capacity of cDC1s to restimulate CD8⁺

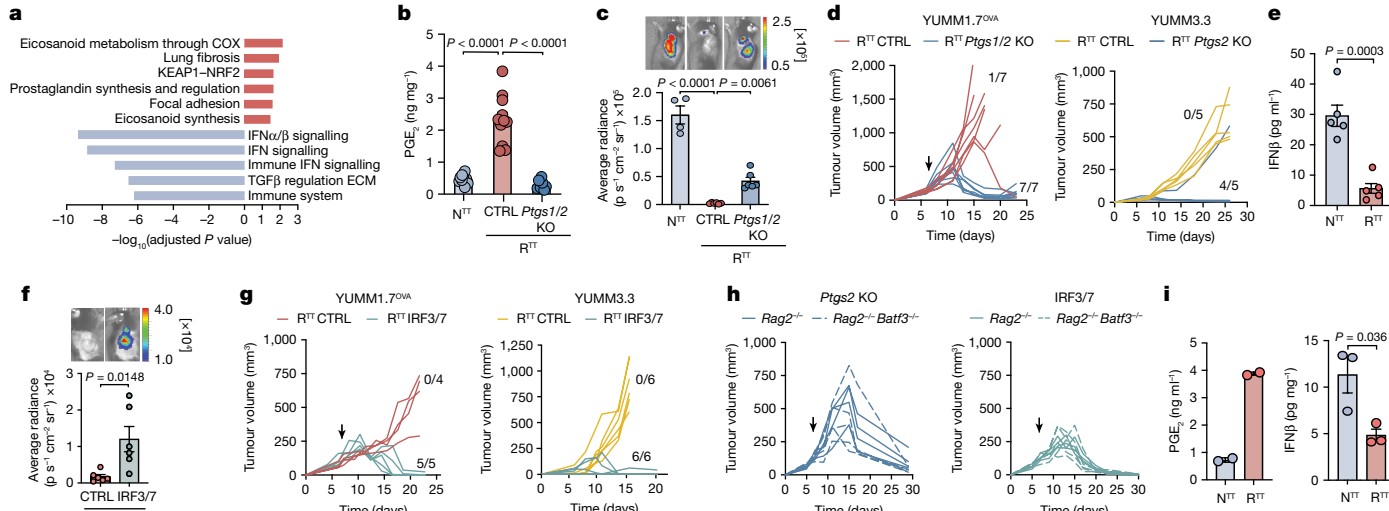

**Fig. 3 | Cancer cells produce PGE₂ and downregulate IFN-I responses to confer immunotherapy resistance. a**, Pathway enrichment analysis of differential gene expression in cancer cells isolated from YUMM1.7^OVA N^TT tumours and R^TT tumours (*n* = 3 tumours per group; Supplementary Table 3). Adjusted *P* values were computed using the Benjamini–Hochberg correction. **b**, PGE₂ ELISA of YUMM1.7^OVA tumours in *Rag2^−/−* mice (*n* = 10 N^TT, *n* = 11 R^TT CTRL, *n* = 7 R^TT *Ptgs1/2* KO over 2 independent experiments). **c**, Top, representative BLI image of T cells 96 h after ACT (key as in Fig. 1b). Bottom, BLI quantification (*n* = 4 mice N^TT, *n* = 6 mice R^TT CTRL and R^TT *Ptgs1/2* KO). **d**, Left, response of YUMM1.7^OVA tumours in *Rag2^−/−* mice to ACT (*n* = 7 mice per group). Right, YUMM3.3 in C57BL/6 mice (*n* = 5 mice per group). **e**, IFNβ ELISA of supernatants from YUMM1.7^OVA N^TT and R^TT cells (*n* = 5 replicates per group over 2 independent experiments). **f**, Top,

representative BLI images of T cells in YUMM1.7^OVA R^TT CTRL and R^TT IRF3/7 tumours in *Rag2^−/−* mice 96 h after ACT (key as in Fig. 1b). Bottom, BLI quantification (*n* = 6 mice per group). **g**, Left, response of YUMM1.7^OVA tumours in *Rag2^−/−* mice to ACT (*n* = 4 mice R^TT CTRL, *n* = 5 mice R^TT IRF3/7). Right, YUMM3.3 in C57BL/6 mice (*n* = 6 mice per group). **h**, Response to ACT of YUMM1.7^OVA R^TT *Ptgs2* KO (left) and R^TT IRF3/7 tumours (right) in *Rag2^−/−* (*n* = 4 mice per group) and *Batf3^−/− Rag2^−/−* mice (*n* = 5 mice per group). **i**, PGE₂ and IFNβ ELISAs of N^TT and R^TT A375 human melanoma (*n* = 2 replicates per group for PGE₂, *n* = 3 tumours per group for IFNβ). Arrows in **d**, **g** and **h** indicate day of ACT. Bar graphs depict the mean ± s.e.m. Statistical analysis was performed with a two-tailed unpaired Student's *t*-test (**e**,**f**,**i**) or one-way ANOVA with Tukey's multiple comparisons test (**b**,**c**).

---

T cells through classical cross-presentation. Only when both cDC1s and cross-dressed inflammatory monocytes were absent (*B2m* KO tumours engrafted into *Rag2^−/− Batf3^−/−* mice) did T cells fail to become restimulated (Fig. 2g). Collectively, our data highlight that pMHCI cross-dressing by inflammatory monocytes, together with stimulatory cytokine expression, underlies their ability to promote restimulation of primed CD8^+ T cells in the TME.

## ISG^+ macrophages in human melanoma

We scored our mouse inflammatory monocyte signature in human myeloid scRNA-seq datasets of melanoma and non-small cell lung cancer (NSCLC)[23,43]. This analysis showed that ISG^+ (CXCL9^+CXCL10^+) macrophages and CD16^+ monocytes are the analogous inflammatory populations across human cancers (Fig. 2h, Extended Data Fig. 4a–c and Supplementary Table 2). Notably, analysing the spatial distribution of T cells with immune cells in human melanoma samples revealed that activated CD8^+ T cells preferentially co-localized with CXCL9^+CXCL10^+ inflammatory macrophages and cDCs in immune hubs. By contrast, regions that lacked CXCL9^+CXCL10^+ macrophages were devoid of T cells (Fig. 2i).

## Cancer cells produce PGE₂ and dampen IFN-I responses

To identify factors derived from N^TT and R^TT cells that determine the intratumoral immune landscape pivotal for T cell restimulation, we isolated cancer cells from tumours and performed RNA-seq. Pathway enrichment analysis of differentially regulated genes revealed upregulation of the prostaglandin synthesis pathway and downregulation of IFN-I signalling in R^TT cells as the top differential pathways compared with N^TT cells (Fig. 3a and Supplementary Table 3). Metabolomics analysis identified PGE₂ as the most enriched eicosanoid in R^TT tumours (Extended Data Fig. 5a). PGE₂ can limit cDC1-mediated support of CD8^+

T cells[9,34,44], but its impact on inflammatory monocytes and on their ability to promote T cell restimulation is unclear.

To understand the role of cancer-cell-derived PGE₂ in immune evasion, we ablated PGE₂ production through cyclooxygenase-1 (COX1; encoded by *Ptgs1*) and COX2 (encoded by *Ptgs2*) KO in R^TT cells and engrafted them into *Rag2^−/−* mice. R^TT tumours deficient in *Ptgs1* and *Ptgs2* (*Ptgs1/2*) displayed reduced PGE₂ levels that were comparable to those of N^TT tumours (Fig. 3b) and increased T cell infiltration (Fig. 3c). Notably, *Ptgs1/2* deletion fully re-sensitized R^TT tumours to ACT and grew unperturbed without ACT (Fig. 3d and Extended Data Fig. 5b). Genetic ablation of only *Ptgs2* also led to re-sensitization of R^TT tumours to ACT (Extended Data Fig. 5c–e). Similarly, *Ptgs2* deletion in the YUMM3.3 R^TT model resulted in tumour rejection in immuno-competent C57BL/6 mice in a T cell-dependent manner, even without ICB treatment (Fig. 3d and Extended Data Fig. 5f,g). Similar effects of *Ptgs2* inactivation have been reported in other mouse models[34,44–46], which highlights the role of PGE₂ in the TME as a strong modulator of T cell responses.

Given the effects of PGE₂, we wondered whether the transcriptional downregulation of the IFN-I program in R^TT cells (Fig. 3a) was merely a reflection of low interferon levels in a PGE₂-induced immunosuppressive TME or an independent driver of immune evasion. IFNβ was significantly reduced in R^TT tumour lysates compared with N^TT tumour lysates (Extended Data Fig. 5h), with R^TT cells producing less IFNβ in vitro. This result implied that there is cancer-cell-intrinsic regulation of IFN-I cytokine production without microenvironmental cues (Fig. 3e). Upstream regulator analysis (Ingenuity) predicted that these transcriptional changes stemmed from decreased activity of the transcription factors IRF3 and IRF7 (IRF3/7), which are important regulators of interferon production (Extended Data Fig. 5i).

To test whether re-establishment of a functional IFN-I pathway in R^TT cancer cells is sufficient to restore responses to ACT, we overexpressed IRF3/7 in R^TT cells and established tumours in *Rag2^−/−* mice. This model

showed restored IFN-I cytokine levels in the TME (Extended Data Fig. 5h) and increased T cell infiltration, which in turn led to full tumour control after ACT (Fig. 3f,g and Extended Data Fig. 5j). YUMM3.3 R[TT] tumours also harboured lower IFNβ levels than their N[TT] counterparts (Extended Data Fig. 5k), and after re-establishment of an interferon response and injection in C57BL/6 mice, the tumours were controlled (Fig. 3g). Moreover, similar to N[TT] tumours, YUMM1.7[OVA] R[TT] tumours with *Ptgs2* KO or IRF3/7 overexpression were controlled in *Rag2*[−/−]*Batf3*[−/−] mice, which lack cDC1s (Fig. 3h). Collectively, these findings suggest that R[TT] cancer cells establish an immune-evasive TME by increasing PGE$_2$ and simultaneously reducing IFN-I cytokine production. Reverting either of these events re-sensitizes tumours to killing by activated T cells, even in the absence of cDC1s (Fig. 3h).

## MAPK signalling regulates PGE$_2$ and IFN-I responses

We recently showed that cross-resistance between targeted therapy (BRAFi/MEKi) and immunotherapy is driven by reactivated onco-genic RAF–MEK–ERK signalling[15]. To explore whether the hyperacti-vated MAPK pathway in R[TT] tumours is the common regulator of both immune-evasive programs, we inhibited the MAPK pathway in cancer cells in vitro and in vivo. This strategy induced the expression of ISGs and reduced COX2 levels (Extended Data Fig. 5l–n), a result consistent with previous reports[44,47]. We examined whether PGE$_2$ could be the cause of the dampened IFN-I program in R[TT] cancer cells. However, *Ptgs2* deletion did not restore ISG expression and it did not increase IFN-I cytokine levels in the TME (Extended Data Fig. 5o,p). Similarly, IRF3/7 overexpression in R[TT] tumours did not attenuate PGE$_2$ produc-tion (Extended Data Fig. 5q).

To address the human relevance of these findings, we assessed PGE$_2$ and IFN-I cytokine production in matched pairs of the human RAFi-sensitive (N[TT]) and RAFi-resistant (R[TT]) melanoma cell lines A375, M249 and LOX[48]. We consistently found an increased production of PGE$_2$ in targeted-therapy-resistant cells, together with a decrease in IFN-I responses. This result was also confirmed in the KRAS-driven NSCLC cell line NCI-H358 after it acquired resistance to a targeted KRAS inhibitor (Fig. 3i and Extended Data Fig. 5r). COX2 levels were regu-lated by the MAPK pathway in the A375 cell line after BRAF inhibition or after NRAS overexpression (Extended Data Fig. 5s). Collectively, these studies indicate a common regulatory module driven by oncogenic MAPK signalling that upregulates PGE$_2$ levels and downregulates IFN-I responses in cancer cells to drive immune evasion.

## PGE$_2$ and IFN-I responses instruct myeloid polarization

We next wanted to understand how genetic ablation of PGE$_2$ synthe-sis or restoration of IFN-I responses in R[TT] cancer cells reinstates an immune-permissive TME rich in inflammatory monocytes. We per-formed scRNA-seq and flow cytometry analyses of CD45[+] cells from YUMM1.7[OVA] N[TT] tumours, R[TT] control (CTRL) tumours, R[TT] *Ptgs1/2* KO tumours and R[TT] IRF3/7 overexpressing tumours 72 h after ACT (Fig. 4a and Extended Data Fig. 6a–e). The most substantial TME changes after deletion of *Ptgs1/2* or overexpression of IRF3/7 were in the monocyte and macrophage (MoMac) compartment. In R[TT] IRF3/7 tumours, inflammatory monocytes were the predominant population, together with a TAM cluster with enhanced stimulatory functions (TAM H2-Ab1) that was absent in R[TT] tumours (Fig. 4a and Extended Data Fig. 6d). In R[TT] *Ptgs1/2* KO tumours, we observed an increase in both monocytes and inflammatory monocytes, with a reduction in immunosuppressive TAMs (*Spp1*[+], *C1q*[+] and *Ctsk*[+]) and an increase in H2-Ab1 TAMs. After PGE$_2$ reduction or IRF3/7 overexpression, cDC1 abundance and functionality, as well as CCR7[+] DCs, were significantly increased and antigen presentation capacity was enhanced (Fig. 4a and Extended Data Fig. 6c–f). Last, NK cells were also rescued in R[TT] *Ptgs1/2* KO tumours and in IRF3/7 overexpressing tumours, but NK cell

depletion in *Ptgs2* KO tumours did not significantly change the infil-tration of cDCs and T cells or overall tumour control in our models (Extended Data Fig. 6g–j).

To investigate the distinctive features of PGE$_2$ depletion or IFN-I response reinstatement in patients with cancer, we established a sig-nature of the top upregulated genes in immune cells after *Ptgs1/2* KO (TME-COX signature, $n = 26$) and the top upregulated genes after IRF3/7 overexpression (TME-IRF3/7 signature, $n = 40$; Supplementary Table 4). Both signatures were strongly correlated with a gene expression signa-ture of CD8[+] T cell infiltration and were highly predictive of survival in patients with melanoma treated with ICB[49] (Extended Data Fig. 7a,b). Furthermore, both signatures were enriched in a group of patients who responded to subsequent tumour-infiltrating lymphocyte (TIL) therapy[43] (Fig. 4b and Extended Data Fig. 7c). From these results, we conclude that PGE$_2$ depletion and IFN-I cytokine increase is associated with an immune-permissive state of the TME and better response to therapy in preclinical models and in patients.

## IFN-I and IFNγ drive the inflammatory state

The TME of N[TT] tumours, R[TT] *Ptgs1/2* KO tumours and R[TT] IRF3/7 tumours share an inflammatory state, and monocytes are the most abundant cell type in this transcriptional state (Fig. 4c). After ACT, inflamma-tory monocytes strongly expanded, especially in N[TT] tumours and R[TT] *Ptgs2* KO tumours, which suggested that both cancer-cell-derived IFN-I cytokines and T cell-derived IFNγ play a part in inducing the inflamma-tory state (Extended Data Fig. 6b). Indeed, looking at the most variable genes in the immune compartment of the TME before and after ACT, a core IFNγ response gene set (for example, *Cxcl9*, *Gbp2* and *Slamf7*) was strongly increased in N[TT] tumours. Independent of the effects of T cell transfer, a set of genes characteristic of an IFNα response (for example, *Irf7*, *Isg15* and *Cxcl10*) was higher in N[TT] TMEs than in R[TT] TMEs, which suggested that both IFN-I and IFN-II cytokines modulate an inflamma-tory TME in N[TT] tumours (Extended Data Fig. 7d,e).

To assess how interferons modulate the inflammatory TME and monocytes, we selectively blocked IFN-I and IFN-II cytokines before and after ACT. Blocking IFN-I signalling before ACT prevented the for-mation of inflammatory monocytes. Specifically, low levels of IFN-I cytokines produced by R[TT] *Ptgs2* KO cells were sufficient for inducing an inflammatory state. After ACT, IFNγ depletion substantially reduced (N[TT]) or completely abolished (R[TT] *Ptgs2* KO) inflammatory monocytes (Fig. 4d and Extended Data Fig. 7f). In N[TT] tumours that produce high levels of IFN-I cytokines, the combinatorial depletion of IFN-I and IFN-II cytokines was necessary to fully abolish inflammatory monocytes after ACT (Fig. 4d). These data indicate that monocytes initially rely on tumour-derived IFN-I cytokines to transition to an inflammatory state, which shifts towards IFNγ when T cells infiltrate the TME.

The expansion of inflammatory monocytes after T cell transfer is coupled with a reduction in macrophages, which suggests that there is a shift in myeloid differentiation towards inflammatory monocytes at the expense of TAM maturation (Extended Data Fig. 6b). To further understand myeloid differentiation trajectories, we performed RNA velocity analysis of the MoMac compartment in N[TT] tumours, R[TT] tumours and R[TT] *Ptgs1/2* KO tumours. The analysis predicted that a common monocyte precursor population (monocyte 1 cluster) can give rise to both TAMs and inflammatory monocytes (Fig. 4e and Extended Data Fig. 7g). This results suggests that in R[TT] tumours, high PGE$_2$ and low IFN-I cytokine levels drive the differentiation of monocytes into suppressive macrophages, whereas in N[TT] tumours, monocytes are maintained, with a subset acquiring the inflammatory state. Indeed, treatment of Ly6C[+] monocytes with PGE$_2$ in vitro promoted their dif-ferentiation towards F4/80[+] macrophages (Extended Data Fig. 7h). In the absence of PGE$_2$ (R[TT] *Ptgs1/2* KO), TAMs were reduced and monocytes were increased, with a fraction becoming inflammatory (Fig. 4a). In R[TT] IRF3/7 tumours, despite high levels of PGE$_2$, the increased levels of

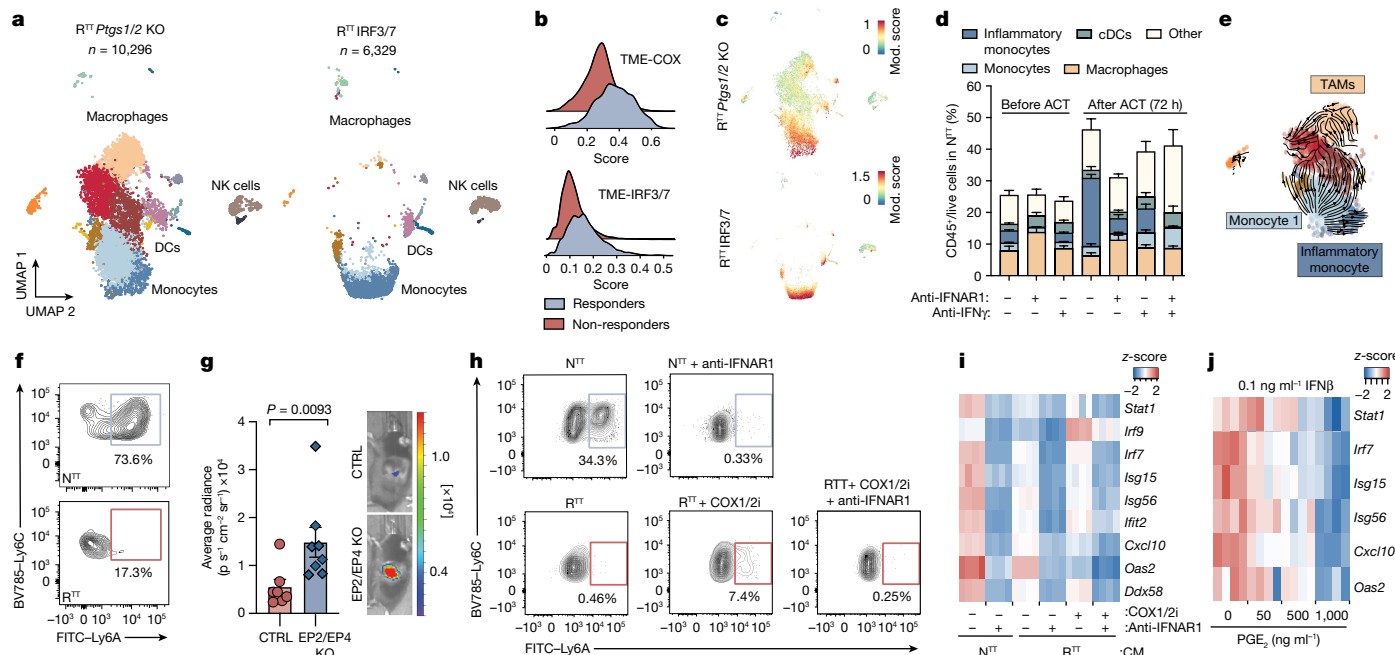

**Fig. 4 | PGE₂ and IFN-I responses determine myeloid cell abundance and their functional inflammatory state in the TME. a**, UMAP of scRNA-seq of CD45⁺ cells in YUMM1.7^OVA R^TT *Ptgs1/2* KO and R^TT IRF3/7 tumours 72 h after ACT (*n* = 3 tumours pooled per group). See Fig. 1c for cell cluster annotation. **b**, Scoring of the TME-COX and TME-IRF3/7 signatures (Supplementary Table 4) in myeloid fractions of responder (*n* = 6) and non-responder (*n* = 7) patients before TIL infusion[43]. **c**, Scoring of an ISG signature[37] in the scRNA-seq from **a**. **d**, Flow cytometry quantification of myeloid populations normalized to the CD45⁺ fraction from YUMM1.7^OVA N^TT tumours in *Rag2^−/−* mice treated with anti-IFNAR1 or anti-IFNγ (*n* = 5 tumours per group, except *n* = 4 in anti-IFNAR1 + ACT and ant-IFNγ + ACT). **e**, RNA velocity of the MoMac compartment from R^TT *Ptgs1/2* KO tumours. **f**, Representative contour plots of Ly6C⁺ monocytes depicting Ly6A expression 72 h after i.t. transfer into N^TT and R^TT tumours in *Rag2^−/−* mice

(*n* = 5 tumours per group). **g**, Left, BLI quantification of T cells 96 h after ACT of YUMM1.7^OVA R^TT cells into *Cd11c^cre-Ptger2^−/− Ptger4^fl/fl* mice (EP2/EP4 KO) or *Cd11c^cre* (CTRL) mice (*n* = 7 mice per CTRL group and *n* = 8 mice per EP2/EP4 KO group); two-tailed Mann–Whitney *U*-test. Right, representative BLI image (key as in Fig. 1b). **h**, Representative contour plots of Ly6C⁺ monocytes depicting expression of Ly6A 48 h after treatment with CM from cancer cells with or without a COX1/2i (indomethacin) and with or without anti-IFNAR1 or isotype (*n* = 3 biological replicates). **i**, Heatmap of scaled ISG expression in BMDCs exposed to CM from N^TT or R^TT cells and with or without a COX1/2i in the presence of anti-IFNAR1 or isotype (*n* = 4 technical replicates) measured by RT–qPCR. **j**, Heatmap of scaled ISG expression in BMDCs treated with IFNβ and PGE₂ measured by RT–qPCR (*n* = 4 technical replicates). Bar graphs depict the mean ± s.e.m.

IFN-I cytokines mediated inflammatory monocyte formation (Fig. 4a). To further analyse monocyte differentiation trajectories in vivo, we intratumorally injected bone-marrow-derived Ly6C⁺ monocytes. In N^TT tumours, 70–80% of these became inflammatory compared with only 10–20% in R^TT tumours (Fig. 4f and Extended Data Fig. 7i). Thus, both cancer cells and T cells shape the immune status of the TME by driving the differentiation of monocytes and their inflammatory state in therapy-responsive tumours.

## PGE₂ impairs the inflammatory state

To rule out that high levels of PGE₂ in the TME directly impair T cell function[30,50], we knocked out the PGE₂ receptors EP2 (encoded by *Ptger2*) and EP4 (encoded by *Ptger4*) in OT-1 T cells, followed by their intravenous injection into R^TT tumour-bearing mice. This strategy had modest effects and did not achieve tumour control (Extended Data Fig. 7j). We then examined effects of PGE₂ on the myeloid compartment using *Cd11c^cre*(*Itgax-cre*)*Ptger2^−/− Ptger4^fl/fl* mice, in which EP2 and EP4 are selectively ablated in CD11c⁺ cells. We observed improved T cell infiltration, and R^TT tumours were controlled in a CD8-dependent manner in both the YUMM1.7 and YUMM3.3 models (Fig. 4g, Extended Data Fig. 7k and Extended Data Fig. 8a). In line with these data, inflammatory monocytes and cDC2s, along with cDC1s, were increased in *Cd11c^cre*(*Itgax-cre*)*Ptger2^−/− Ptger4^fl/fl* mice compared with control mice (Extended Data Fig. 8b). These data suggest that disruption of PGE₂ signalling, either by inhibiting PGE₂ production in cancer cells or

by blocking downstream signalling in myeloid cells, is sufficient to restore the inflammatory state in myeloid cells and subsequent T cell function.

Given that IFN-I cytokines and PGE₂ represent distinct biological classes of mediators, we investigated their individual and combined effects on the inflammatory state of myeloid cells. We exposed mouse Ly6C⁺ monocytes and bone-marrow-derived DCs (BMDCs) to conditioned medium (CM) from N^TT cells, R^TT cells and R^TT IRF3/7 cells. CM from N^TT cells or R^TT IRF3/7 cells, but not R^TT cells, induced an inflammatory state in BMDCs and monocytes (Fig. 4h,i and Extended Data Fig. 8c–e). This inflammatory response was blocked by anti-IFNAR1 treatment. Using CM of R^TT cells treated with a COX1 and COX2 inhibitor (COX1/2i) also increased the inflammatory state, which fully depended on the marginal remaining IFN-I cytokines produced by R^TT cells (Fig. 4h,i and Extended Data Fig. 8c–e). Furthermore, inhibiting the MAPK pathway with a MEKi in R^TT cells during conditioning of the medium also resulted in an increase in ISG expression in BMDCs, thereby highlighting the role of oncogenic MAPK signalling in regulating PGE₂ and IFN-I cytokine production (Extended Data Fig. 8f). Exposure of human monocytes to CM from N^TT cells or R^TT cells of various human melanoma and NSCLC cell lines, with or without COX1/2i, demonstrated effects similar to those in mouse monocytes. This result provides support for a role of counter-regulated PGE₂ and IFN-I cytokines in myeloid dysfunction in human cancers (Extended Data Fig. 8g). Finally, low IFNβ levels were sufficient to induce ISG expression and the inflammatory marker AXL in BMDCs, and adding PGE₂ significantly impaired their response to

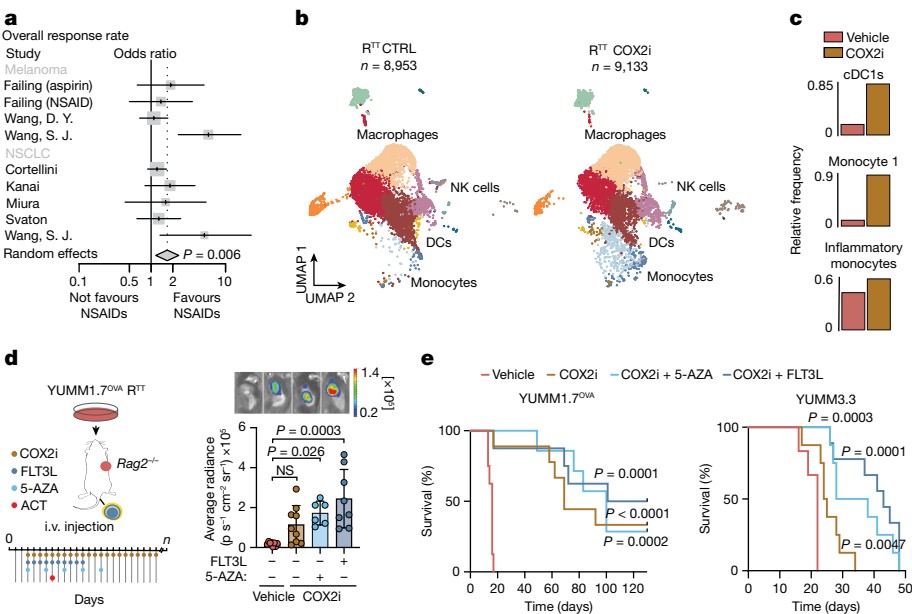

**Fig. 5 | Pharmacological modulation of PGE$_2$ and IFN-I responses reinstates an immune-permissive TME and immunotherapy response. a**, Forest plot of pooled odds ratios and 95% CIs across clinical studies for overall response rates in patients receiving ICB with or without NSAID co-medication ($n = 722$ patients over 8 independent cohorts; Extended Data Fig. 9a and Supplementary Table 5). Statistical analysis was performed with a random effects model and data are presented as the mean ± 95% CI. **b**, UMAP of scRNA-seq of CD45$^+$ cells of R$^{TT}$ CTRL and COX2i-treated YUMM1.7$^{OVA}$ R$^{TT}$ tumours 72 h after ACT ($n = 3$ tumours pooled per condition). See Fig. 1c for cell cluster annotation. **c**, Relative frequency of cell types across conditions as assessed by scRNA-seq. **d**, Left, treatment schedule of YUMM1.7$^{OVA}$ R$^{TT}$ tumours in *Rag2$^{-/-}$* mice with celecoxib (COX2i) in

combination with FLT3L or 5-AZA. Right, BLI quantification. Top, representative BLI images 72 h after ACT for vehicle ($n = 8$ mice), COX2i ($n = 9$ mice), COX2i + 5-AZA ($n = 6$ mice) and COX2i + FLT3L ($n = 8$ mice) groups (key as in Fig. 1b). Bar graphs depict the mean ± s.e.m. One-way ANOVA with Tukey's multiple comparisons test. **e**, Left, Survival of *Rag2$^{-/-}$* mice bearing YUMM1.7$^{OVA}$ R$^{TT}$ tumours treated with ACT and vehicle ($n = 8$ mice), COX2i ($n = 9$ mice), COX2i + 5-AZA ($n = 7$ mice) or COX2i + FLT3L ($n = 8$ mice). Right, survival of C57BL/6 mice bearing YUMM3.3 R$^{TT}$ tumours treated with an anti-PD1 and anti-CTLA4 with vehicle ($n = 6$ mice), COX2i ($n = 8$ mice), COX2i + FLT3L ($n = 9$ mice) or COX2i + 5-AZA ($n = 9$ mice). Log-rank Mantel–Cox test.

IFN-I cytokines, thereby hampering the acquisition of an inflammatory state (Fig. 4j and Extended Data Fig. 8h).

## Pharmacological targeting of PGE$_2$ and IFN-I

Nonsteroidal anti-inflammatory drugs (NSAIDs) inhibit PGE$_2$ production by COX enzymes and are commonly used for pain management in patients with cancer. Retrospective studies[51,52] and our own meta-analysis revealed that NSAID co-medication significantly improves overall response rates to ICB in melanoma and in patients with NSCLC (odds ratio = 1.68, 95% confidence interval (CI) = 1.16–2.42, $P = 0.006$), as well as progression-free survival compared with patients receiving only ICB (Fig. 5a, Extended Data Fig. 9a–d and Supplementary Tables 5 and 6). Nevertheless, these benefits did not translate into durable responses or long-term survival (Extended Data Fig. 9c). To examine how COX2 inhibition affects the immune TME, we treated R$^{TT}$ tumour-bearing *Rag2$^{-/-}$* mice with celecoxib, adoptively transferred T cells intravenously and performed scRNA-seq (Fig. 5b). In line with findings in other models[53], COX2 inhibition significantly increased the total amount of cDCs, including cDC1s and CCR7$^+$ DCs, monocytes and their inflammatory state and immunostimulatory TAMs while reducing suppressive TAMs (Fig. 5b,c and Extended Data Fig. 10a–c). Continuous COX2 inhibition with celecoxib and etoricoxib increased CD8$^+$ T cell infiltration and led to shrinkage of R$^{TT}$ tumours in combination with ACT (Extended Data Fig. 10d). Notably, when COX2i treatment was stopped, tumours rapidly relapsed, which suggested that continuous TME remodelling is required for T cell restimulation (Extended Data Fig. 10d).

To further enhance the effects of COX2 inhibition, we explored different mechanism-based drug combinations aimed at inducing IFN-I responses or expanding antigen-presenting cells. To investigate

whether the induction of IFN-I responses in R$^{TT}$ tumours could synergize with COX2 inhibition, we used 5-azacitidine (5-AZA), a clinically used DNA methyltransferase inhibitor that induces IFN-I responses[54]. Treatment with 5-AZA and its combination with a COX2i led to strong repolarization of the immune TME, increasing inflammatory monocytes and T cell infiltration that in turn led to tumour regression in all *Rag2$^{-/-}$* mice receiving ACT (Fig. 5d and Extended Data Fig. 10e–h). Similar to COX2 inhibition, short-term treatment with 5-AZA only led to transient tumour control, which further underlined the need for a continuous stimulatory TME (Extended Data Fig. 10h).

Next, we combined ACT with a COX2i and FLT3L, a cytokine that promotes the expansion of cDCs[32,55]. This strategy increased inflammatory hubs (Extended Data Fig. 10i), improved T cell expansion and led to more durable responses compared with the COX2i alone (Fig. 5d,e and Extended Data Fig. 10h). Of note, FLT3L treatment alone did not lead to significant tumour control (Extended Data Fig. 10h). After 130 days, 3 out of 9 mice remained alive in the COX2i group, 4 out of 8 mice in the COX2i + FLT3L group and 2 out of 7 in the COX2i + 5-AZA group (Fig. 5e). Reinjection of YUMM1.7$^{OVA}$ R$^{TT}$ cells in long-term tumour-free mice did not form tumours, which indicated the presence of immune memory and T cell recall (Extended Data Fig. 11a). The benefit of combining a COX2i with FLT3L or 5-AZA was even more substantial in the YUMM3.3 R$^{TT}$ model, in which COX2i + ICB had modest effects, but the addition of FLT3L or 5-AZA induced tumour control and significantly improved survival (Fig. 5e and Extended Data Fig. 11b–e). Similar benefits of these combination therapies were observed in several KRAS-driven models, including the colorectal cancer model CT-26, the NSCLC model KPAR and the PDAC model EPP2 (Extended Data Fig. 11f–n). Altogether, our results demonstrate that an immune-evasive TME orchestrated by cancer-cell-derived PGE$_2$ and low IFN-I cytokines levels can be

pharmacologically targeted using rational therapy combinations (Extended Data Fig. 11o).

## Discussion

Although the lymph node has long been recognized as a crucial environment for determining CD8[+] T cell function, findings by us and others assign a complementary role to the TME[3–6,24]. In this study, we showed that inflammatory monocytes, which in human tumours correspond to CXCL9[+]CXCL10[+] macrophages, drive T cell restimulation in the TME. We demonstrated that PGE$_2$ and IFN-I responses, controlled by oncogenic MAPK signalling in cancer cells, disrupt this process. These findings provide mechanistic insights into the recent discoveries that CXCL10[+] macrophages[56], often present in immune hubs together with CD8[+] T cells in patients[5], and CXCL9:SPP1 macrophage polarity[24] are predictive of responses to ICB.

We showed that inflammatory monocytes exhibit immunostimulatory capacities, as evidenced by their expression of CXCR3 ligands (Cxcl9 and Cxcl10), which recruit and position T cells, and IL-15, which promotes the expansion and survival of the effector pool[3]. Unlike cDC1s, which cross-present antigens, inflammatory monocytes obtain intact pMHCI complexes from tumour cells through cross-dressing. Both inflammatory monocytes and cDC1s facilitated intratumoral T cell restimulation in our models. In future studies, it will be important to dissect these seemingly redundant functions of cDC1s and inflammatory monocytes and macrophages, and the contribution of other cell types, such as cDC2s[37] and CD4[+] T cells[4,57,58]. This information will help determine whether the signals provided by these cells, including different modes of TCR engagement (cross-presentation versus cross-dressing), enhance the effectiveness of intratumoral CD8[+] T cell restimulation or differentially affect T cell function, as has been shown for a subset of macrophages that promote T cell exhaustion by providing a suboptimal TCR trigger[59]. The plasticity, high abundance and short half-life of monocytes and macrophages render them promising therapeutic targets for boosting the efficacy of immunotherapies, in particular in tumours in which functional intratumoral cDC1s are limited, a common feature of immune-evasive tumours[14,34]. Ultimately, it will be pivotal to understand whether T cell restimulation within the TME is strictly required across tumour entities and/or modes of immunotherapy.

We showed that patients with NSCLC and patients with metastatic melanoma receiving ICB and NSAIDs concomitantly had improved therapy responses and progression-free survival[51,52]. However, the use of NSAIDs did not translate into long-term benefit, which is possibly due to incomplete inhibition of PGE$_2$ production or discontinuation of treatment, which, in our preclinical models led to rapid tumour regrowth. We propose mechanism-based interventions that combine immunotherapy with suppression of PGE$_2$ levels through a COX2i and increase in IFN-I responses or cDC function through 5-AZA or FLT3L administration, respectively. Given that combining these clinically approved agents may cause toxicity and/or chronic IFN-I signalling resulting in T cell exhaustion[60], further optimization in terms of drug timing and sequence will be required before use in patients. Previous studies have shown that PGE$_2$ limits inflammatory gene expression in infection models[61]. A thorough understanding of the pathways that underlie this process in intratumoral myeloid cells has the potential to reveal new therapeutic targets for counteracting immune evasion.

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

## Methods

### Cell lines

YUMM1.7 and YUMM3.3 mouse melanoma[62] cell lines (obtained from M. Bosenberg, Yale University) were cultured in Dulbecco's modified Eagle's medium (DMEM)–F12 produced in-house. A375, M249 (ref. 63) (obtained from J. Massague, MSKCC), KPAR[64] (obtained from J. Downward, Francis Crick Institute) and EPP2 (ref. 65) (obtained from J. Zuber, IMP) cell lines were cultured in DMEM (Gibco). LOX[48] (obtained from J. Massague, MSKCC), CT-26 (ref. 66) and NCI-H358 cell lines were purchased from the American Type Culture Collection and cultured in RPMI-1640 (Gibco). The NCI-H358 $R^{TT}$ derivative was generated by culturing NCI-H358 parental cells in the presence of 1 µM KRAS inhibitor (Amgen) for 90 days until cells became resistant. YUMM1.7$^{OVA}$ clones and all $N^{TT}$ and $R^{TT}$ derivatives were generated as previously described[15]. $R^{TT}$ BRAFi-resistant cancer cells (YUMM1.7 and YUMM3.3 model) and all genetically engineered derivatives were cultured continuously in 100 nM dabrafenib (Selleckchem). MEKi-resistant cancer cells were cultured continuously in 10 nM trametinib (Selleckchem). Human $N^{TT}$ and $R^{TT}$ melanoma cell line derivatives (A375, M249 and LOX) were generated as previously described[48], and $R^{TT}$ cells were maintained in culture on 1 µM vemurafenib (LC-Labs). HEK-293T cells were purchased from Takara (Lenti-X 293T, 632180) and cultured in DMEM high-glucose produced in-house. BMDCs were cultured according to an adapted version of a previously described protocol[67]. In brief, for the first 6–7 days, cells were cultured at a density of $1 \times 10^6$ cells per ml. On day 4, fresh medium was added to minimize cell death. After that, cells were either seeded for assays or counted and re-seeded at a density of 300,000 cells per ml. BMDCs were cultured in full T cell medium supplemented with 200 ng ml$^{-1}$ FLT3L-Ig (BioXcell) and 5 ng ml$^{-1}$ GM-CSF (in-house produced). Bone-marrow-derived Ly6C$^+$ monocytes were cultured in DMEM medium (Gibco). Human MONO-MAC-1 (obtained from J. Zuber, IMP) and BLaER-1 (ref. 68) (obtained from M. Gaidt, IMP) cell lines were cultured in RPMI-1640 (Gibco). All media for cell lines were supplemented with 10% FBS, 2 mM L-glutamine (Gibco) and 100 IU ml$^{-1}$ penicillin–streptomycin (Thermo Fisher). BLaER-1 and NCI-H358 cells were additionally supplemented with 1× sodium pyruvate. CD8$^+$ T cells were cultured in full T cell medium containing RPMI-1640 supplemented with 10% FBS, 2 mM L-glutamine and 100 IU ml$^{-1}$ penicillin–streptomycin, 1× sodium pyruvate (Gibco), 1× non-essential amino acids (Gibco), 20 mM HEPES (produced in-house) and 0.05 mM β-mercaptoethanol (Millipore). All cells were cultured at 37 °C and 5% $CO_2$. Cells were routinely tested negative for mycoplasma contamination. STR Profiling was performed in-house for the YUMM1.7, YUMM3.3, EPP2 and KPAR cell lines. Moreover, sensitivity to MAPK inhibitors was confirmed for A375, M249 and LOX (BRAFi), CT-26 (MEKi) and for NCI-H358 (KRAS inhibitor).

### Animal experiments and ethics

All mice were bred and housed in pathogen-free conditions with a housing temperature of 22 ± 1 °C, 55 ± 5% humidity and a photoperiod of 14 h of light and 10 h of dark. Within each experiment, age-matched and sex-matched groups were used. *B6.129S(C)-Batf3tm1Kmm/J* (*Batf3*$^{-/-}$) mice, *B6(Cg)-Zbtb46tm1(HBEGF)Mnz/J* (*zDC*-DTR) mice, *B6.Cg-Tg(Itgax-cre)1-1Reiz/J* (*Cd11c-cre*) mice and *NOD.Cg-Prkdcscid Il2rgtm1Wjl/SzJ* (NSG) mice were purchased from Jackson Laboratories. *B6.Cg-Rag2tm1.1Cgn/J Ly5.2* (*Rag2*$^{-/-}$), BALB/c and C57BL/6J mice were obtained from the Vienna Biocenter in-house breeding facility. *Itgax*$^{cre}$*Ptger2*$^{-/-}$*Ptger4*$^{fl/fl}$ mice were provided by J. Boettcher (TUM, Munich). For *Rag2*$^{-/-}$ *Batf3*$^{-/-}$ strain generation, *Batf3*$^{-/-}$ mice were crossed to *Rag2*$^{-/-}$ mice, and homozygous offspring (*Rag2*$^{-/-}$ × *Batf3*$^{-/-}$) were confirmed by genotyping and used in subsequent experiments to evaluate the lack of cDC1s in the context of ACT. For *Rag2*$^{-/-}$ *zDC*-DTR strain generation, *zDC*-DTR mice were crossed to *Rag2*$^{-/-}$ mice and homozygous offspring were confirmed by genotyping and used in subsequent experiments to evaluate the effects of DC depletion. For ACT experiments and injection of YUMM1.7$^{OVA}$ cell lines, *Rag2*$^{-/-}$ mice were used. For the injection of YUMM3.3, KPAR and EPP2 cell lines, C57BL/6 mice were used. For the injection of the CT-26 cell line, BALB/c mice were used. For the generation of BMDCs and Ly6C$^+$ monocytes, bones (femurs and tibias) were collected from in-house-bred C57BL/6 mice. For all above strains, mice were used between 6 and 12 weeks old. For OT-1$^{Luc}$ CD8$^+$ T cell isolation, 6–24-week-old OT-1$^{Luc}$ Thy1.1 mice[69] were used. All mouse experiments were performed according to our licence approved by the Austrian Ministry (GZ: MA58-2260492-2022-22; GZ: 340118/2017/25; BMBWF-66.015/0009-V/3b/2019; GZ: 801161/2018/17; and GZ: 2021-0.524.218 and their amendments). Mice were euthanized when the humane end point was reached (for example, weight loss > 20%, signs of distress and pain), when tumours displayed signs of continuous necrosis or when tumours reached the maximum allowed tumour volume of 1,500 mm$^3$.

### Tumour cell injections

For subcutaneous injections, mice were anaesthetized with 2–4% isoflurane. For the YUMM1.7$^{OVA}$ model and all its derivatives, $0.5–1 \times 10^6$ YUMM1.7$^{OVA}$ cancer cells were subcutaneously injected into the flank of each mouse in a volume of 50 µl. For contralateral experiments, alternating flanks were used for the injection of $N^{TT}$ and $R^{TT}$ cells to avoid preferential growth biases. For the YUMM3.3 model, $0.3–1 \times 10^6$ cells were subcutaneously injected in a volume of 50 µl. For the CT-26 model, $0.25 \times 10^6$ cells were subcutaneously injected in 50 µl. For the KPAR model $0.35 \times 10^6$ cells were subcutaneously injected in 50 µl. For the EPP2$^{Luc}$ cell line derivative, orthotopic injections were performed as previously described[65]. In brief, surgeries were performed under isoflurane (2–4%) anaesthesia on a heated plate. A small incision on the upper left quadrant of the shaved abdomen was made and the was spleen identified. After externalization of the pancreas, $1 \times 10^6$ cells were intrapancreatically injected. Organs were re-situated, and the peritoneum closed with a resorbable 6-0 Vicryl suture, followed by skin closure with sterile wound clips. Animals received intraperitoneal (i.p.) injections of 5 mg kg$^{-1}$ carprofen pre-emptively and every 12–48 h after surgery. The health status of mice was monitored daily, and the tumour burden was assessed by BLI. All cell lines were resuspended in PBS mixed 1:1 with Matrigel (Corning) in the final injection volume. Subcutaneous tumours were monitored by calliper measurements every 2–4 days, and tumour volume was calculated according to the following formula: volume = $(D \times d^2)/2$, in which $D$ and $d$ are the long and short tumour diameters, respectively.

### Isolation and activation of naive OT-1$^{Luc}$ CD8$^+$ T cells

Spleen and lymph nodes were isolated from OT-1$^{Luc}$ mice, and red blood cell lysis was performed with ammonium–chloride–potassium lysis buffer (Thermo Fisher) according to the manufacturer's protocol. T cell isolation was performed using a Magnisort mouse CD8$^+$ naive T cell enrichment kit (Thermo Fisher) according to the manufacturer's protocol. T cells were activated for the first 24 h by seeding them on a plate coated with 2 µg ml$^{-1}$ anti-CD3 (145-2C11, eBioscience) overnight, and adding 1 µg ml$^{-1}$ anti-CD28 (37.51, eBioscience) and 20 ng ml$^{-1}$ carrier-free IL-2 (BioLegend). T cells were expanded for approximately 6–7 days in the presence of IL-2 and maintained daily at a concentration of $1 \times 10^6$ cells per ml in fresh T cell medium.

### ACT, intratumoral injection and BLI

Unless otherwise specified, when tumours reached a volume of 100–150 mm$^3$, $4 \times 10^6$ in vitro-activated OT-1$^{Luc}$ CD8$^+$ T cells were i.v. injected into mice in a volume of 100 µl PBS. For i.t. injections, $4 \times 10^6$ in vitro-activated OT-1$^{Luc}$ CD8$^+$ T cells were injected in a volume of 50 µl PBS. For measuring T cell infiltration by BLI, D-luciferin (150 mg kg$^{-1}$, Goldbio) was injected retro-orbitally or by tail vein injection into anaesthetized mice, and mice were imaged with an IVIS machine (Caliper Life

Sciences) and analysed using Living Image software (v.4.4; Caliper Life Sciences). In N$^{TT}$ tumours, T cell recruitment to the tumour is detectable by BLI within 24–48 h. This initial recruitment is followed by a phase of T cell expansion, with peak BLI signals between 96 and 120 h. Hence, we depict 96 h post-ACT images (unless otherwise specified in figure legends) as a suitable time point to assess T cell expansion in immune-permissive TMEs.

### In vivo treatments

For treatment with ICB, mice were i.p. injected with anti-PD1 (clone RMP1-14, BioXcell) and anti-CTLA4 (clone 9D9, BioXcell) in 100 µl of PBS when tumours reached a volume of 150–200 mm³ (usually between 6 and 8 days after injection). The YUMM3.3 model was treated with 200 µg anti-PD1/anti-CTLA4, the CT-26 model with 100 µg anti-PD1, and the EPP2 model with 100 µg anti-PD1. ICB treatment was administered every 3 days and continued for at least for 3 weeks, as indicated in the figure legends. Control mice were treated with an isotype control antibody (rat IgG2a anti-trinitrophenol, clone 2A3, BioXcell, and mouse IgG2b, clone MPC-11, BioXcell). For COX2i treatment, celecoxib (LC Laboratories) was reconstituted in a 60:40 (DMSO to PEG400, dH$_2$O) mixture as previously described[53]. Etoricoxib (Sellekchem) was dissolved first in a small volume of DMSO and then in 1% sodium carboxymethyl cellulose. COX2i was given by oral gavage every day (30 mg kg$^{-1}$) in a volume of 200 µl. For both COX2i regiments (celecoxib and etoricoxib), the treatment was started at day 3 after injection, when tumours were palpable, and continued every day until the termination of the experiment. 5-AZA (Sigma-Aldrich) was reconstituted in DMSO to a stock concentration of 10 mg ml$^{-1}$ and further diluted in PBS for in vivo treatments and given as i.p. injections (1 mg kg$^{-1}$) in 100–250 µl every 3 days, as previously described[54]. For NK cell depletion, 200 µg anti-NK1.1 (clone PK136, BioXcell) was administered every 3 days through i.p. injections, starting at day 1 after tumor induction. NK cell depletion was confirmed by flow cytometry. For blocking T cell egress from the lymph node, mice were given an i.p. injection of 20 µg per mouse of FTY720 (Sigma) in 100 µl saline. Treatment was started on the day of T cell transfer and administered for 5–7 consecutive days. Control mice received saline injection. FLT3L (recombinant FLT3L-Ig, hum/hum, BioXCell) treatment (30 µg per mouse in 100 µl PBS i.p.) was started at day 3 after injection and administered every day for 9 consecutive days. In vivo IFNAR blockade was performed with InVivoMab anti-mouse IFNAR-1 (clone MAR1-5A3, BioXcell) and was administered i.p. (200 µg per mouse) in 100 µl. For IFNγ, the neutralizing anti-mouse IFNγ monoclonal antibody was used (clone XMG1.2, BioXcell). Treatment was started on the day of tumour engraftment and administered every 3 days. InVivoMab IgG1 isotype control (BioXCell) was used as the control. For experiments in which CD8 depletion was performed, mice were treated with 50 µg anti-CD8 (clone 2.43, in-house produced), whereas control mice were treated with isotype control (rat IgG2b anti-keyhole limpet haemocyanin, clone LTF-2) starting the day before tumour engraftment and then every 3 days.

### DC vaccination with BMDCs

BMDCs were cultured with FLT3L and GM-CSF as described above. At day 10–12 after isolation, DCs were activated overnight with polyI:C (5 µg ml$^{-1}$, Invitrogen), pulsed with recombinant SIINFEKL peptide (5 µg ml$^{-1}$, Genscript) and sorted by FACS on the basis of alive MHCII$^+$CD103$^+$CD11c$^+$ cells. Next, $1 \times 10^6$ cells in a volume of 50 µl PBS were i.t. injected. Control mice received 50 µl PBS. For DC vaccinations, 2 doses of i.t. injections were administered on day 4 and day 6 after tumour engraftment.

### In vivo depletion of DCs with diphtheria toxin

For generation of bone marrow chimeras, *Rag2*$^{-/-}$ Ly5.1 mice were preconditioned (2×5 Gy), before transferring back $10 \times 10^6$ bone marrow cells by i.v. injection. As donor mice, *Rag2*$^{-/-}$ Ly5.2 zDC-DTR mice were

used. After 8 weeks of reconstitution, mice were used for experiments. N$^{TT}$ cells were injected, and DCs were depleted by injecting 25 µg kg$^{-1}$ of body weight of diphtheria toxin (Sigma-Aldrich) i.p. in PBS, starting on the day of tumour engraftment and then every 3 days for 3–4 doses. Reconstitution efficiency and depletion of intratumoral DCs was confirmed by flow cytometry.

### Lentivirus generation and cell transduction

Lenti-X (HEK-293T) cells were transfected with 4,000 ng of the plasmid of interest, 2,000 ng of VSV-G plasmid and 1,000 ng of PAX2 plasmid using polyethylenimine (Avantor). Virus-containing supernatant was collected 24 h and 48 h after transfection and subsequently filtered through a 0.45 µm filter. The cell lines of interest were transduced with the collected virus mixed with 8 µg ml$^{-1}$ polybrene (Merck).

### Generation of CRISPR–Cas9 KO and overexpression cell lines

Doxycycline-inducible Cas9 (iCas9) clones from parental cell lines were generated to allow inducible expression of Cas9. sgRNAs were chosen on the basis of the best VBC score[70] (Supplementary Table 7) and were cloned into a vector containing a puromycin selection marker and mCherry or eGFP (hU6-sgRNA–PuroR–mCherry/eGFP). sgRNAs targeting the *ROSA26* locus were used as controls for KO cell lines. After transduction, cells were selected with puromycin (5–8 µg ml$^{-1}$) for 5 days. All sgRNA sequences are provided in Supplementary Table 7. For the generation of single-cell-derived clonal cell lines, cells were FACS sorted on the basis of the fluorescent marker on the sgRNA backbone, at 1 cell per well into 96-well plates. To avoid immunogenicity caused by antibiotic selection markers or fluorophores in the YUMM3.3 model, we transiently transfected the cell lines with an all-in-one vector containing Cas9, the sgRNA of interest and eGFP (U6-IT-EF1As-Cas9-P2A-eGFP). For transient transfection, 7,000 ng of the plasmid with polyethylenimine was used, and single-cell clones were established. For IRF3/7 overexpression, synthesized cDNA sequences were ordered from Twist Biosciences and cloned into two different expression vectors with distinctive selection/fluorescent markers (SFFV-IRF3–mCherry and SFFV-IRF7–PuroR). After transduction, cells were selected with puromycin (5–8 µg ml$^{-1}$ for 5 days) and bulk FACS-sorted on the basis of mCherry expression. The same cell line engineered with an empty vector containing an mCherry and a puromycin resistance cassette was used as a control. KO and overexpression of the target proteins was confirmed by genotyping, western blotting or quantitative PCR with reverse transcription (RT–qPCR). For the YUMM1.7 and YUMM3.3 *Ptgs2* KO cell lines, single-cell-derived clonal cell lines were generated, and several were tested in vivo for growth kinetics.

### EP2 and EP4 KO in T cells

sgRNAs targeting the *Ptger2* and *Ptger4* mouse genes were designed according to the VBC score[70] and cloned into a dual hU6-sgRNA-mU6-sgRNA-EF1α-mCherry-PuroR backbone (Supplementary Table 7). As a control, we used a sgRNA targeting a gene desert in chromosome 1. The lentiviral vector was produced as described above. T cells were isolated from Cas9–OT-1 mice, which were a gift from J. Zuber (IMP), as described above. Twelve hours after CD3/CD28 activation, T cells were spin-infected with the lentiviral vector containing the sgRNAs in a 1:1 ratio for 1 h at 32 °C and 800*g*. At 12 h after infection, T cells were removed from the activation plate, washed with PBS and cultured in the presence of 20 ng ml$^{-1}$ IL-2. Selection with puromycin was performed 30 h after viral transduction. Before ACT, mCherry levels were assessed, and KO was confirmed by functional in vitro assays.

### Flow cytometry and cell sorting

For flow-cytometry-based characterization of the TME, tumours were isolated between day 7 and 11 after injection, cut into pieces and digested for 1.5 h at 37 °C with collagenase A (1 mg ml$^{-1}$, Roche) and DNAse (20 µg ml$^{-1}$, Worthington) in unsupplemented RPMI-1640 medium. Digested tumours

were strained through a 70 µm filter and resuspended in FACS buffer (0.5% BSA and 2 mM EDTA in PBS). Fc-block was performed with anti-CD16/32 (clone 2.4G2, Pharmingen) for 10 min at 4 °C to avoid Fc-specific antibody capture, and staining for cell surface markers was performed for 30 min at 4 °C. For intracellular staining, a *Foxp3* Transcription Factor staining kit was used (eBioscience). Live/dead exclusion was performed by staining with the fixable viability dye eFluor780 (1:1,000, eBioscience). DCs were defined in most experiments as MHCII$^+$CD11c$^+$CD24$^+$ out of alive CD45$^+$ cells. cDC1s were identified as CD103$^+$CD11b$^-$ out of the total DCs, cDC2s as CD103$^-$CD11b$^+$ and inflammatory cDC2 as CD103$^-$CD11b$^+$AXL$^+$. AXL was previously described to identify inflammatory cDC2s[37]. Monocytes were defined as Ly6C$^+$CD11b$^+$F4/80$^-$, and inflammatory monocytes were identified as monocytes that were Ly6A$^+$. Ly6A was previously described to identify monocytes expressing high levels of ISGs[38]. Macrophages were defined as Ly6C$^-$F4/80$^+$Cd11b$^+$. Acquisition of the samples was performed using a BD LSR Fortessa machine (BD Biosciences) with FACS Diva software (v.9.0.1), and analysis was conducted using FlowJo software (v.10.8 or newer). For cell sorting, a BD Aria cell sorter (BD Biosciences) with FACS Diva software (v.9.0.1) was used.

## Antibodies for flow cytometry

The following antibodies (all anti-mouse) were used for flow cytometry stainings (target (clone, catalogue number, manufacturer, dilution)): AXL PE-Cy7 (MAXL8DS, 25-1084-82, eBioscience, 1:200); CD103 PerCP/cyanine5.5 (2E7, 121415, BioLegend, 1:100); CD103 PE (2E7, BioLegend, 121405, 1:100); CD11b APC (M1/70, 17-0112-81, eBioscience, 1:200); CD11b PerCP/cyanine5.5 (M1/70, 101229, BioLegend, 1:200); CD11c BV605 (HL3, 563057, BD Pharmigen, 1:100); CD11c FITC (N418, 117305, BioLegend, 1:100); CD24 BV510 (M1/69, 101831, BioLegend, 1:100); CD24 FITC (M1/69, 11-0242-82, eBioscience, 1:100); CD279/PD-1 BV785 (29F.1A12, 135225, BioLegend, 1:200); CD279/PD-1 FITC (29F.1A12, 135213, BioLegend, 1:200); CD40 APC (3/23, 124611, BioLegend, 1:200); CD45 BV711 (30-F11, 103147, BioLegend, 1:500); CD45 FITC (30-F11, 103107, BioLegend, 1:500); CD86 BV510 (GL-1, 105039, BioLegend, 1:100); CD3 BV605 (17A2, 564009, BD Horizon, 1:100); CD3 AF647 (17A2, 100209, BD Horizon, 1:100); CD3 AF488 (17A2, 100212, BD Horizon, 1:100); CD8a eFluor 450 (53-6.7, 48-0081-80, eBioscience, 1:100); CD8a AF647 (53-6.7, 128041, BioLegend, 1:100); MHCI (H-2Kb) APC (AF6-88.5.5.3, 17-5958-82, Bioscience, 1:200); MHCI (H-2Kb) PE (AF6-88.5.5.3, 17-5958-80, Bioscience, 1:200); MHCII (I-A/I-E) eFluor450 (M5/114.15.2, 48-5321-80, eBioscience, 1:200); MHCII (I-A/I-E) APC (M5/114.15.2, 107613, BioLegend, 1:200); NK-1.1 BV711 (PK136, 108745, BioLegend, 1:100); TCF1 PE (S33-966, 564217, BD Pharmigen, 1:50); TIM3 BV711 (RMT3-23, 119727, BioLegend, 1:100); CD88 PE (20/70, 135805, BioLegend, 1;100); Ly-6A/E (Sca-1) FITC (D7, 108105, BioLegend, 1:100); SIINFEKL-HK2B PE (25-D1.16, 12-5743-81, Invitrogen, 1:100); F4/80 PE (BM8, B123110, BioLegend, 1:200); and rat IgG1, K Isotype control PE (R3-34, 5546, BD Pharmigen). Further information is provided in Supplementary Table 8.

## RNA extraction of cancer cells sorted from tumours, in vitro cell lines and myeloid cells

Tumours were surgically removed between days 10 and 12 after injection. The tissue was processed as described above, and cancer cells were isolated by flow cytometry on the basis of alive, CD45$^-$ cells and a fluorescent marker. For cancer cell lines and in vitro assays with myeloid cells, cells were washed with PBS and snap-frozen in liquid nitrogen and kept at −70 °C until further processing. RNA was extracted using a magnetic bead-based RNA extraction protocol (in-house produced). In brief, cells were lysed and incubated with beads together with DNase I (NEB) followed by magnetic isolation. RNA was purified by further elution with nuclease-free water.

## RT−qPCR

Reverse transcription was performed for cDNA formation with 1 µg of RNA per sample utilizing a LunaScript RT SuperMix kit (NEB) according to the manufacturer's instructions. RT−qPCR was performed with 10 ng cDNA per sample either with Luna Universal qPCR master mix (NEB) or an in-house produced MTD qPCR Dye 2× HS master mix according to the manufacturer's protocol. Each sample included four technical replicates. The RT−qPCR reaction was carried out in a Bio-Rad CFX384 real-time cycler and contained 1 min of initial denaturation (95 °C) and 45 annealing cycles lasting 15 s at 95 °C and 30 s at 60 °C. The analysis of gene expression levels was determined by the quantification cycles (Cq). Internal controls and the housekeeping gene *GAPDH* were used to correct for differences in sample quality and to normalize expression values. qPCR primer pair sequences are listed in Supplementary Table 7.

## In vitro assays with BMDCs

For cancer cell CM experiments, supernatants (in full T cell medium) from confluent cancer cells were collected after 48 h, filtered through a 45 µm filter and frozen at −70 °C until further use. Full T cell medium was supplemented with 20 µM of the COX1/2i indomethacin (Selleckchem) or 5 nM of the MEKi trametinib (Selleckchem) for the evaluation of MAPK and COX1/2 activity before media conditioning. BMDCs were differentiated as described above and collected at day 6. Next, 0.5–1 × 10$^6$ cells were seeded in triplicate in a 12-well plate in CM and treated with 10 µg ml$^{-1}$ InVivoMab anti-mouse IFNAR-1 antibody or InVivoMab IgG1 isotype control (BioXCell). Cells were cultured for 24 h, collected and processed for flow cytometry analysis or RNA extraction. For treatment with PGE$_2$ and IFNβ, cells were collected at day 6 and seeded at a concentration of 0.5–1 × 10$^6$ cells per ml. Cells were treated for 24–48 h with recombinant PGE$_2$ (100 ng ml$^{-1}$, Sigma-Aldrich) and recombinant mouse/human IFNβ (R&D Systems) at the concentrations indicated in the corresponding figures. Same volumes of acetone and PBS were used as a control for PGE$_2$ and IFNβ, respectively.

## Isolation of bone-marrow-derived Ly6C$^+$ monocytes for intratumoral injection and in vitro assays

Ly6C$^+$ monocytes were directly isolated from the bone marrow of CD45.1$^+$ C57BL/6 mice using a monocyte isolation kit (Miltenyi Biotec) following the manufacturer's instructions. For intratumoral monocyte transfer, 1 × 10$^6$ monocytes were i.t. injected into N$^{TT}$ and R$^{TT}$ tumours established in CD45.2$^+$ *Rag2*$^{-/-}$ mice. Tumours were isolated for FACS analysis 72 h after intratumoral transfer. For in vitro assays to assess effects of PGE$_2$, Ly6C$^+$ monocytes were seeded at a density of 1 × 10$^6$ cells per ml and cultured in recombinant IL-4 and GM-CSF (both produced in-house) and exposed to 200 ng ml$^{-1}$ PGE$_2$ or vehicle for 3 or 5 days. For CM experiments, monocytes were seeded at a density of 1 × 10$^6$ cells per ml in CM obtained from N$^{TT}$, R$^{TT}$ or R$^{TT}$ IRF3/7 cells with or without 20 µM COX1/2i (indomethacin) during media conditioning and subsequently supplemented with or without 10 µg ml$^{-1}$ InVivoMab anti-mouse IFNAR1 anti-mouse (BioXCell) or isotype IgG1 control (BioXCell).

## In vitro monocyte co-culture assay

Ly6C$^+$Ly6A$^+$ or Ly6C$^+$Ly6A$^-$ monocytes were FACS-sorted from N$^{TT}$ tumours grown in *Rag2*$^{-/-}$ mice or BALB/c mice and co-cultured for 72 h with naive OT-1 T cells (1:3 ratio: 100,000 monocytes for 300,000 naive OT-1 cells) previously labelled with 0.25 µM CFSE for 30 min at 37 °C.

## In vitro human monocyte assays

BLaER-1 cells were transdifferentiated into monocytes as previously described[68]. In brief, BlaER-1 transdifferentiation medium was freshly prepared by adding 10 ng ml$^{-1}$ human recombinant (hr-)IL-3 (PeproTech), 10 ng ml$^{-1}$ hr-M-CSF (PeproTech) and 100 nM β-oestradiol (Sigma-Aldrich) to complete RPMI medium. Cells were resuspended in transdifferentiation medium and plated in a 12-well plate at 0.7 × 10$^6$ cells per ml. Cells were incubated at 37 °C for 5–6 days until mature monocytes were differentiated. For CM experiments, BLaER-1 or MONO-MAC-1 human monocytes were seeded at a density of 0.7 × 10$^6$ cells per ml in CM obtained from N$^{TT}$ or R$^{TT}$ cells from the human

melanoma cell lines A375, M249 and LOX or the human NSCLC cell line NCI-H358 with or without 20 µM COX1/2i (indomethacin) during media conditioning. Cells were cultured in CM for 24 h and collected for RNA extraction, as described above.

## Evaluation of pMHCI cross-dressing on monocytes

For mismatched MHCI haplotype experiments, $1 \times 10^6$ YUMM1.7$^{OVA}$ N$^{TT}$ cells from C57BL/6 origin (H-2K$^b$) were injected in the flank of BALB/c (H2-K$^d$) mice. BALB/c mice were treated with anti-CD8 (50 µg in 100 µl, in-house produced), whereas control mice were treated with isotype control (rat IgG2b anti-keyhole limpet haemocyanin, clone LTF-2) starting the day before tumour engraftment and then every 3 days to avoid T cell-mediated mismatched MHCI rejection of YUMM1.7 cells. On day 10, tumours were collected and processed for flow cytometry staining of H2-K$^b$ or FACS-sorted on the basis of Ly6A expression for in vitro assays.

## Sample preparation for scRNA-seq

For scRNA-seq experiments involving TME characterization, tumours were isolated at day 10 after injection (72 h after ACT) and were processed as described above. The CD45$^+$ live fraction was isolated by FACS, and approximately $1 \times 10^5$ cells were collected. For scRNA-seq of OT-1 T cells, tumours were isolated 5 days after i.t. injection of $4 \times 10^6$ T cells. Alive T cells were isolated from tumours by FACS for CD45$^+$CD3$^+$CD8$^+$ markers. Dissociated cell concentrations were measured using NucleoCounter NC250 (Chemometec) following the manufacturer's instructions. For scRNA-seq samples from experiments 3 and 4 (see below), a Chromium Next GEM Single Cell Fixed RNA Sample preparation kit was used according to the manufacturer's protocol. In brief, $1 \times 10^6$ cells were fixed for 22 h at 4 °C, quenched and long-term stored at −80 °C according to 10x Genomics Fixation of Cells & Nuclei for Chromium Fixed RNA profiling (CG000478) using a Chromium Next GEM Single Cell Fixed RNA Sample preparation kit (PN-1000414, 10x Genomics). About 250,000 cells per sample were used for probe hybridization using a Chromium Fixed RNA Kit, Mouse Transcriptome, 4rxn × 4BC (PN-1000496, 10x Genomics), pooled equally and washed following the Pooled Wash Workflow as described in the Chromium Fixed RNA Profiling Reagent kit protocol (CG000527, 10x Genomics). For all the other scRNA-seq samples, a Chromium Next GEM Single cell 3′ kit with Dual Index was used according to the manufacturer's instructions. GEMs were generated on Chromium X (10x Genomics) with a target of 10,000 cells recovered, and libraries prepared according to the manufacturer's instructions (CG000527, 10x Genomics). Sequencing was performed on NovaSeq S4 lane PE150 (Illumina) with a target of 15,000 reads per cell.

## scRNA-seq analysis of CD45$^+$ TME

CD45$^+$ immune cells were collected in four different 10x Genomics sequencing experiments. Experiment 1, Chromium Single Cell 3′ scRNA-seq samples were pre-processed using cellranger count (v.6.1.1) (YUMM3.3 samples: N$^{TT}$/108155 and R$^{TT}$/108157). Experiment 2, 3′ CellPlex multiplex experiment with 4 samples pre-processed using cellranger multi (v.6.1.1) (YUMM1.7$^{OVA}$ samples: NTT + ACT, RTT + ACT, RTT *Ptgs1/2* KO + ACT, RTT CTRL ROSA26 + ACT). Experiments 3 and 4, Chromium Flex multiplex experiments with 4 samples each pre-processed using cellranger multi (v.7.1.0) and the built-in Probe Set (v.1.0.1 mm10-2020-A). Experiment 3, YUMM1.7$^{OVA}$ samples: R$^{TT}$ mCherry CTRL, R$^{TT}$ IRF3/7, R$^{TT}$ COX2i and R$^{TT}$ COX2i + 5-AZA, all ACT treated. Experiment 4, YUMM1.7$^{OVA}$ contained biological replicates of experiment 2 samples and untreated YUMM1.7$^{OVA}$ samples (noA): NTTnoA/271221, RTTnoA/271222, NTT/271223 ACT, RTT/271224 ACT. The prebuilt 10x Genomics mm10 reference refdata-gex-mm10-2020-A was used. Further processing was performed in R (v.4.2.2) with Seurat (v.4.3.0). For generating a CD45$^+$ immune reference map, we integrated cells from the first three experiments as follows. The cellranger filtered feature−barcode matrices were used, retaining cells with more than 1,000 detected genes and less than 15% of mitochondrial and

less than 40% of ribosomal RNA reads. An integrated feature−barcode matrix from the three experimental batches was generated accounting for the inclusion of a probe-based assay by keeping genes found in at least five cells in each experiment and excluding ribosomal and mitochondrial genes. Data were log-normalized, scaled (regressing out the difference between the G2M and S phase signature scores), dimensionality reduction was performed using principal component analysis on the top 3,000 most variable genes, and batch correction across batches was performed using Harmony[71] (v.0.1.1). The 40 harmony embeddings were used for UMAP visualizations. The first 40 harmony dimensions were used to identify immune cell subclusters with a resolution of 0.5 that were further assigned to cell types using known markers and publicly available myeloid reference datasets[21,72]. Cells were scored for the expression of published signatures using the AddModuleScore function[73]. Wilcoxon rank-sum test implemented in Presto (v.1.0.0) was used to identify differentially expressed genes (DEGs). Seurat's reference-based mapping was used to predict cell-type identity and map cells of the biological replicate experiment to our annotated reference set using the FindTransferAnchors and MapQuery functions after a quality control process retaining cells between 1,000 and 4,500 detected genes for 27,1222 and 27,1224 cells, respectively, and 1,300 and 8,000 detected genes for 27,1221 and 27,1223 cells, respectively, and limiting count tables to the gene universe of the reference. Depth-normalized counts for pseudobulk and GSEA functional analyses of this experiment were generated using cellranger aggr. Differences between ACT and untreated conditions (no ACT) from the replicate experiment (experiment 4) were explored on a pseudo-bulk level in an unsupervised clustering analysis with heatmap visualization. The fibroblast cluster was removed before further processing. Sum aggregation on the depth-normalized UMI counts was followed by variance stabilizing transformation, selection of the 300 most variable genes, standardization, $k$-means clustering ($k = 3$) and Enrichr analysis against the Reactome_2022 using Enrichr. The relative frequency bar plots depict the changes in the relative abundance of a cell type across different experimental conditions. For each condition, we calculated the normalized abundance of a specific cell type by comparing the absolute number of the cell type to the absolute number of all cells in the same condition. This normalization accounts for differences in total number of cells captured between conditions. We then calculated the relative cell abundance of the cell type in all conditions of the experiment. This was done by comparing the normalized abundance of the cell type to the sum of normalized abundances of the same cell type across conditions of the experiment. This step produces values between 0 and 1 for each condition for each cell type, with the sum of these values across all conditions of the experiment equalling 1 for each cell type.

## scRNA-seq analysis intratumoral CD8$^+$ OT-1$^{Luc}$ T cells

Single-cell gene expression of isolated N$^{TT}$ and R$^{TT}$ T cells was assayed in a Chromium Flex experiment, and read processing was performed using cellranger multi (v.7.1.0) using probeset (v.1.0.1 mm10-2020-A). Cellranger-filtered feature−barcode matrices were used and further filtered to retain cells with more than 800 detected genes, less than 10% of mitochondrial and less than 10% of ribosomal RNAs reads, and removal of cells of contaminant clusters was identified using SingleR and ImmGen reference (fibroblasts, MoMac populations). Data were log-normalized and scaled, and dimensionality reduction was performed using principal component analysis on the top 2,000 most variable genes. Harmony was used for the integration of cells from different samples, and 15 harmony embeddings were used for UMAP visualizations. Published tumour single-cell data were used for signature scoring[29]. Gene lists are provided in Supplementary Table 2.

## RNA velocity analysis

To understand differentiation trajectories of myeloid cells within the TME, we performed RNA velocity analysis[74] of the MoMac compartment.

Loom files containing the splicing annotation were created for each sample using the velocyto run command from the package velocyto (0.17,17) with default parameters and with no masked intervals. The loom files were combined with the scRNA-seq object that had been filtered to keep the data for monocyte and macrophage populations (Monocyte_1, Monocyte_2, Infl_Mono, TAM_CCL6, TAM_Ctsk, TAM_C1q, TAM_H2-Ab1, TAM_Spp1 and TAM_cycling) and for each condition ($N^{TT}$, $R^{TT}$ and $R^{TT}$ *Ptgs1/2* KO). First-order and second-order moments were computed using scvelo (0.2.5) pp.moments (n_pcs = 30, n_neighbors = 30), and the dynamical model was run with default parameters. Python (v.3.8.12) was used.

## SCENIC analysis

Gene regulatory networks for each cell population in each condition were calculated using SCENIC[75]. The motif database used was mm9-tss-centered-10kb-7species.mc9nr.feather. The co-expression network was calculated using GENIE3. The gene regulatory network was built using SCENIC wrapper functions.

## Analysis of publicly available myeloid datasets and inflammatory signatures

For the melanoma and lung samples from a previously published[23] dataset (Gene Expression Omnibus (GEO) identifier GSE154763), the raw counts were pre-processed as described in the publication, and clustering was calculated using a resolution of 0.8. The monocyte and inflammatory monocyte gene set was derived by using the wilcoxauc() function from presto and by selecting the genes with a log fold change > 0.6 (Supplementary Table 2). Then, the gene symbols were converted to human symbols. The human inflammatory monocyte gene set was used to calculate an enrichment score per cluster. In brief, the gene average expression was calculated for each cluster in the LUNG and MEL datasets on normalized data. Then, an enrichment score was calculated using GSVA with the following parameters: minSize = 5, maxSize = 500, kcdf = "Gaussian". The projection of the signature on the UMAP embedding was done using the function AddModule-Score() and then plotting the resulting score using FeaturePlot() with min.cutoff = 0.3 for the inflammatory monocyte score and min.cutoff = 0.4 for the Monocyte_1 score. For another dataset[43], the annotated seurat-object for myeloid populations corresponding to the original figure 4a was obtained, and gene sets were analysed as described above. For querying published inflammatory gene signatures, a previously published ISG⁺ DC signature[37] was generated by taking the top DEGs in the cDC2 cluster[37]. The Bosteels Inf-cDC2 DC signature was previously generated[37] and was obtained by re-analysing the scRNA-seq dataset (GEO identifier GSM4505993), in which the top 20 DEGs were taken in the identified inflammatory cDC2 cluster. All of them were subsequently scored in our dataset using the AddModuleScore[73], and the resulting score was plotted using FeaturePlot(). Gene lists are provided in Supplementary Table 2.

## Single-cell spatial transcriptomics of human melanoma samples

Single-cell spatial transcriptomics profiling was performed using the CosMx technology (Nanostring). Biopsy samples were obtained from patients with an age at diagnosis that ranged from 24 to 85 years with a median of 66 years; 34% were women and 66% were men. We obtained cell-segmented data for 74 FOVs (an area of 500 × 500 μm) from tissue microarray cores of 34 melanoma metastases, in total consisting of 980 genes × 171,536 cells. Tumour samples were obtained from 21 lymph nodes, 7 subcutaneous metastases, 1 lung metastasis and 1 brain metastasis and 4 not annotated, from 31 patients containing 72 FOVs. Two FOVs were from tonsils as control. Most tumour tissue were from patients who were treatment-naive at the time of surgery. Tissue collection was approved by the Regional Ethics committee at Lund University (numbers 191/2007 and 101/2013). Patients provided informed consent. The majority of tissue microarray cores contained

tertiary lymphoid structures, and FOVs were preferentially directed to these regions. Low-quality FOVs, cells with <20 counts and potential multiplets of cells (area exceeding the sample geometric mean + 5 standard deviation) were discarded. Using Seurat, genes for which the mean expression was below 3× the median of the negative probe mean expression, and genes with the highest 99% quantile expression, *MALAT1* and *IGKC* (due to potential spillover to neighbouring cells), were removed, which retained 641 genes. The data were normalized using SCTransform[76], counts that were zero before SCTransform were restored, and counts were log-transformed as $\log_2$(counts+1). The top 30 principal components were used for UMAP reduction and clustering (k.param = 15, resolution = 0.5, Louvain algorithm). Resulting clusters were assigned to biological annotations using known marker genes, and annotations were mapped back to FOV coordinates. Expression of C1QC, CXCL9 or CXCL10 > 0 was considered as positive. Cell-type fractions were derived for each FOV. Pearson correlation values between cell-type fractions across FOVs were determined and displayed. In Fig. 2i, *CXCL9⁺CXCL10⁺* macrophage/DCs (number 9 and number 10) were either *CXCL9⁺* or *CXCL10⁺*. Macrophage/DCs (number 5) were negative for *CXCL9*, *CXCL10* and *C1qC*.

## Generation of the TME-COX and TME-IRF3/7 signature

For the TME-COX signature, the FindMarkers function was used in Seurat, with tresh.use = 0.25 and min.pct = 0.1, to compare $R^{TT}$ CTRL (ROSA26) and $R^{TT}$ *Ptgs1/2* KO scRNA-seq samples. The top DEGs ($\log_2$ fold change ≤ 1.5, adjusted *P* value < 0.05) were used and converted to human orthologues using DIOPT[77]. For the TME-IRF3/7 signature, the FindMarkers function was used in Seurat, comparing $R^{TT}$ CTRL (mCherry) and $R^{TT}$ IRF3/7 and taking the top 40 DEGs.

## TME signatures in immunotherapy-treated human samples

Gene expression data for patients receiving ICB were obtained from a previous study[49] (NCBI BioProject accession number PRJEB23709). The TME-COX, TME-IRF3/7 and CD8⁺ T cell scores for each tumour sample were defined as the geometric mean of the expression values of each of the gene sets, respectively (Supplementary Table 4). The univariate Cox proportional hazards models, in which the TME-COX and TME-IRF3/7 scores were included as continuous variables, were used for testing the statistical association between gene signature expression and patient survival, separately for both signatures. The tumour samples were then divided into three groups on the basis of the signature score (bottom third, mid-third and top third) and Kaplan–Meier plots were generated for visualization. The association between signature expression and CD8⁺ T cell abundance was evaluated by calculating the Person's correlation coefficient between the signature score and a CD8⁺ score for each signature separately. For this, all scores were normalized to a median of zero and standard deviation of one. The two overlapping genes were removed from the CD8⁺ signature before comparing it to TME-IRF3/7 signature expression. For evaluating the enrichment of TME-COX and TME-IRF37 gene signatures in responder and non-responder patients to TIL therapy (baseline) from a previous study[43], mouse gene identifiers were first converted to human orthologues (with DIOPT v.9; best dcore = yes, best score reverse = yes, DIOPT score > 7) and single-cell level signature enrichment scores for the 'humanized' gene sets were calculated using AddModuleScore_UCell[78].

## Analysis of transcriptomics data

For plots shown in Fig. 3a, a cut-off of adjusted *P* value < 0.05 and $\log_2$ fold change > 2 and < −2 was used on DEGs expressed in YUMM1.7^OVA $N^{TT}$ and $R^{TT}$ GFP⁺ cancer cells FACS-sorted out of tumours (Supplementary Table 3). Pathway enrichment analysis was performed using Enrichr[79,80]. For the plot in Extended Data Fig. 5i, upstream regulator analysis (Ingenuity)[81] was used to identify upstream regulators using DEGs with an adjusted *P* value < 0.05.

## Quantification of PGE₂ and IFNβ by ELISA

For in vitro analysis of $PGE_2$ production, $2 \times 10^6$ cells were seeded in 10 ml medium, and supernatants were collected after 48 h and kept at −70 °C until analysis. For IFNβ, $0.3 \times 10^6$ cells were seeded, and 1 ml of supernatant was collected from confluent cells in a 6-well plate after 48 h of culture and kept at −70 °C until analysis. For analysis of $PGE_2$ and IFNβ from mouse tumours, whole tumours were isolated between days 4 and 10 after engraftment, accurately weighed and immediately snap-frozen in liquid nitrogen. They were stored at −70 °C until further processing. For $PGE_2$ analysis, tumours were subsequently digested using a MACS dissociator according to the manufacturer's protocol in PBS supplemented with 1 mM EDTA and 10 μM indomethacin. Lysate was further diluted in dissociation buffer depending on the tumour condition and weight (100 μl per mg of tumour) and further quantified using a $PGE_2$ ELISA kit (Cayman) or a mouse IFNβ Quantikine ELISA kit (Biotechne) according to the manufacturer's protocol. Values were normalized by taking into account dilution factors and tumour weight. For human IFNβ analysis from human cells, $1 \times 10^6$ A375, M249, LOX or NCI-H358 cells were injected into NSG mice and collected on day 21. Tumours were processed as described above and quantified using a Human IFNβ Quantikine ELISA kit (Biotechne) according to the manufacturer's protocol.

## Eicosanoid analysis from tumours by HPLC−MS

YUMM1.7$^{OVA}$ N$^{TT}$ and R$^{TT}$ tumours were isolated at day 10 after injection and weighed, and a solution of isopropanol and methanol (1:1, v/v) was added to the tissue for metabolite extraction. The material was subsequently homogenized and incubated for 1 h at −20 °C. The samples were then centrifuged at 14,000$g$ for 3 min. A second extraction round was performed by adding 80% methanol and $H_2O$ (v/v) to the pellet and centrifuged, and both supernatants were combined. Finally, the samples were incubated for another 2 h at −20 °C, and after final centrifugation, the supernatants were stored at −70 °C until further analysis. Samples were subsequently measured on a ZIC-pHILIC column or a RP column. Metabolites were annotated using the compound discoverer 3.0 software (Thermo Fisher) using an internal database or the mzCloud database (at least 75% match on the basis of measured molecular weight and MS$^2$ spectra). For filtering, a RSD of corrected quality control areas was used, being less than or equal to 25%. Group CV of at least 1 group is less than or equal to 40%.

## Western blotting

Cells were lysed with RIPA buffer (Cell Signaling Technology) supplemented with complete Protease Inhibitor Cocktail (Sigma Aldrich) and HALT phosphatase inhibitor (Thermo Fisher Scientific). Lysates were sonicated and cleared by centrifugation at 14,000$g$ for 10 min at 4 °C. Protein concentrations were quantified according to the manufacturer's instructions using a BCA Protein Assay kit (Pierce, Thermo Fisher Scientific). Immunoblotting was conducted according to standard protocols. The primary antibodies used for immunoblotting were as follows: anti-vinculin (Sigma-Aldrich, 1:1,000), anti-COX2 (CST, 1:1,000) and anti-H3 (acetyl K27) (Abcam, 1:5,000). The secondary antibodies used were as follows: anti-rabbit IgG HRP-linked (Cell Signaling Technology, 1:10,000) and anti-mouse IgG HRP-linked (Cell Signaling Technology, 1:10,000).

## Volumetric IF microscopy and image analysis

Volumetric microscopy of mouse tumours was performed as previously described[9]. In brief, tumours were fixed in Antigenfix solution (Diapath) for 6–8 h, dehydrated in 30% sucrose overnight, embedded in TissueTek OCT freezing medium (Sakura Finetek) and stored at −80 °C. Using a Leica CM3050 S cryostat, consecutive sections of 50 μm thickness were generated, subsequently permeabilized, blocked and stained in 0.1 M Tris (Carl Roth) supplemented with 1% BSA, 0.3% Triton X-100 (Merck),

normal mouse serum (Merck) and donkey serum (Merck). Stained sections were mounted in Mowiol (Merck) and imaged on an inverted TCS SP8 confocal microscope (Leica) using a HC PL APO CS2 ×20/0.75 NA objective. Images were acquired as tiled image stacks, covering whole tumour sections in the $xy$ plane, with 2 μm $z$-spacing to provide 3D image volumes of at least 20 μm depth. For further analyses, images were adaptively deconvoluted using the Leica TCS SP8 LIGHTNING tool (v.3.5.7.23225) and analysed using Imaris 9.9 software (Oxford Instruments). The Imaris surface generation tool was used to reconstruct and visualize 3D objects for individual cells. Where indicated, signals outside rendered cells were masked to visualize intracellular proteins. For analysis of immune cell infiltration by histocytometry, statistics for object localizations were exported into Excel (v.16.88; Microsoft) and analysed using GraphPad Prism software (GraphPad). Quantification of the number of cells was performed relative to the volume of the imaged section. Interacting cells were described as being less than <5 μm apart from each other.

## Antibodies for immunofluorescence microscopy

The following antibodies were used for staining of mouse tissues: anti-CD3 (BioLegend, clone 17A2), anti-CD103 (R&D Systems, goat polyclonal), anti-FSCN1 (Santa Cruz Biotechnology, clone 55-k2), anti-Ly6C (BioLegend, clone HK1.4) and anti-MHCII I-A/I-E (BioLegend, clone M5/114.15.2). All antibodies were either validated by the manufacturer or were previously reported for IF microscopy. The populations were defined as follows: T cells (CD3$^+$), monocytes (Ly6C$^+$CD103$^-$MHCII$^+$), cDC1s (FSCN1$^-$CD103$^+$MHCII$^+$), CCR7$^+$ cDC1 (FSCN1$^+$CD103$^+$MHCII$^+$) and CCR7$^+$ cDC2 (FSCN1$^+$CD103$^-$MHCII$^+$). Nur77–GFP was directly assessed by transferring Nur77–GFP reporter OT-1 T cells.

## Meta-analysis of NSAID immunotherapy cohorts

The meta-analysis was performed in accordance with the updated Preferred Reporting Items for Systematic Reviews and Meta-analyses (PRISMA) reporting guidelines[82]. The literature search was conducted using the PubMed (MEDLINE) database and last updated on 31 December 2023. The full search strategy is available in Supplementary Table 6. The literature review included studies of (1) adult patients with (2) melanoma or NSCLC (3) undergoing FDA-approved immunotherapy, including anti-PD1, anti-PD-L1 or anti-CTLA4, (4) co-medication with NSAIDs and (5) available sufficient patients' outcome data to calculate odds ratios for overall response rates or hazard ratios for progression-free and overall survival. Patients were not excluded when receiving concomitant chemotherapy and/or radiotherapy. Included studies report time of overall survival, time of progressions-free survival and overall response rates (defined as complete responses and partial responses divided by patient population). All studies published since 1 January 2011 (FDA approval of first immunotherapy, for example, ipilimumab) were included. Survival data are reported as univariate or multivariate hazard ratios; if both were available, multivariate analysis was prioritized. Odds ratios and hazard ratios with 95% CIs for overall response rates, progression-free and overall survival from included studies were utilized to calculate the pooled odds and hazard ratios. The heterogeneity of the pooled results was evaluated using $Q$-tests to assess between-study heterogeneity and quantified by the Higgins $I^2$ test. If $P$ was <0.10 for the $Q$-test or $I^2$ was >50%, significant heterogeneity was assumed, and the random-effects model was used to summarize the data. Statistical analysis was performed using R software (v.4.3.2) with meta (General Package for Meta-Analysis, v.7.0-0).

## Statistical analysis and reproducibility

Statistical analyses were performed using GraphPad Prism (v.9.1.2 or newer) and Microsoft Excel (v.16.88). Normality of the data distribution was calculated using a D'Agostino and Pearson test or Shapiro−Wilk test. The number of samples ($n$) used per experiment and the statistical test used are indicated in the figure legends. All in vitro and in vivo

experiments were repeated at least twice and always with multiple replicates, except for the following experiments that were performed only once: scRNA-seq involving pharmacological treatment of the YUMM1.7 $R^{TT}$ model, intratumorally injected T cells and the YUMM3.3 model. IF stainings for which representative images are shown were repeated at least twice, except for the $N^{TT}$ in $Batf3^{-/-}$ and Nur77 reporter experiment, which was performed once but with $n = 3$ tumours and was also confirmed with flow cytometry. Pharmacological combination treatments of the KPAR model were performed once. No statistical methods were used to determine sample size for in vivo experiments, and numbers were chosen on the basis of previous preliminary experiments. Scientists were not blinded to experimental groups, and experiments were repeated by different investigators. Mice were randomly assigned to treatment groups on the basis of tumour size at the day of treatment start or randomly allocated across separate cages when treatment had to be started at day 3. $P$ values < 0.05 were considered significant.

## Reporting summary

Further information on research design is available in the Nature Portfolio Reporting Summary linked to this article.

## Data availability

Gene expression data from YUMM1.7 $N^{TT}$ and $R^{TT}$ cancer cells sorted from tumours have been previously deposited into the GEO with accession number GSE132443. Specifically, samples from $N^{TT}$ tumours (GSM3864154, GSM3864155 and GSM3864157) and $R^{TT}$ tumours (GSM3864170, GSM3864172 and GSM3864173) were used to generate plots in Fig. 3a and Extended Data Fig. 5i. Raw and processed files of the scRNA-seq generated in this study have been deposited into the GEO with accession number GSE241750. Expression data for patients receiving ICB are publicly available[49] (accession number PRJEB23709). scRNA data from gene signatures in responder and non-responder patients to TIL therapy[43], including the Seurat object for the myeloid compartment was obtained upon direct request. Expression data for the melanoma and lung samples are publicly available[23] (GSE154763) and the raw count matrix was obtained upon request. All other formats of raw data are available upon request. Source data are provided with this paper.

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

**Acknowledgements** We thank the members of A.C.O.'s laboratory and J.Z.'s laboratory (Research Institute of Molecular Pathology) for discussions and proofreading the manuscript; P. S. Jung for the help with the summary scheme; S. Balay, E. Mancheño and J. van der Veeken for critical reading of the manuscript; G. Jonsson and J. Penninger for providing *Itgax-cre* mice; J. Moon for providing OT-1$^{Luc}$ Thy1.1 mice; V. Buchholz for providing Nur77–GFP OT-1 T cells; M. Bosenberg for providing the parental YUMM1.7 and YUMM3.3 cell lines; Z. Zhang and S. Cheng for sharing the raw count matrix for their scRNA-seq analysis; M. Gaidt for sharing the BLaER-1 cell line; staff at the BioOptics facility at the Vienna Biocenter for cell sorting; T. Grentzinger and staff at the Next Generation Sequencing Facility at the Vienna Biocenter for scRNA-seq; T. Köcher from the Metabolomics Facility at the Vienna Biocenter for the metabolomic analysis of eicosanoids; and M. Kostic for editing the manuscript. Research in the laboratory of A.C.O. is supported by Boehringer Ingelheim, the Austrian Research Promotion Agency (headquarter grant FFG-852936), the European Research Council ('CombaTCancer', grant number 759590 and 'UnlockIT' grant number 101125797), and the Vienna Science and Technology Fund (LS-16-063). M.V. is supported by the SNF postdoctoral mobility grant (grant number 214359), the Marie Skłodowska–Curie postdoctoral fellowship programme (grant number 101108907) and the VIP$^2$ postdoctoral programme, which has received funding from the European Union's Framework Programme for Research and Innovation Horizon 2020 (2014-2020) (Marie Skłodowska–Curie grant agreement number 847548). This project has received funding from the European Union's Horizon 2020 research and innovation programme under the Marie Skłodowska–Curie grant agreement number 955951. S.V. is supported by the Sigrid Juselius Foundation. For the purpose of open access, the author has applied a CC BY public copyright licence to any Author Accepted Manuscript version arising from this submission.

**Author contributions** A.E., G.E. and A.C.O. conceived the study, designed the experiments and interpreted the results. A.C.O. supervised the study. A.E and G.E. developed experimental tools, performed in vitro experiments, in vivo treatment studies, flow cytometry analyses, gene expression profiling and parts of the computational analyses, and analysed the data. F.B. and A.-M.P. performed IF stainings. L.H.-H. established parental cell lines and phenotypes, experimental design and expression profiling. M.N. and L.C. analysed gene expression data, scRNA-seq results, including the SCENIC and RNA velocity analyses. E.P., F.A., T.V.N., M.M.i.O. and N.F. contributed to the generation of experimental tools and in vivo and in vitro studies. M.S. performed the meta-analysis of patient data. I.K. performed western blotting and helped with mouse colony maintenance and genotyping. F.B., M.V., S.M.C., S.R., F.H., J.Z. and T.W. contributed to experimental design, generation of experimental tools or in vivo experiments. G.C., D.D.L. and D.B. provided the TIL therapy of patients with melanoma scRNA-seq dataset. G.J. and M.L., provided spatial CosMx data and analysis of human melanoma samples. J.P.B. provided EP2/EP4 KO mice and supervised IF stainings and analyses. C.Q. and X.B. analysed patient data. S.V. analysed RNA-seq data and analysed signatures derived from the scRNA-seq in datasets of patients with melanoma. A.E., G.E. and A.C.O. wrote the manuscript, with input from S.V., J.P.B. and J.Z., and all authors read and approved it.

**Competing interests** The laboratories of A.C.O. and J.Z. received research support and funding from Boehringer Ingelheim. J.Z. is a founder and scientific advisor of Quantro Therapeutics. G.C. has received grants from Celgene, Boehringer-Ingelheim, Roche, BMS, Iovance Therapeutics, and Kite Pharma. The institution that G.C. is affiliated with has received fees for his participation on an advisory board or for presentation at a company-sponsored symposium from Genentech, Roche, BMS, AstraZeneca, NextCure, Geneos Tx, and Sanofi/Avensis. G.C. has patents in the domain of antibodies and vaccines targeting the tumour vasculature, as well as technologies related to T cell expansion and engineering for T cell therapy. G.C. has received royalties from the University of Pennsylvania regarding CAR T cell technology. G.C. is a named inventor on patent applications filed by the Ludwig Institute for Cancer Research pertaining to the subject matter disclosed herein. D.D.L. has received a grant from Hoffmann-La Roche. All other authors declare no competing interests.

**Additional information**
**Correspondence and requests for materials** should be addressed to Anna C. Obenauf.

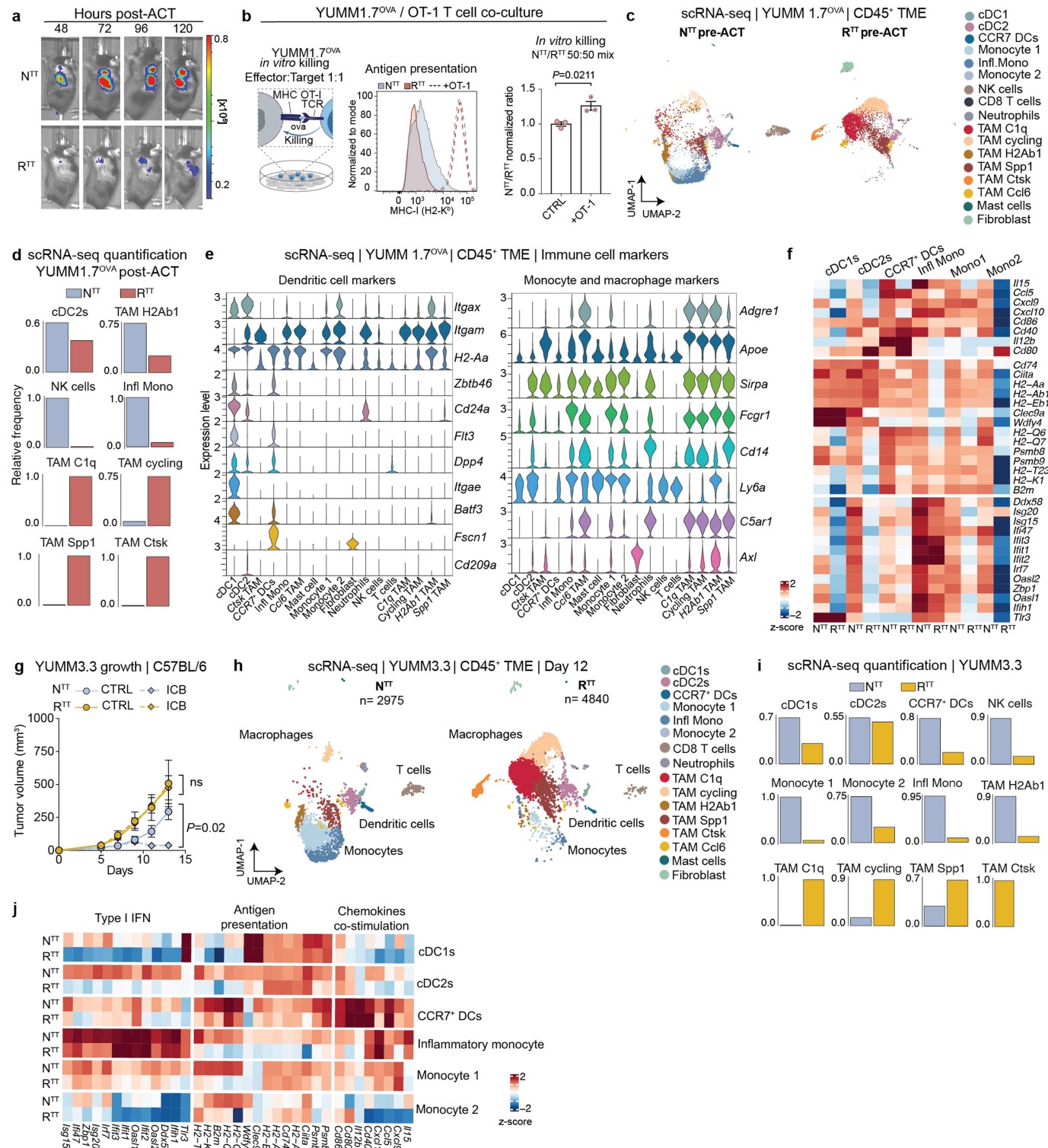

**Extended Data Fig. 1 | Characterization of immune permissive versus suppressive TMEs in mouse melanoma models. a**, Representative images of bioluminescent live imaging (BLI) of T cell infiltration after adoptive CD8+ OT-1[Luc] T cell transfer (ACT) in YUMM1.7[OVA] N[TT] and R[TT] tumors. **b**, Scheme outlining the in vitro killing of N[TT] and R[TT] cells with activated CD8+ OT-1 T cells (1:1 ratio) (left), histograms (flow cytometry) displaying MHCI levels of N[TT] and R[TT] cells (middle) and normalized ratio of 50:50 N[TT]/R[TT] cells mixed cultures in an in vitro killing assay with CD8+ OT-1 T cells. Normalization was performed with N[TT] and R[TT] percentages from the same 50:50 mixed cultures without CD8+ OT-1 T cells to account for proliferation differences (n = 3 biological replicates) (right). Two-tailed unpaired Student's t-test. **c**, UMAP of scRNA-seq of CD45+ cells of YUMM1.7[OVA] N[TT] and R[TT] tumors in Rag2[−/−] before ACT (n = 3 tumors pooled/group).

**d**, Relative frequency of cell types across conditions in YUMM1.7[OVA] tumors post-ACT (n = 4 tumors pooled/group). **e**, Expression of myeloid cell markers in each cluster (scRNA-seq). **f**, Heatmap of scaled gene expression (scRNA-seq) across cell clusters between N[TT] and R[TT] YUMM1.7[OVA] tumors post-ACT. **g**, Response to immune checkpoint blockade (ICB, anti-PD1/CTLA-4) of YUMM3.3 N[TT] and R[TT] tumors (n = 6 N[TT], n = 7 N[TT] + ICB, n = 8 R[TT], n = 9 R[TT] + ICB). Two-way ANOVA with Tukey's multiple comparison. **h**, UMAP of scRNA-seq of CD45+ cells of YUMM3.3 N[TT] and R[TT] (n = 1 tumor/group). **i**, Relative frequency of cell types across conditions. **j**, Heatmap of scaled gene expression (scRNA-seq) across cell clusters between YUMM3.3 N[TT] and R[TT]. Bar graphs and growth curves depict the mean ± s.e.m. ns = not significant.

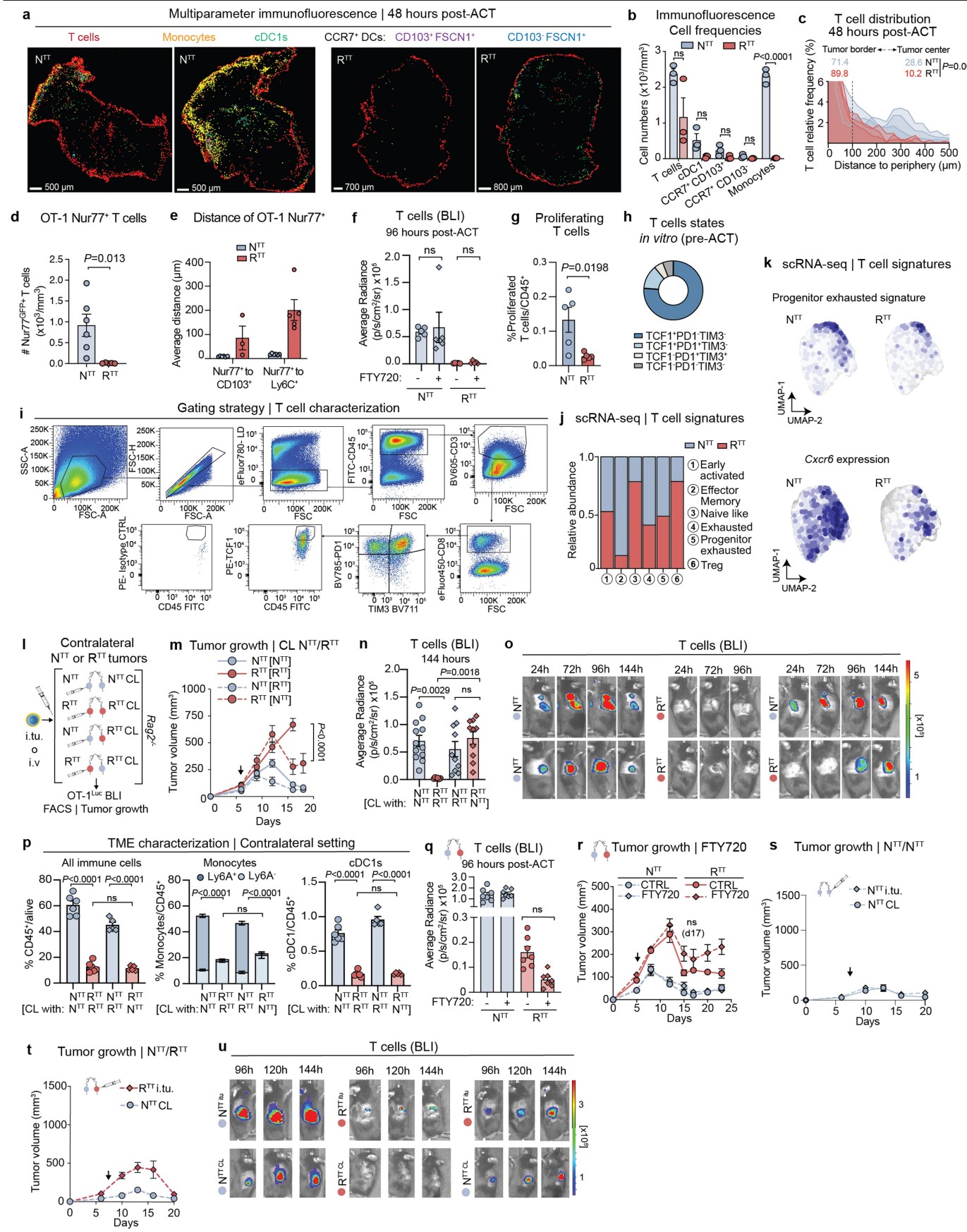

**Extended Data Fig. 2** | See next page for caption.

**Extended Data Fig. 2 | Permissive TMEs act as local reservoirs for T cell expansion. a**, Representative immunofluorescence (IF) of YUMM1.7$^{OVA}$ N$^{TT}$ and R$^{TT}$ tumors 48 h post-ACT ($n = 3$ tumors/group). Scale bars depicted in image. **b**, Quantification of immune populations based on IF staining ($n = 3$ tumors/group). **c**, Quantification of relative T cell frequency and distance to the tumor periphery in N$^{TT}$ and R$^{TT}$ tumors. Percentages of T cells found in the tumor border or tumor center are depicted ($n = 3$ tumors per condition). **d**, Quantification of Nur77$^+$ OT-1 cells in YUMM1.7$^{OVA}$ N$^{TT}$ and R$^{TT}$ tumors 48 h post-ACT ($n = 6$ tumors for N$^{TT}$, $n = 5$ for R$^{TT}$). **e**, Average distances between Nur77$^+$ OT-1 cells and cDC1s or monocytes based on IF stainings ($n = 6$ tumors for N$^{TT}$, $n = 5$ for R$^{TT}$). **f**, Quantification of CD8$^+$ OT-1$^{Luc}$ T cell infiltration by BLI post-ACT in YUMM1.7$^{OVA}$ N$^{TT}$ and R$^{TT}$ in $Rag2^{-/-}$ mice treated with FTY720 ($n = 5$ mice/group). **g**, Quantification of T cell proliferation by CFSE dilution via flow cytometry 72 h post intratumoral (i.tu.) T cell injection into N$^{TT}$ and R$^{TT}$ tumors in $Rag2^{-/-}$ ($n = 5$ mice/group). **h**, Relative frequencies of T cell phenotypes (flow cytometry) after activation in vitro prior to ACT ($n = 1$, experiment repeated 3 times). **i**, Gating strategy for the phenotypic characterization of T cells. **j**, Relative frequencies of CD8$^+$ T cell populations based on defined T cell signatures[29] of i.tu. injected T cells isolated and subjected to scRNA-seq 5 days post-transfer ($n = 14$ N$^{TT}$, $n = 18$ R$^{TT}$ pooled tumors). **k**, Projection of a progenitor exhausted T cell signature score[29] and *Cxcr6* gene expression on scRNA-seq from intratumoral T cells. **l**, Scheme outlining the contralateral (CL) injection of tumors in opposite flanks of the same mouse followed by intravenous (i.v) or i.tu. ACT in one of the tumors. **m**, Growth curves depicting the response to i.v ACT of primary or CL tumor in the same mouse ($n = 4$ mice N$^{TT}$/N$^{TT}$, $n = 3$ mice R$^{TT}$/R$^{TT}$ and $n = 6$ mice N$^{TT}$/R$^{TT}$). **n**, Quantification of T cell infiltration by BLI in mice harboring CL N$^{TT}$/N$^{TT}$ tumors ($n = 6$ mice), R$^{TT}$/R$^{TT}$ tumors ($n = 3$ mice) or an N$^{TT}$ and R$^{TT}$ tumor ($n = 10$ mice) after i.v ACT. **o**, Representative BLI images from Extended Data Fig. 2m, n. **p**, Flow cytometry characterization of the TME of tumors in the CL setting ($n = 3$ mice N$^{TT}$/N$^{TT}$, $n = 3$ mice R$^{TT}$/R$^{TT}$, $n = 5$ mice N$^{TT}$/R$^{TT}$ tumors). **q**, Quantification of T cell infiltration by BLI post-ACT in N$^{TT}$ and R$^{TT}$ in $Rag2^{-/-}$ mice treated with FTY720 ($n = 7$ mice/group). **r**, Response to ACT of mice bearing CL tumors (N$^{TT}$/R$^{TT}$) and treated with FTY720 or untreated (CTRL), ($n = 7$ mice/group). **s**, Response to ACT of $Rag2^{-/-}$ mice bearing CL tumors (N$^{TT}$/N$^{TT}$) with one of the N$^{TT}$ tumors injected i.tu ($n = 6$ mice). **t**, Response to ACT of $Rag2^{-/-}$ mice bearing CL tumors (N$^{TT}$/R$^{TT}$) with R$^{TT}$ tumors injected i.tu ($n = 6$ mice). **u**, Representative BLI images of T cell infiltration in mice harboring N$^{TT}$/N$^{TT}$, R$^{TT}$/R$^{TT}$ or R$^{TT}$/N$^{TT}$ tumors, with one of the tumors receiving i.tu. T cell transfer ($n = 6$ mice). Bar graphs and growth curves depict the mean ± s.e.m. Error bars in **s** and **t** not visible due to synchronous tumor regression. Statistical analysis was performed with a two-tailed unpaired student's $t$-test in **b**, **c**, **d**, and **g**. A one-way ANOVA with Tukey's multiple comparisons test in **f**, **n**, **p** and **q**. Two-way ANOVA with Tukey's multiple comparisons test in **m** and **r** on day 17 pi. ns = not significant. Arrow indicates day of ACT.

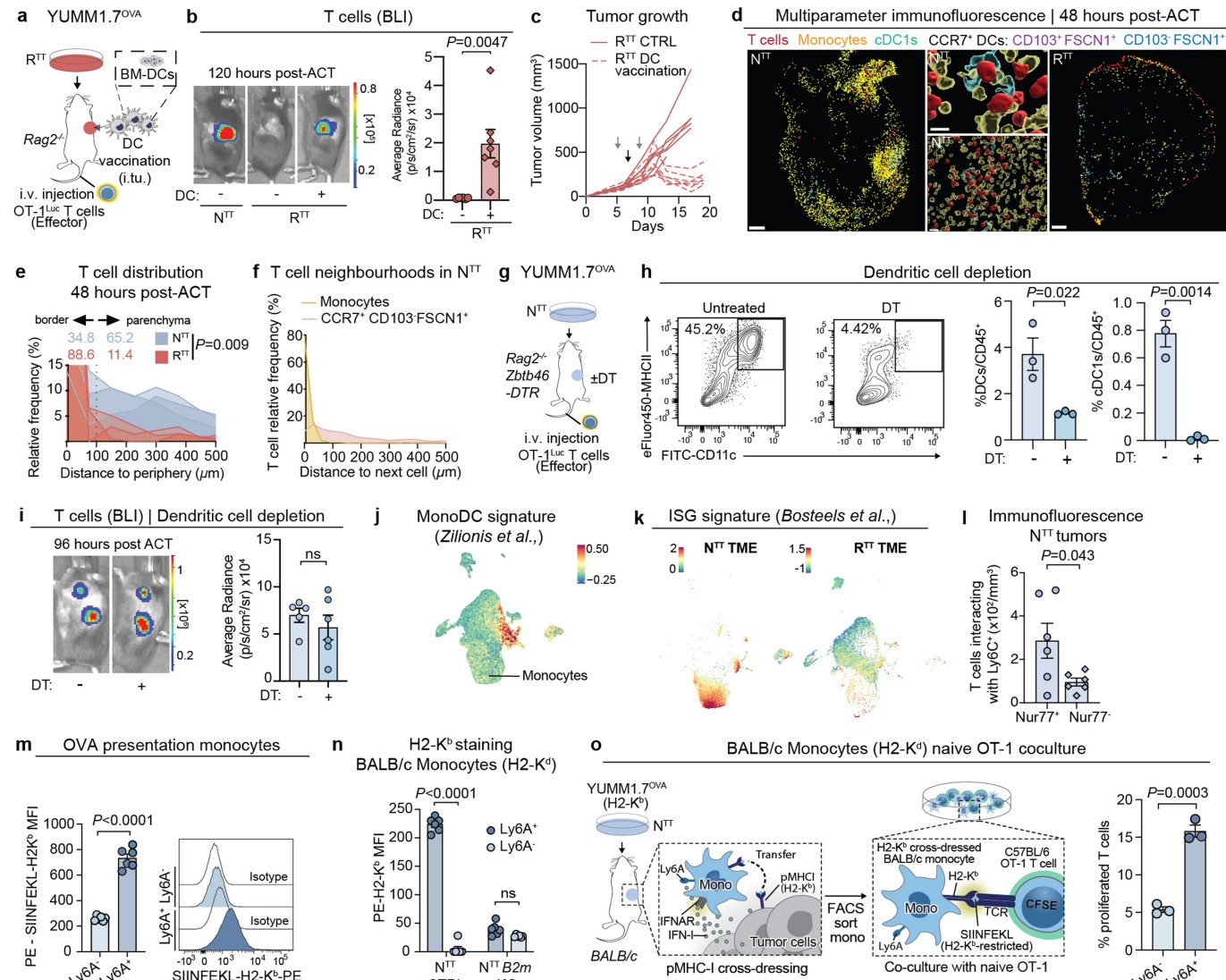

**Extended Data Fig. 3 | Inflammatory monocytes cross-dress with cancer-derived pMHCI complexes. a**, Scheme outlining the dendritic cell (DC) vaccination of YUMM1.7$^{OVA}$ R$^{TT}$ tumors with bone marrow-derived cDC1s (BM-DCs) injected i.tu. and treated with ACT. **b**, Left, representative BLI images and right, quantification of T cell infiltration by BLI ($n$ = 6 mice CTRL and $n$ = 7 mice for DC vaccination). **c**, Response to ACT of R$^{TT}$ tumors vaccinated with BM-DCs followed by ACT. Black arrow indicates the day of ACT and gray arrows the days of cDC1 vaccination ($n$ = 6 mice CTRL and $n$ = 7 mice for DC vaccination). **d**, Representative IF staining of YUMM1.7$^{OVA}$ N$^{TT}$ and R$^{TT}$ tumors in *Batf3$^{-/-}$ Rag2$^{-/-}$* mice, 48 h post ACT, scale bar in N$^{TT}$ = 500 μm, zoom-ins=20 μm and 10 μm, scale bar in R$^{TT}$ = 1000 μm, ($n$ = 3 tumors/group). **e**, Quantification of relative T cell frequency and distance to the tumor periphery in N$^{TT}$ and R$^{TT}$ tumors injected in *Rag2$^{-/-}$ Batf3$^{-/-}$* mice. Percentages of T cells found in the tumor border or tumor center are depicted ($n$ = 3 tumors/group). **f**, Quantification of relative T cell frequency and distance to the next immune cell in N$^{TT}$ tumors in *Rag2$^{-/-}$ Batf3$^{-/-}$* mice ($n$ = 3 tumors). **g**, Scheme outlining the depletion of dendritic cells with diphtheria toxin (DT) in N$^{TT}$ tumors in *Rag2$^{-/-}$* mice reconstituted with bone marrow from *Rag2$^{-/-}$ Zbtb46-DTR* mice. DT was

administered every 3 days starting on the day of tumor injection for the duration of the experiment. **h**, Contour plots depicting the frequency of dendritic cells (left) and flow quantification (flow cytometry) ($n$ = 3 tumors/group). **i**, Representative BLI images (left) and quantification of T cell infiltration ($n$ = 5 mice for untreated, $n$ = 6 DT treated) (right). **j**, Scoring of a monocyte derived dendritic cell (MonoDC) signature[21] in the YUMM1.7$^{OVA}$ scRNA-seq. **k**, Scoring of the *Bosteels et al.*[40] ISG signature (Supplementary Table 2) in the YUMM1.7$^{OVA}$ scRNA-seq. **l**, Quantification of Nur77$^+$ or Nur77$^-$ OT-1 T cells interacting with Ly6c$^+$ monocytes based on IF staining of N$^{TT}$ tumors 48 h post-ACT ($n$ = 6 tumors). **m**, Median fluorescence intensity (MFI) quantification of SIINFEKL-H2K$^b$ staining ($n$ = 6 mice/group) (left) and histograms (right). **n**, MFI quantification of N$^{TT}$-derived H2-K$^b$ on BALB/c pMHC-I cross-dressed monocytes ($n$ = 6 mice/group) in N$^{TT}$ and N$^{TT}$ *B2m* KO. **o**, Scheme of the process of cross-dressing and in vitro T cell activation (left) and quantification of OT-1 T cell proliferation after 72 h of co-culture with monocytes FACS-sorted from YUMM1.7$^{OVA}$ N$^{TT}$ tumors in BALB/c mice ($n$ = 3 technical replicates). Bar graphs depict the mean ± s.e.m. Statistical analysis was performed with a two-tailed unpaired student's *t*-test in **b, e, h, i, l, m, n** and **o**. ns = not significant.

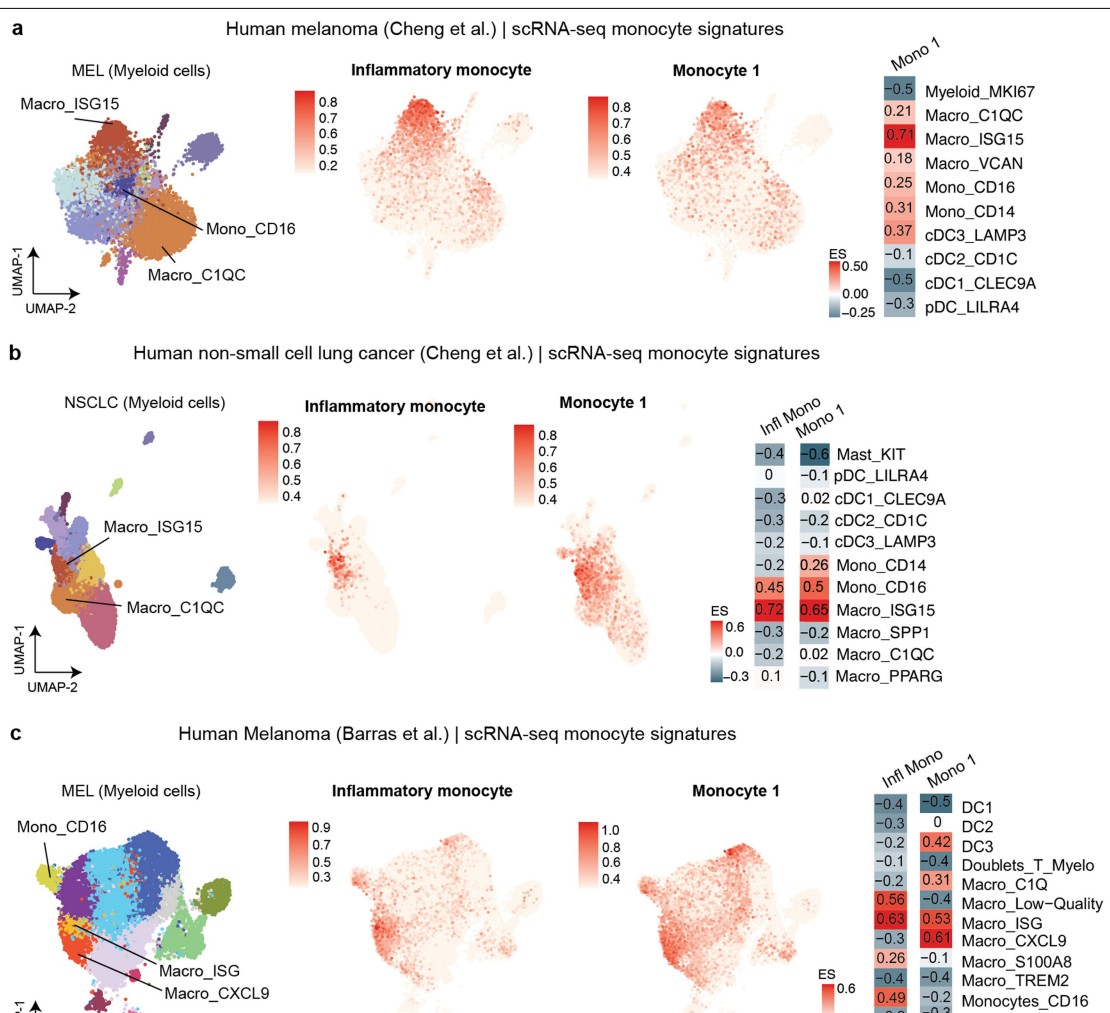

**Extended Data Fig. 4 | Inflammatory monocytes correspond to ISG⁺ and CXCL9⁺ macrophages in human data-sets. a**, Inflammatory monocyte and Monocyte 1 gene signature projection on a UMAP, and enrichment scores (ES) calculated with Gene Set Variance Analysis (GSVA) for different myeloid populations in a human melanoma scRNA-seq myeloid data-set (MEL) (*Cheng et al.*[23]). **b**, Inflammatory monocyte and Monocyte 1 gene signature projection on a UMAP, and ES calculated with GSVA for different myeloid populations, in human non-small cell lung cancer (NSCLC) scRNA-seq myeloid data-set (*Cheng et al.*[23]). **c**, Inflammatory monocyte and Monocyte 1 gene signature projection on a UMAP, and ES calculated with GSVA for different myeloid populations, in human melanoma scRNA-seq myeloid data-set (*Barras et al.*[43]). See also Supplementary Table 2.

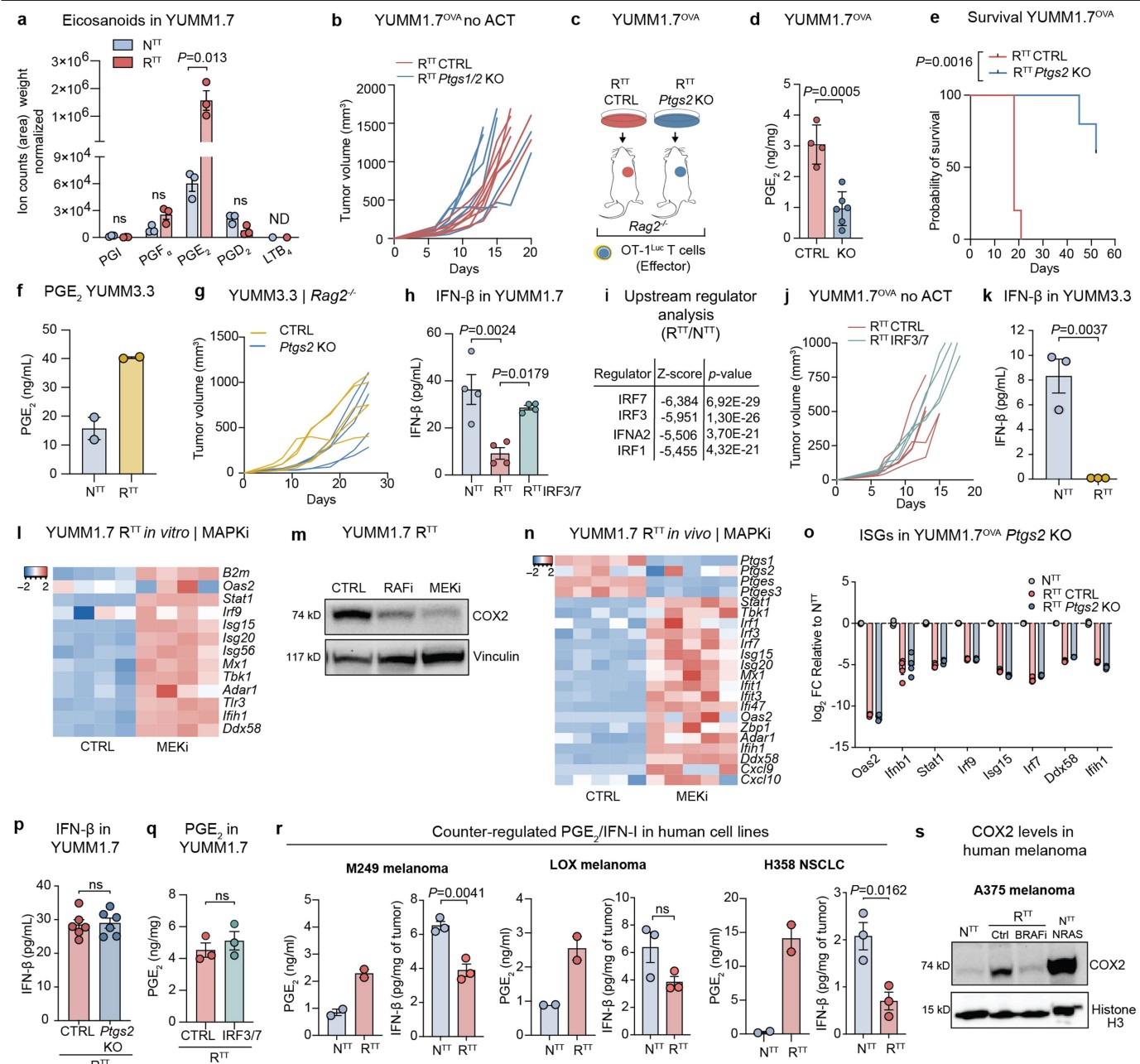

**Extended Data Fig. 5** | See next page for caption.

**Extended Data Fig. 5 | Oncogenic MAPK signaling counter-regulates the production of PGE$_2$ and IFN-I. a**, Targeted metabolomic analysis of eicosanoids in YUMM1.7$^{OVA}$ N$^{TT}$ and R$^{TT}$ tumors isolated at day 10 post-injection from *Rag2$^{-/-}$* mice (*n* = 3 tumors/group). **b**, Tumor growth of non-ACT R$^{TT}$ CTRL and R$^{TT}$ *Ptgs1/2* KO (*n* = 7 mice/group) in *Rag2$^{-/-}$* mice. **c**, Scheme outlining the injection of YUMM1.7$^{OVA}$ R$^{TT}$ *Ptgs2* KO into *Rag2$^{-/-}$* mice followed by ACT. **d**, Quantification of PGE$_2$ by ELISA from YUMM1.7$^{OVA}$ R$^{TT}$ *Ptgs2* KO tumors isolated at day 10 post-injection (*n* = 4 for R$^{TT}$ CTRL and *n* = 6 for R$^{TT}$ *Ptgs2* KO, over 2 independent experiments). **e**, Survival of *Rag2$^{-/-}$* mice harboring YUMM1.7$^{OVA}$ R$^{TT}$ CTRL or R$^{TT}$ *Ptgs2* KO tumors upon ACT treatment (*n* = 5 mice/group). **f**, Quantification of PGE$_2$ by ELISA from YUMM3.3 N$^{TT}$ and R$^{TT}$ tumors (*n* = 2 biological replicates). **g**, Tumor growth of YUMM3.3 R$^{TT}$ CTRL and R$^{TT}$ *Ptgs2* KO (*n* = 5 mice/group) in *Rag2$^{-/-}$* mice. **h**, Quantification of IFN-β by ELISA from YUMM1.7$^{OVA}$ N$^{TT}$, R$^{TT}$ and R$^{TT}$ IRF3/7 tumors grown in *Rag2$^{-/-}$* mice normalized to tumor weight (*n* = 4 tumors/group, pooled from 2 independent experiments). **i**, Upstream regulator analysis (Ingenuity) of differentially expressed genes in cancer cells sorted from R$^{TT}$ vs. N$^{TT}$ YUMM1.7$^{OVA}$ tumors grown in *Rag2$^{-/-}$* mice. Benjamini-Hochberg correction for multiple testing. **j**, Tumor growth of non-ACT treated YUMM1.7$^{OVA}$ R$^{TT}$ CTRL (*n* = 5 mice) and R$^{TT}$ IRF3/7 (*n* = 4 mice) in *Rag2$^{-/-}$* mice. **k**, Quantification of IFN-β by ELISA from YUMM3.3 N$^{TT}$ and R$^{TT}$ tumors grown in *C57BL/6* mice normalized to tumor weight (*n* = 3 tumors/group). **l**, Heatmap of scaled ISG expression in YUMM1.7$^{OVA}$ R$^{TT}$ upon treatment with MEK inhibitor (MEKi) for

48 h. **m**, Western blot of R$^{TT}$ YUMM1.7$^{OVA}$ depicting COX2 protein levels upon treatment with RAF inhibitor (RAFi) or MEKi for 48 h. For gel source data see Supplementary Fig. 1. Experiment repeated 2 independent times. **n**, Heatmap of scaled gene expression in YUMM1.7$^{OVA}$ R$^{TT}$ cancer cells upon treatment with MEKi in vivo (*n* = 5 tumors per condition). **o**, ISG expression in YUMM1.7$^{OVA}$ R$^{TT}$ *Ptgs2* KO cell line compared to N$^{TT}$ and R$^{TT}$ CTRL by RT-qPCR (*n* = 4 technical replicates). All expression values are depicted as the log$_2$FC of the expression of each gene compared to N$^{TT}$. **p**, Quantification of IFN-β by ELISA from YUMM1.7$^{OVA}$ R$^{TT}$ CTRL and R$^{TT}$ *Ptgs2* KO tumors grown in *Rag2$^{-/-}$* mice normalized to tumor weight (*n* = 6 tumors/group). **q**, Quantification of PGE$_2$ by ELISA from YUMM1.7$^{OVA}$ R$^{TT}$ CTRL and R$^{TT}$ IRF3/7 tumors grown in *Rag2$^{-/-}$* mice (*n* = 3 tumors/group). **r**, Quantification of IFN-β and PGE$_2$ by ELISA from N$^{TT}$ and R$^{TT}$ variants of M249, LOX and H358 cell lines. For IFN-β analysis, tumors were grown in NSG mice and normalized to tumor weight (*n* = 3 tumors/group). For PGE$_2$, supernatants from in vitro cell lines were analyzed (*n* = 2 biological replicates/group). **s**, Western blot of R$^{TT}$, N$^{TT}$ and N$^{TT}$ NRAS variants of A375 human melanoma cell lines depicting COX2 protein levels upon treatment with RAFi for 48 h. For gel source data see Supplementary Fig. 1. Experiment performed once. Bar graphs depict the mean ± s.e.m. Statistical analysis was performed with a two-tailed student's *t*-test in **a**, **d**, **k**, **p**, **q** and **r** and one-way ANOVA with Tukey's multiple comparison in **h**. p-value in **e** was calculated using a Log-rank Mantel Cox test. ns = not significant. ND = Not detected.

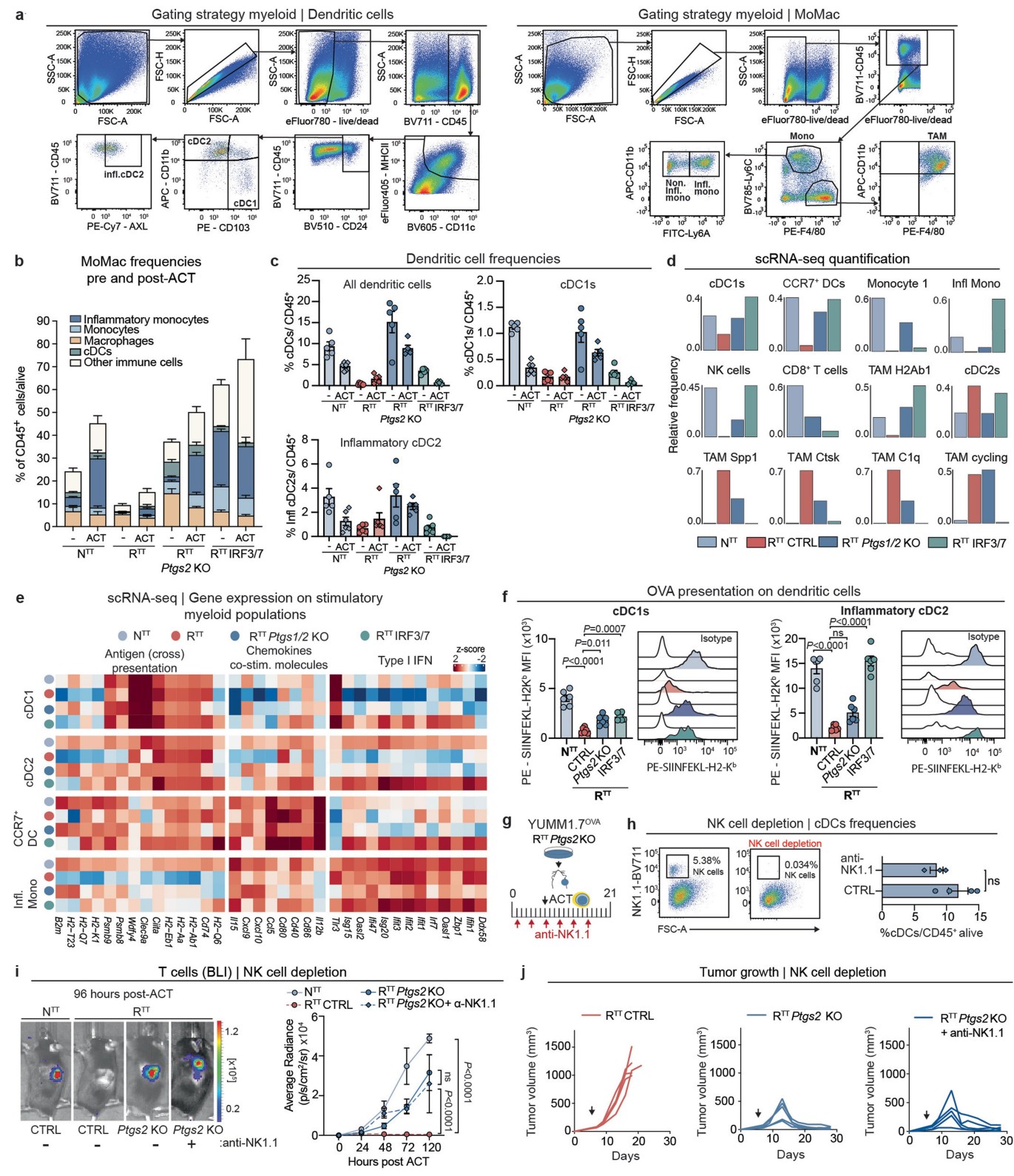

**Extended Data Fig. 6** | See next page for caption.

**Extended Data Fig. 6 | High PGE$_2$ and low IFN-I instruct an immune evasive TME. a**, Gating strategy for the identification of intratumoral myeloid populations. **b**, Flow cytometry myeloid characterization pre and 72 h post-ACT ($n$ = 6 for R$^{TT}$ ± ACT, IRF37 R$^{TT}$ ± ACT, $n$ = 5 for N$^{TT}$ ± ACT, R$^{TT}$ *Ptgs2* KO + ACT, $n$ = 4 R$^{TT}$ *Ptgs2* KO). **c**, Flow cytometry characterization of dendritic cell populations pre and 72 h post-ACT, ($n$ = 5 for N$^{TT}$, R$^{TT}$ *Ptgs2* KO ± ACT, IRF37 R$^{TT}$ + ACT, $n$ = 6 for N$^{TT}$ + ACT, R$^{TT}$ ± ACT, IRF37 R$^{TT}$). **d**, Relative frequency of cell types across conditions in YUMM1.7$^{OVA}$ (scRNA-seq), for R$^{TT}$ CTRL the R$^{TT}$ mCherry sample was used. **e**, Heatmap of scaled gene expression (scRNA-seq) between N$^{TT}$, R$^{TT}$ CTRL, R$^{TT}$ *Ptgs1/2* KO and R$^{TT}$ IRF3/7 for individual cell clusters. **f**, Flow cytometry characterization of SIINFEKL peptide on MHCI of dendritic cells isolated from tumors and pulsed ex vivo with SIINFEKL peptide ($n$ = 6 tumors/group) and MFI quantification. **g**, Scheme outlining in vivo NK cell depletion in *Rag2*$^{-/-}$ mice harboring YUMM1.7$^{OVA}$ R$^{TT}$ *Ptgs2* KO tumors. **h**, Representative plot depicting NK1.1$^+$ cells in NK cell-depleted vs CTRL R$^{TT}$ *Ptgs2* KO tumors measured by flow cytometry (left) and quantification of cDCs ($n$ = 4 CTRL and $n$ = 3 anti-NK1.1 treated tumors) (right). **i**, Representative BLI images (left) and quantification of CD8$^+$ OT-1$^{Luc}$ T cell infiltration by BLI (right), ($n$ = 3 mice for N$^{TT}$, $n$ = 5 mice for all other groups). **j**, Tumor response to ACT ($n$ = 5 mice/group). Arrow indicates day of ACT. Bar graphs and growth curves depict the mean ± s.e.m. Data in **i** depicts the mean ± s.e.m. Statistical analysis was performed with a two-tailed unpaired student's *t*-test in **h**. A one-way ANOVA with Tukey's multiple comparison in **f**. Two-way ANOVA with Tukey's multiple comparison in **i**. ns = not significant.

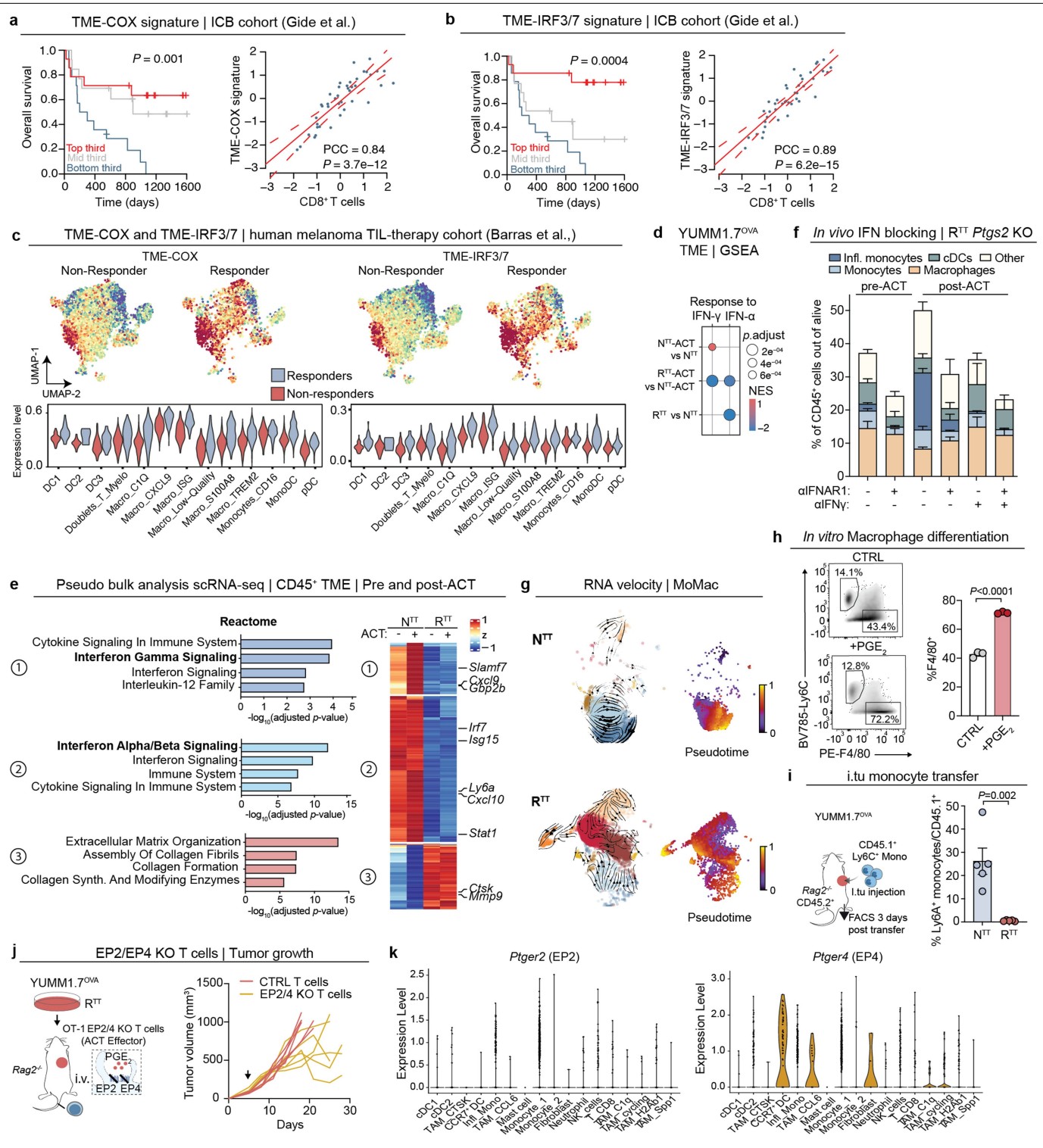

**Extended Data Fig. 7** | See next page for caption.

**Extended Data Fig. 7 | Type I and type II IFNs determine the inflammatory state. a, b,** Overall survival of melanoma patients stratified according to the TME-COX and TME-IRF3/7 signatures (Supplementary Table 4) (left) and correlation of intratumoral CD8+ T cell signature (right) in the *Gide et al.* RNA-seq data-set[49]. PCC=Pearson correlation coefficient. **c,** TME-COX and TME-IRF3/7 signatures projected on the myeloid UMAP from baseline tumors (split for responder and non-responders to TIL therapy) from the *Barras et al.*[43] patient cohort (above) and individual score across different myeloid populations (below). **d,** Gene set enrichment analysis (GSEA) of N^TT and R^TT tumors pre and post-ACT based on scRNA-seq analysis. Normalized enrichment score (NES). Multiple correction testing with false discovery rate. **e,** Pathway enrichment analysis of the top variable genes in the immune tumor microenvironment (TME) from N^TT and R^TT tumors pre and post-ACT analyzed by scRNA-seq. 1, 2 and 3 depict each gene cluster identified. Benjamini-Hochberg correction for multiple testing. **f,** TME characterization of YUMM1.7^OVA R^TT *Ptgs2* KO tumors pre and 72 h post-ACT upon αIFN-γ and αIFNAR1 treatment (*n* = 5 tumors/group,

except *n* = 4 for isotype and αIFNAR + ACT). **g,** Trajectory analysis and pseudotime of the MoMac compartment of YUMM1.7^OVA N^TT and R^TT tumors. **h,** Density plots of F4/80+ macrophages after exposure of BM-derived monocytes to PGE$_2$ for 48 h (left) and quantification (right) (*n* = 3 replicates per condition). **i,** Scheme outlining the intratumoral (i.tu.) transfer of CD45.1+ BM-derived monocytes into N^TT and R^TT tumors injected into CD45.2+ *Rag2*−/− mice (left) and inflammatory monocytes frequencies 3 days after transfer (right) (*n* = 5 tumors/group). **j,** Scheme outlining the knockout of EP2 (*Ptger2*) and EP4 (*Ptger4*) in OT-1 T cells and i.v. injection into YUMM1.7^OVA R^TT tumor-bearing *Rag2*−/− mice (left) and tumor growth (*n* = 6 mice/group) (right). Arrow indicates the day of ACT. **k,** Expression of *Ptger2* and *Ptger4* across populations from the YUMM1.7^OVA scRNA-seq. Bar graphs depict the mean ± s.e.m. Statistical analysis was performed with a two-tailed unpaired student's *t*-test in **h** and **i**. *p-value* in **a** and **b** was calculated using a Cox's proportional hazards model (survival) and with a two-sided Pearson's correlation (correlations).

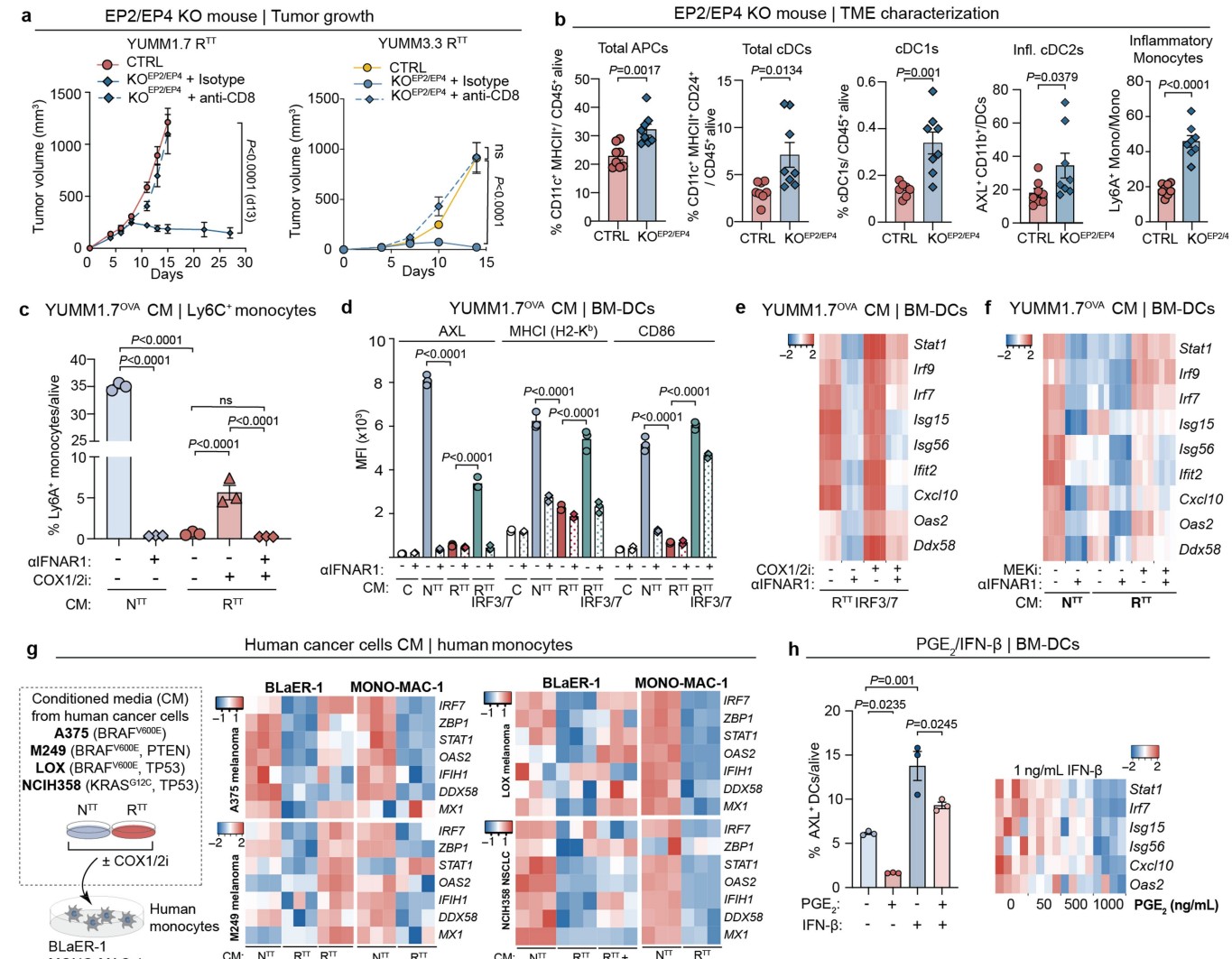

**Extended Data Fig. 8 | PGE₂ suppresses myeloid responsiveness to IFN-I.**
**a**, Tumor growth of YUMM1.7 (left) and YUMM3.3 (right) R^TT in *Itgax-Cre* (CTRL) (*n* = 7 mice for YUMM3.3 and *n* = 8 for YUMM1.7) or CD11c^Cre^(*Itgax*-Cre)*Ptger2*^−/−^/ *Ptger4*^fl/fl^ treated with anti-CD8 to deplete CD8⁺ T cells (*n* = 5 mice for YUMM3.3 and *n* = 6 for YUMM1.7) or isotype control (*n* = 7 mice for YUMM3.3 and *n* = 10 for YUMM1.7). **b**, TME characterization (flow cytometry) of R^TT tumors in *CD11c^Cre^-Ptger2^−/−^Ptger4^fl/fl^* mice (KO^EP2/EP4^) or *CD11c^Cre^* control mice (CTRL) (*n* = 8 tumors/group). **c**, Quantification of Ly6A⁺ cells after exposing BM-derived Ly6C⁺ monocytes to conditioned media (CM) from cancer cell lines ± COX1/2 inhibitor (COX1/2i, indomethacin) during media conditioning ± αIFNAR1 during 48 h of culture (*n* = 3 biological replicates per condition). **d**, MFI quantification of AXL, MHC-I and CD86 (flow cytometry) of BM-DCs treated with fresh media (C) or CM from N^TT, R^TT and R^TT IRF3/7 cells for 24 h ± αIFNAR1 or isotype control (*n* = 3 biological replicates per condition). **e**, Heatmap of scaled expression of ISGs in BM-DCs exposed to CM from R^TT IRF3/7 treated

with COX1/2i (indomethacin) or untreated, in the presence of αIFNAR1 or isotype control measured by RT-qPCR (*n* = 4 technical replicates). **f**, Heatmap of scaled ISG expression in BM-DCs exposed to CM from N^TT and R^TT treated with MEKi or untreated, in the presence of αIFNAR1 or isotype control (*n* = 4 technical replicates) measured by RT-qPCR. **g**, Scheme outlining the culture of human monocyte models to CM from cancer cell lines ± COX1/2i during media conditioning (left) and heatmaps of scaled ISG expression (right). **h**, Quantification of AXL⁺ BM-DCs treated with IFN-β and PGE₂ (flow cytometry) (left) (*n* = 3 biological replicates) and heatmap of scaled ISG expression measured by RT-qPCR (right) (*n* = 4 technical replicates). Bar graphs and growth curves depict the mean ± s.e.m. Statistical analysis were performed with a two-way ANOVA with Tukey's multiple comparisons test in **a**, a two-tailed unpaired student's *t*-test (APCs, cDC1 and infl.monocytes) and two-tailed Mann Whitney U (total cDCs and infl.cDC2s) in **b**, one-way ANOVA with Tukey's multiple comparison in **c**, **d** and **h**. ns = not significant.

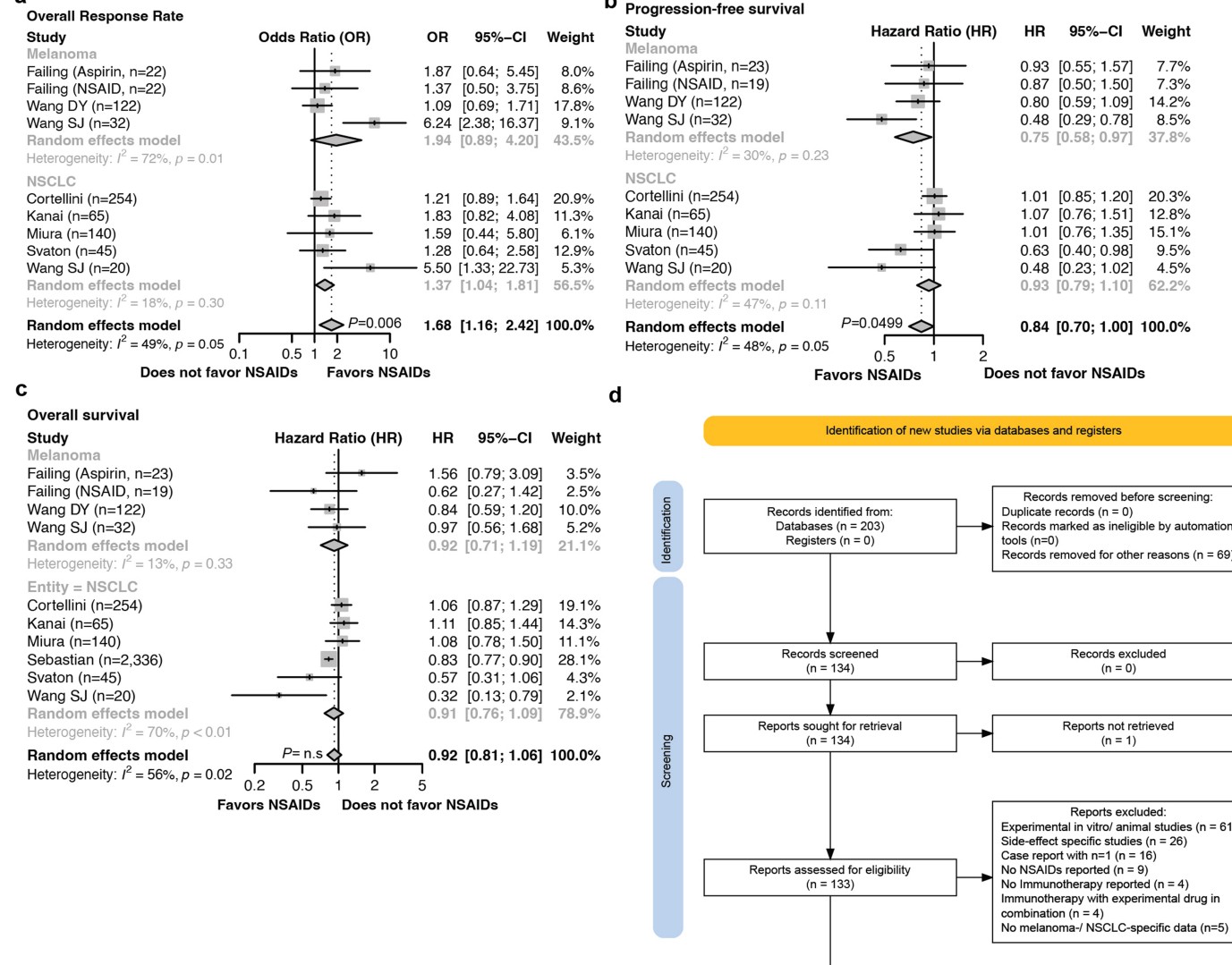

**Extended Data Fig. 9 | Meta-analysis of retrospective clinical studies.**
**a-c**, Forest plot of pooled odds ratios (ORs) and hazard ratios (HRs) with 95% confidence intervals (CI) across included studies of patients receiving ICB with and without concomitant non-steroidal anti-inflammatory drugs (NSAIDs) medication. Number of participants are included in each figure panel. **a**, Odds ratios comparing overall response rates. **b**, Hazard ratios comparing progression-free survival. **c**, Hazard ratios comparing overall survival. $I^2$: I-squared; $P$: probability. **d**, PRISMA flow diagram of literature search and study selection process for meta-analysis. Statistical analysis was performed with a random effects model, data are presented as mean values ± 95% confidence interval. See also Supplementary Tables 5 and 6.

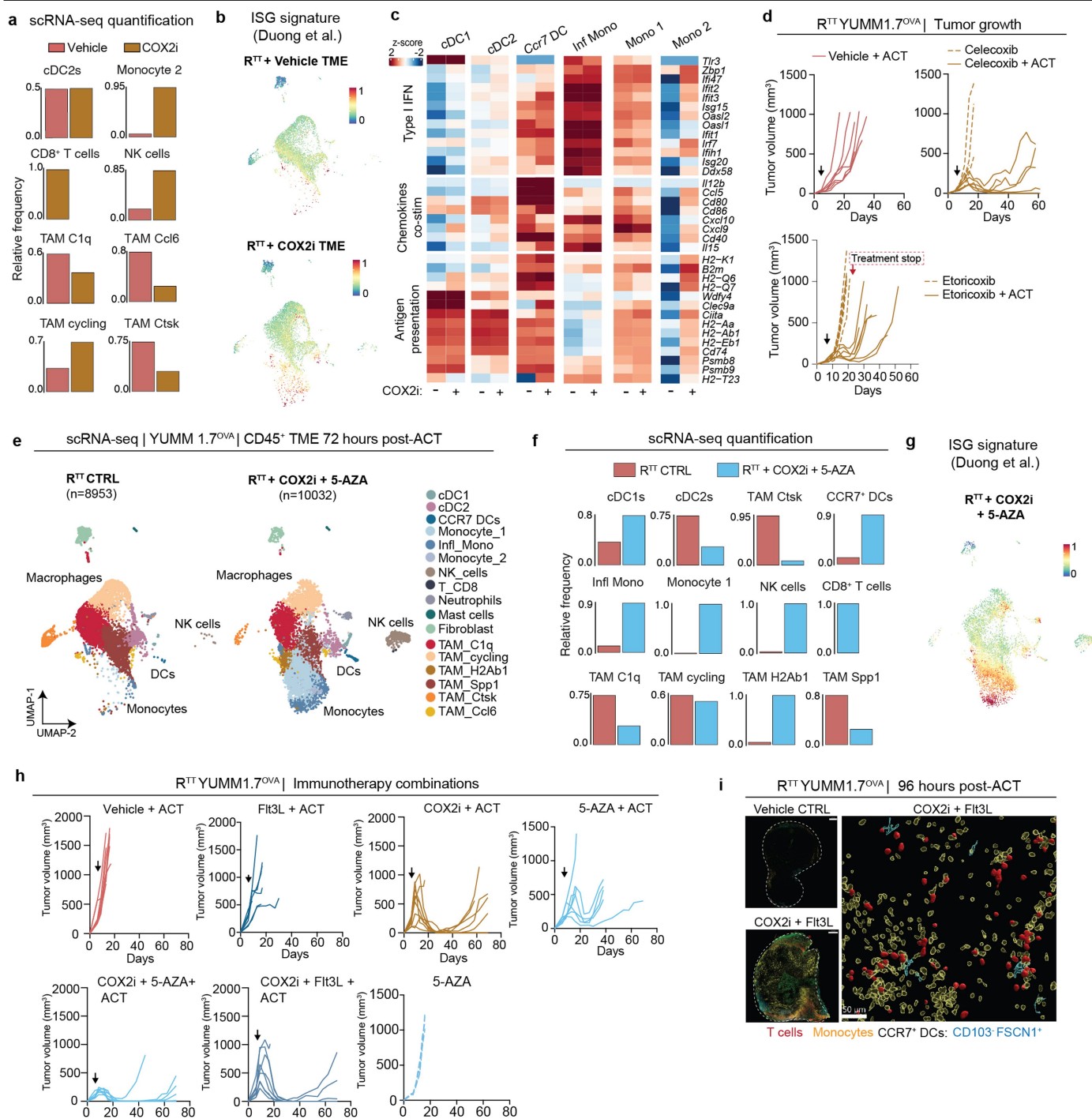

**Extended Data Fig. 10 | Pharmacological inhibition of PGE₂ and type I IFN modulators resensitize cross-resistant tumors to immunotherapy.**
**a**, Relative frequency of each cell type across conditions (scRNA-seq). **b**, Scoring of the *Duong et al.* inflammatory signature[37] in R^TT COX2i-treated tumors. **c**, Heatmap of scaled gene expression (scRNA-seq) in YUMM1.7^OVA R^TT vehicle-treated tumors (CTRL) and R^TT celecoxib (COX2i)-treated tumors (*n* = 3 tumors were pooled/group). **d**, Response of YUMM1.7^OVA R^TT tumors to vehicle + adoptive T cell transfer (ACT) (*n* = 7 mice), celecoxib (*n* = 4 mice), celecoxib + ACT (*n* = 6 mice) and etoricoxib ± ACT (*n* = 7 mice/group). Red arrow and box indicate COX2i treatment stop. **e**, UMAP of scRNA-seq of CD45⁺ cells from YUMM1.7^OVA

R^TT mice treated with vehicle or COX2i + 5-AZA isolated 72 h post-ACT (*n* = 3 tumors pooled/group). **f**, Relative frequency of each cell type across conditions in **e** (scRNA-seq). **g**, Scoring of the *Duong et al.* inflammatory signature[37] in COX2i + 5-AZA treated tumors. **h**, Response of YUMM1.7^OVA R^TT tumors to 5-azacytidine (5-AZA) (*n* = 4 mice), 5-AZA + ACT (*n* = 7 mice), vehicle+ACT (*n* = 8 mice), Flt3L+ACT (*n* = 7 mice), COX2i+ACT (*n* = 9 mice), COX2i + 5-AZA + ACT (*n* = 7 mice) and COX2i+Flt3L+ACT (*n* = 8 mice). **i**, Representative image of IF staining of vehicle-treated and COX2i+Flt3L R^TT YUMM1.7^OVA tumors 96 h post-ACT (*n* = 2 tumors per condition). Scale bar = 1000 μm, zoom-in = 50 μm. Black arrows indicates day of ACT.

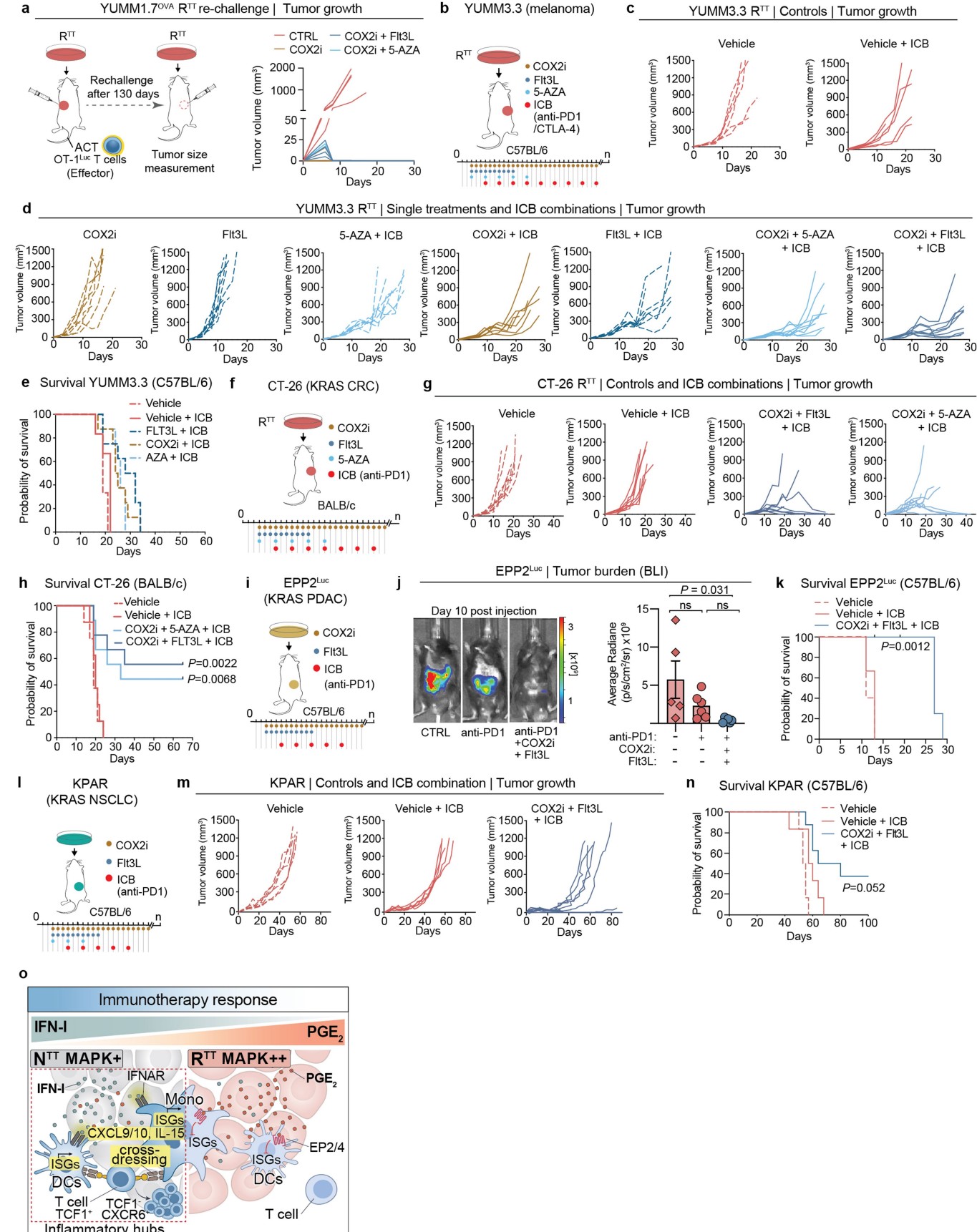

**Extended Data Fig. 11** | See next page for caption.

**Extended Data Fig. 11 | Combination therapies in melanoma, lung, pancreatic and colorectal mouse cancer models. a**, Scheme outlining the re-injection of YUMM1.7$^{OVA}$ R$^{TT}$ cells into tumor-free responder mice from Fig. 5e (left) and tumor growth of re-challenged mice ($n = 4$ mice for COX2i + Ftl3L, $n = 3$ COX2i and $n = 2$ COX2i + 5-AZA) or naive control mice ($n = 5$ mice). **b**, Scheme outlining the injection of YUMM3.3 R$^{TT}$ tumors into C57BL/6 mice and the treatment regimen with COX2i, Flt3L, 5-AZA and anti-PD1/CTLA-4 (ICB) or isotype control. **c**, Response to control vehicle treatment or vehicle + ICB treatment of YUMM3.3 R$^{TT}$ tumors ($n = 6$ mice). **d**, Response to single treatments and ICB combination treatments of YUMM3.3 R$^{TT}$ tumors. COX2i, Flt3L, 5-AZA + ICB, COX2i+ICB and Flt3L+ICB ($n = 8$ mice), COX2i + 5-AZA + ICB and COX2i+Flt3L+ICB ($n = 10$ mice). **e**, Survival plot. **f**, Scheme outlining the injection of CT-26 R$^{TT}$ tumors into BALB/c mice and the treatment regimen with COX2i, Flt3L, 5-AZA and anti-PD1 (ICB) or isotype control. **g**, Response to treatment of CT26 R$^{TT}$ tumors. Vehicle and Vehicle+ICB ($n = 8$ mice), COX2i + 5-AZA + ICB and COX2i+Flt3L+ICB ($n = 9$ mice). **h**, Survival plot. **i**, Scheme outlining the intrapancreatic injection of EPP2$^{Luc}$ cells into C57BL/6 mice and the treatment regimen with COX2i, Flt3L and anti-PD1 (ICB) or isotype control. **j**, Left, representative BLI images of EPP2$^{Luc}$ tumor burden and right, quantification of control ($n = 5$ mice), anti-PD1 ($n = 6$ mice), anti-PD1 + COX2i + Ftl3L ($n = 6$ mice). **k**, Survival plot of **j. l**, Scheme outlining the injection of KPAR tumors into C57BL/6 mice and the treatment regimen with COX2i, Flt3L and anti-PD1 (ICB) or isotype control. **m**, Response to treatment of KPAR tumors, vehicle ± ICB ($n = 6$ mice) and ICB + COX2i + Ftl3L ($n = 8$ mice). **n**, Survival plot of m. **o**, Scheme summarizing the role of inflammatory monocytes in T cell restimulation and the functional convergence of PGE$_2$ and IFN-I to determine an inflammatory TME, T cell restimulation and immunotherapy response. Bar graphs depict the mean ± s.e.m. Statistical analysis was performed with a one-way ANOVA with Tukey's multiple comparisons test in **j**. Log-rank Mantel Cox test in **h, k, n**. ns = not significant.

# Reporting Summary

## Statistics

For all statistical analyses, confirm that the following items are present in the figure legend, table legend, main text, or Methods section.

| n/a | Confirmed | |
|---|---|---|
| ☐ | ☒ | The exact sample size (*n*) for each experimental group/condition, given as a discrete number and unit of measurement |
| ☐ | ☒ | A statement on whether measurements were taken from distinct samples or whether the same sample was measured repeatedly |
| ☐ | ☒ | The statistical test(s) used AND whether they are one- or two-sided<br>*Only common tests should be described solely by name; describe more complex techniques in the Methods section.* |
| ☐ | ☒ | A description of all covariates tested |
| ☐ | ☒ | A description of any assumptions or corrections, such as tests of normality and adjustment for multiple comparisons |
| ☐ | ☒ | A full description of the statistical parameters including central tendency (e.g. means) or other basic estimates (e.g. regression coefficient) AND variation (e.g. standard deviation) or associated estimates of uncertainty (e.g. confidence intervals) |
| ☐ | ☒ | For null hypothesis testing, the test statistic (e.g. *F*, *t*, *r*) with confidence intervals, effect sizes, degrees of freedom and *P* value noted<br>*Give P values as exact values whenever suitable.* |
| ☒ | ☐ | For Bayesian analysis, information on the choice of priors and Markov chain Monte Carlo settings |
| ☒ | ☐ | For hierarchical and complex designs, identification of the appropriate level for tests and full reporting of outcomes |
| ☐ | ☒ | Estimates of effect sizes (e.g. Cohen's *d*, Pearson's *r*), indicating how they were calculated |

*Our web collection on statistics for biologists contains articles on many of the points above.*

## Software and code

Policy information about availability of computer code

| Data collection | Flow cytometry data was acquired using an LSR Fortessa (BD) or BD Aria. Immunofluorescence staining's were imaged on a TCS SP8 confocal microscope (Leica). For tracking bioluminescence signal from transferred T cells an IVIS imager (Perkin Elmer) was used. Single-cell RNA-seq libraries were sequenced on NovaSeq (Illumina). Western blot images were acquired on a ChemiDoc Imaging system (BioRad). qPCR data was obtained using a CFX384 Real-Time-Cycler (BioRad). |
|---|---|
| Data analysis | scRNA-seq analysis TME<br>CD45+ immune cells were collected in 4 different 10x Genomics sequencing experiments: Exp1-Chromium Single Cell 3' scRNA-seq samples pre-processed using cellranger count v6.1.1 (Y3.3 samples: NTT/108155 and RTT/108157); Exp2- 3' CellPlex multiplex experiment with 4 samples pre-processed using cellranger multi v6.1.1 (YUMM1.7OVA samples: NTT_3A6_ACT, RTT_2B12_ACT, RTT_PTGS12KO_ACT, RTT_ROSA26_ACT). Exp3 and Exp4-Chromium Flex multiplex experiments with 4 samples each pre-processed using cellranger multi v7.1.0 and the build in Probe Set v1.0.1 mm10-2020-A. Exp3- YUMM1.7OVA samples: RTT mCherry CTRL, RTT IRF3/7, RTT COX2i and RTT COX2i + 5-AZA; Exp4- YUMM1.7OVA contained biological replicates of Exp2 samples and untreated YUMM1.7OVA samples (noA): NTTnoA/271221, RTTnoA/271222, NTT/271223, RTT/271224). The prebuilt 10X mm10 reference refdata-gex-mm10-2020-A was used. Further processing was performed in R v4.2.2 with Seurat v4.3.0. For generating a CD45+ immune reference map we integrated cells from the first three experiments as follows. The cellranger filtered feature-barcode matrices were used, retaining cells with more than 1000 detected genes and less than 15% of mitochondrial and less than 40% of ribosomal RNA reads. An integrated feature-barcode matrix from the 3 experimental batches was generated accounting for the inclusion of a probe-based assay by keeping genes found in at least 5 cells in each experiment, and excluding ribosomal and mitochondrial genes. Data was log-normalized, scaled (regressing out the difference between the G2M and S phase signature scores), dimensionality reduction was performed using PCA on the top 3000 most variable genes, batch correction across batches was performed using Harmony84 v0.1.1, the 40 harmony embeddings were used for UMAP visualizations. The first 40 harmony dimensions were used to identify immune cell subclusters with a resolution of 0.5, that were further assigned to cell types using known markers and publicly |

available myeloid reference data-sets. Cells were scored for the expression of published signatures using the AddModuleScore function85. Wilcoxon rank sum test implemented in Presto v1.0.0 was used to identify differentially expressed genes. Seurat's reference-based mapping was used to predict celltype identity and map cells of the biological replicate experiment to our annotated reference set using the FindTransferAnchors and MapQuery functions, after a quality control process retaining cells between 1000 and 4500 detected genes for 271222 and 271224, and 1300 and 8000 detected genes for 271221 and 271223, and limiting count tables to the gene universe of the reference. Depth normalized counts for pseudobulk and GSEA functional analyses of this experiment were generated using cellranger aggr. Differences between ACT and untreated conditions (no ACT) from the replicate experiment (Exp4) were explored on a pseudo-bulk level in an unsupervised clustering analysis with heatmap visualization. The fibroblast cluster was removed prior to further processing. Sum aggregation on the depth-normalized UMI counts, was followed by variance stabilizing transformation, selection of 300 most variable genes, standardization, kmeans clustering (k=3) and enrichr analysis against the Reactome_2022 using enrichR.

scRNA-seq analysis intratumoral  CD8+ OT-1 T cells
Single-cell gene-expression of isolated NTT and RTT T cells was assayed in a Chromium Flex experiment, read processing was performed using cellranger multi v7.1.0 using probeset v1.0.1 mm10-2020-A. Cellranger filtered feature-barcode matrices were used on, further filtered to-retain cells with more than 800 detected genes, less than 10 percent of mitochondrial and less than 10 percent of ribosomal RNAs reads and remove cells of contaminant clusters identified using SingleR and ImmGen reference (fibroblasts, MoMac populations). Data was log-normalized, scaled, dimensionality reduction was performed using PCA on the top 2000 most variable genes, harmony was used for the integration of cells from different samples, 15 harmony embeddings were used for UMAP visualizations, and published tumor single cell data for signature scoring. Gene lists are deposited in Supplementary Table 2.

RNA velocity analysis
Loom files containing the splicing annotation were created for each sample using the velocyto run command from the package velocyto (0.17,17), with default parameters, with no masked intervals. Then, the loom files were combined with the scRNA-seq object that had been filtered to keep the data for monocyte and macrophage populations (Monocyte_1, Monocyte_2, Infl_Mono, TAM_CCL6, TAM_Ctsk, TAM_C1q, TAM_H2Ab1, TAM_Spp1, TAM_cycling), for each condition (NTT, RTT, RTT PTGS12KO). First and second order moments were computed using scvelo (0.2.5) pp.moments (n_pcs = 30, n_neighbors = 30) and the dynamical model was run with default parameters. Python version 3.8.12 was used.

SCENIC analysis
Gene regulatory networks for each cell population in each condition were calculated using SCENIC48. The motif database used was mm9-tss-centered-10kb-7species.mc9nr.feather. Co-expression network was calculated using GENIE3. The gene regulatory network was built using SCENIC wrapper functions.

TME signatures in immunotherapy-treated human samples
Gene expression data for patients receiving checkpoint blockade were obtained from Gide et al. (accession no. PRJEB23709). The TME-COX, TME-IRF3/7 and CD8+ T cell scores for each tumor sample were defined as the geometric mean of the expression values of each of the gene sets, respectively (Supplementary Table 4). Univariate Cox proportional hazards models, in which the TME-COX and TME-IRF3/7 scores were included as continuous variables, were used for testing the statistical association between gene signature expression and patient survival, separately for both signatures. The tumor samples were then divided into three groups based on the signature score (bottom third, mid third, top third) and Kaplan-Meier plots were generated for visualization. The association between signature expression and CD8+ T cell abundance was evaluated by calculating the Person's correlation coefficient between the signature score and a CD8+ score for each signature separately. For this all scores were normalized to a median of zero and standard deviation of one. The two overlapping genes were removed from the CD8+ signature before comparing it to TME-IRF3/7 signature expression. For evaluating the enrichment of TME-COX and TME-IRF37 gene signatures in responder and non-responder patients to TIL therapy (baseline) from Barras et al. mouse gene ids were first converted to human orthologs (with DIOPT v9; Best Score = Yes, Best Score Reverse = Yes,  DIOPT Score > 7) and single-cell level signature enrichment scores for the "humanized" gene sets were calculated using AddModuleScore_UCell.

For acquisition of flowcytometry data the FACS DIVA software (v9.0.1) was used. For analysis of flowcytometry data we used Flowjo (version 10.8 or newer).
Statistical analysis was performed with Graphpad Prism (v9.1.2 or newer) or Microsoft Excel  (v 16.88)
Immunofluorescence images were adaptively deconvoluted using the Leica TCS SP8 LIGHTNING tool  v3.5.7.23225 and further analyzed using Imaris (v9.9).
Metabolites from the untargeted metabolomics approach were annotated using the compound discoverer software (v3.0).
Bioluminescence signal to track T cell infiltration was analyzed using the Living Image software (v4.4)

For manuscripts utilizing custom algorithms or software that are central to the research but not yet described in published literature, software must be made available to editors and reviewers. We strongly encourage code deposition in a community repository (e.g. GitHub). See the Nature Portfolio guidelines for submitting code & software for further information.

# Data

Policy information about availability of data

All manuscripts must include a data availability statement. This statement should provide the following information, where applicable:

- Accession codes, unique identifiers, or web links for publicly available datasets
- A description of any restrictions on data availability
- For clinical datasets or third party data, please ensure that the statement adheres to our policy

Gene expression data from YUMM1.7 NTT and RTT cancer cells sorted from tumors was previously deposited under the Gene Expression Omnibus accession number GSE132443. Specifically samples from NTT tumors (GSM3864154,GSM3864155, GSM3864157) and RTT tumors (GSM3864170, GSM3864172,GSM3864173) were used to generate plots in Fig. 3a and Extended Data Fig. 5i. Raw and processed files of the single-cell RNA sequencing generated in this study have been deposited under the GEO accession number GSE241750. Expression data from the Gide et al. study is publicly available (accession no. PRJEB23709). Single cell RNA data from the Barras et al. study, including the Seurat object for the myeloid compartment was obtained upon direct request. Expression data from the Cheng et al. study was publicly available (GSE154763) and the raw count matrix was obtained upon request. Source data is provided for all graphs that are shown in the manuscript. All other formats of raw data are available upon request.

# Research involving human participants, their data, or biological material

Policy information about studies with [human participants or human data](). See also policy information about [sex, gender (identity/presentation), and sexual orientation]() and [race, ethnicity and racism]().

| | |
|---|---|
| Reporting on sex and gender | Biopsies were obtained from patients with an age at diagnosis that ranged from 24 to 85 with a median of 66. 34% were female patients and 66% were male patients. |
| Reporting on race, ethnicity, or other socially relevant groupings | *Please specify the socially constructed or socially relevant categorization variable(s) used in your manuscript and explain why they were used. Please note that such variables should not be used as proxies for other socially constructed/relevant variables (for example, race or ethnicity should not be used as a proxy for socioeconomic status).*<br>*Provide clear definitions of the relevant terms used, how they were provided (by the participants/respondents, the researchers, or third parties), and the method(s) used to classify people into the different categories (e.g. self-report, census or administrative data, social media data, etc.)*<br>*Please provide details about how you controlled for confounding variables in your analyses.* |
| Population characteristics | We obtained cell-segmented data for 74 FOVs (Field of View, an area of 500x500 µm) from TMA cores of 34 melanoma metastases, in total consisting of 980 genes x 171,536 cells. Tumor tissues were obtained from 21 lymph nodes, 7 subcutaneous metastases, 1 lung metastasis and 1 brain metastasis (4 NAs), from 31 patients containing 72 FOVs. 2 FOVs were from tonsils as control. Most tumor tissue were from patients that were treatment naïve at the time of surgery. Tissue collection was approved by the Regional Ethics committee at Lund University (Dnr. 191/2007 and 101/2013). Patients signed an informed consent. The majority of TMA cores contained tertiary lymphoid structures (TLS) and FOVs were preferentially directed to these regions. Low-quality FOVs, cells with < 20 counts and potential multiplets of cells (area exceeding the sample geometric mean + 5 standard deviation) were discarded. |
| Recruitment | *Describe how participants were recruited. Outline any potential self-selection bias or other biases that may be present and how these are likely to impact results.* |
| Ethics oversight | Tissue collection was approved by the Regional Ethics committee at Lund University (Dnr. 191/2007 and 101/2013). Patients signed an informed consent. |

Note that full information on the approval of the study protocol must also be provided in the manuscript.

# Field-specific reporting

Please select the one below that is the best fit for your research. If you are not sure, read the appropriate sections before making your selection.

☒ Life sciences ☐ Behavioural & social sciences ☐ Ecological, evolutionary & environmental sciences

For a reference copy of the document with all sections, see [nature.com/documents/nr-reporting-summary-flat.pdf]()

# Life sciences study design

All studies must disclose on these points even when the disclosure is negative.

| | |
|---|---|
| Sample size | The effect size was determined from prior research or pilot experiments relevant to our study, especially from Haas et al., 2021. This estimate informed our power analysis and allowed us to choose a sample size that would balance statistical power with practical considerations. The statistical tests planned for the analysis were considered when determining sample size. Practical considerations such as time, cost, and resource availability were also considered. Ethical considerations (3R) played a role in determining sample sizes for in vivo experiments. We aimed to use the smallest number of subjects necessary in in vivo studies to achieve reliable and valid results. |
| Data exclusions | No data points were excluded, except for when tumors became necrotic as this is an end-point in our current working animal license or when mice displayed weight loss and sickness and had to be euthanized. |
| Replication | All in vitro and in vivo experiments were repeated at least twice and always with multiple replicates, except for the following experiments which were performed only once: scRNA-seq of experiments involving pharmacological treatment of the YUMM1.7 RTT model, intratumoral injection of T cells and the YUMM3.3 model. IF stainings for which representative images are shown were repeated at least twice, except for the NTT in Batf3-/- and Nur77 reporter experiment which was performed once but with n=3 tumors and was also confirmed with flow cytometry. Pharmacological combination treatments of the KPAR model were performed once. |
| Randomization | Mice were allocated to treatment or control group based on the average tumor size to achieve similar tumor burden between groups at the time of treatment start. For experiments where drug treatment had to be started at day 3 post injection, mice were randomly allocated to treatment groups. Within an experiment age and sex-matched mice were used. For all other experiments samples were randomly allocated. |
| Blinding | Scientists were not blinded to genotypes or treated groups prior to data collection and during analysis in order to prevent mislabeling of genotypes and treatment groups. Experiments were repeated by different independent investigators. |

# Reporting for specific materials, systems and methods

We require information from authors about some types of materials, experimental systems and methods used in many studies. Here, indicate whether each material, system or method listed is relevant to your study. If you are not sure if a list item applies to your research, read the appropriate section before selecting a response.

## Materials & experimental systems

| n/a | Involved in the study |
|-----|----------------------|
| ☐ | ☒ Antibodies |
| ☐ | ☒ Eukaryotic cell lines |
| ☒ | ☐ Palaeontology and archaeology |
| ☐ | ☒ Animals and other organisms |
| ☒ | ☐ Clinical data |
| ☒ | ☐ Dual use research of concern |
| ☒ | ☐ Plants |

## Methods

| n/a | Involved in the study |
|-----|----------------------|
| ☒ | ☐ ChIP-seq |
| ☐ | ☒ Flow cytometry |
| ☒ | ☐ MRI-based neuroimaging |

## Antibodies

Antibodies used

Antibodies for flowcytometry
anti-mouse AXL PE-Cy7 eBioscience Cat#25-1084-82, clone: MAXL8DS 1:200
anti-mouse CD103 PerCP/Cyanine5.5 BioLegend Cat#121415 , clone: 2E7 1:100
anti-mouse CD103 PE BioLegend Cat#121405 , clone: 2E7 1:100
anti-mouse CD11b APC eBioscience Cat#17-0112-81, clone: M1/70 1:200
anti-mouse CD11b PerCP/Cyanine5.5 BioLegend Cat#101229, clone: M1/70 1:200
anti-mouse CD11c BV605 BD Pharmingen Cat#563057, clone: HL3 1:100
anti-mouse CD11c FITC BioLegend Cat#117305, clone: N418 1:100
anti-mouse CD16/CD32 (Mouse BD Fc Block) BD Pharmigen Cat#553141, clone: 2.4G2 1:50
anti-mouse CD24 BV510 BioLegend Cat#101831, clone: M1/69 1:100
anti-mouse CD24 FITC eBioscience Cat#11-0242-82, clone: M1/69 1:100
anti- mouse CD279/PD-1 BV785 BioLegend Cat#135225, clone: 29F.1A12 1:200
anti-mouse CD279/PD-1 FITC  BioLegend Cat#135213, clone: 29F.1A12 1:200
anti-mouse CD40 APC BioLegend Cat#124611, clone: 3/23 1:200
anti-mouse CD45 BV711 BioLegend Cat#103147, clone: 30-F11 1:500
anti-mouse CD45 FITC BioLegend Cat#103107, clone:30-F11 1:500
anti-mouse CD86 BV510 BioLegend Cat#105039, clone: GL-1 1:100
anti-mouse CD3 BV605  BD Horizon  Cat#564009, clone:17A2 1:100
anti- mouse CD3 AF647 BioLegend Cat#100209, clone:17A2 1:100
anti-mouse CD3 AF488 BioLegend Cat#100212, clone:17A2 1:100
anti-mouse CD8a eFluor 450 eBioscience Cat#48-0081-80, clone:53-6.7 1:100
anti-mouse CD8a AF594 BioLegend Cat#100758, clone:53-6.7 1:100
anti-mouse Ly-6C BV785 BioLegend Cat#128041, clone:HK1.4 1:100
anti-mouse MHCI (H-2Kb) APC eBioscience Cat#17-5958-82,clone:AF6-88.5.5.3 1:200
anti-mouse MHCI (H2Kb) PE eBioscience Cat#12-5958-80, clone:AF6-88.5.5.3 1:200
anti-mouse MHCII (I-A/I-E) eFluor450 eBioscience Cat#48-5321-80, clone:M5/114.15.2 1:200
anti-mouse I-A/I-E APC BioLegend Cat#107613, clone:M5/114.15.2 1:200
anti-mouse NK-1.1 BV711 BioLegend Cat#108745, clone: PK136 1:100
anti-mouse TCF1 PE  BD Pharmingen Cat#564217, clone: S33-966 1:50
anti-mouse TIM3 BV711 BioLegend Cat#119727, clone: RMT3-23 1:100
anti-mouse CD88 PE BioLegend Cat#135805, clone: 20/70 1:100
anti-mouse Ly-6A/E (Sca-1) FITC BioLegend Cat#108105, clone: D7 1:100
anti-mouse SIINFEKL-HK2B PE Invitrogen Cat#12-5743-81, clone: 25-D1.16 1:100
anti-mouse F4/80 PE BioLegend Cat#B123110, clone: BM8 1:200
Rat IgG1, K Isotype control PE BD Pharmingen Cat#5546, clone: R3-34

Antibodies for T cell activation
CD3e Monoclonal Antibody, functional grade eBioscience Cat#16-0031-81, clone: 145-2C11
CD28 Monoclonal Antibody, functional grade eBioscience Cat#16-0281-81, clone: 36.51

Antibodies for in vivo treatment
InVivoMAb anti-mouse PD-1 (CD279) BioXCell Cat#BE0146, clone:RMP1-14
InVivoMAb anti-mouse CTLA-4 (CD152) BioXCell Cat#BE0164, clone:9D9
InVivoMAb anti-mouse NK1.1 BioXCell Cat#BE0036, clone:PK136
Monoclonal antibody anti-mouse CD8 in house produced clone: 2.43
InVivoMAb anti-mouse IFNAR-1  BioXCell Cat#BE0241, clone: MAR1-5A3
InVivoMAb anti-mouse IFNg BioXCell Cat#BE0055, clone: XMG1.2
InVivoMAb recombinant Flt-3L-Ig BioXCell Cat#BE0098, clone: hum/hum
InVivoMAb rat IgG2b isotype control  BioXCell Cat#BE0090, clone: LTF2
InVivoMAb rat IgG2a isotype control  BioXCell Cat#BE0089, clone: 2A3
InVivoMAb mouse IgG1 isotype control BioXCell Cat#BE0083, clone: MOPC-21

Antibodies for western blot
anti-mouse Vinculin mAb Sigma Aldrich  Cat#V9131, hVIN-1

anti-mouse COX2 mAb Cell Signaling Technology  Cat# 12282S, D5H5
anti-rabbit IgG HRP-linked  Cell Signaling Technology  Cat#7074
anti-mouse IgG HRP-linked  Cell Signaling Technology  Cat#7076

Antibodies for in vitro treatment
InVivoMAb anti-mouse IFNAR-1  BioXCell Cat#BE0241, clone: MAR1-5A3
InVivoMAb mouse IgG1 isotype control BioXCell Cat#BE0083, clone: MOPC-21

Antibodies for IF staining
anti-mouse CD3 BV421 Biolegend  Cat#100227, clone 17A2
anti-mouse MHCII (I-A/I-E) BV510 Biolegend  Cat#107635, clone M5/114.15.2
anti-mouse CD103 unconjugated  R&D Systems Cat#AF1990, polyclonal
anti-mouse FSCN1 AF594 SCBT Cat#sc-21743, clone 55-k2
anti-mouse LY6C AF647 Biolegend  Cat#128010, clone HK1.4

Validation

Antibodies for flowcytometry
https://www.thermofisher.com/antibody/product/Axl-Antibody-clone-MAXL8DS-Monoclonal/25-1084-82
https://www.biolegend.com/en-us/products/percp-cyanine5-5-anti-mouse-cd103-antibody-5599
https://www.biolegend.com/en-us/products/pe-anti-mouse-cd103-antibody-3574
https://www.thermofisher.com/antibody/product/CD11b-Antibody-clone-M1-70-Monoclonal/17-0112-82
https://www.biolegend.com/en-us/products/percp-anti-mouse-human-cd11b-antibody-4315
https://www.bdbiosciences.com/en-at/products/reagents/flow-cytometry-reagents/research-reagents/single-color-antibodies-ruo/bv605-hamster-anti-mouse-cd11c.563057
https://www.biolegend.com/en-us/search-results/fitc-anti-mouse-cd11c-antibody-1815?GroupID=BLG11937&gclid=Cj0KCQjwi7GnBhDXARIsAFLvH4mVVU5ae5X-Cx6DF9ti4yolV5-O__oZQAqSb2y-Dr5SkTcc4O_-saAaAgb2EALw_wcB
https://www.bdbiosciences.com/en-at/products/reagents/flow-cytometry-reagents/research-reagents/single-color-antibodies-ruo/purified-rat-anti-mouse-cd16-cd32-mouse-bd-fc-block.553141
https://www.biolegend.com/de-at/products/brilliant-violet-510-anti-mouse-cd24-antibody-9925
https://www.thermofisher.com/antibody/product/CD24-Antibody-clone-M1-69-Monoclonal/11-0242-82
https://www.biolegend.com/en-us/products/brilliant-violet-785-anti-mouse-cd279-pd-1-antibody-9874
https://www.biolegend.com/en-us/products/fitc-anti-mouse-cd279-pd-1-antibody-7004
https://www.biolegend.com/de-at/products/apc-anti-mouse-cd40-antibody-4984
https://www.biolegend.com/de-at/products/brilliant-violet-711-anti-mouse-cd45-antibody-10439
https://www.biolegend.com/de-at/products/fitc-anti-mouse-cd45-antibody-99
https://www.biolegend.com/de-at/products/brilliant-violet-510-anti-mouse-cd86-antibody-8745
https://www.biolegend.com/en-gb/products/brilliant-violet-605-anti-mouse-cd3-antibody-8503
https://www.biolegend.com/en-gb/products/alexa-fluor-647-anti-mouse-cd3-antibody-2693
https://www.biolegend.com/en-gb/products/alexa-fluor-488-anti-mouse-cd3-antibody-2835
https://www.thermofisher.com/antibody/product/CD8a-Antibody-clone-53-6-7-Monoclonal/48-0081-82
https://www.biolegend.com/en-gb/products/alexa-fluor-594-anti-mouse-cd8a-antibody-9608
https://www.biolegend.com/en-gb/products/brilliant-violet-785-anti-mouse-ly-6c-antibody-11982
https://www.thermofisher.com/antibody/product/MHC-Class-I-H-2Kb-Antibody-clone-AF6-88-5-5-3-Monoclonal/17-5958-82
https://www.thermofisher.com/antibody/product/MHC-Class-I-H-2Kb-Antibody-clone-AF6-88-5-5-3-Monoclonal/12-5958-82
https://www.thermofisher.com/antibody/product/MHC-Class-II-I-A-I-E-Antibody-clone-M5-114-15-2-Monoclonal/48-5321-82
https://www.biolegend.com/en-gb/products/apc-anti-mouse-i-a-i-e-antibody-2488
https://www.biolegend.com/en-gb/products/brilliant-violet-711-anti-mouse-nk-1-1-antibody-9576
https://www.bdbiosciences.com/en-at/products/reagents/flow-cytometry-reagents/research-reagents/single-color-antibodies-ruo/pe-mouse-anti-tcf-7-tcf-1.564217
https://www.biolegend.com/en-gb/products/brilliant-violet-711-anti-mouse-cd366-tim-3-antibody-14918
https://www.biolegend.com/en-gb/products/pe-anti-mouse-cd88-c5ar-antibody-6240
https://www.biolegend.com/en-gb/products/fitc-anti-mouse-ly-6a-e-sca-1-antibody-227
https://www.thermofisher.com/antibody/product/OVA257-264-SIINFEKL-peptide-bound-to-H-2Kb-Antibody-clone-eBio25-D1-16-25-D1-16-Monoclonal/12-5743-81
https://www.biozym.com/8667/pe-anti-mouse-f4/80
https://www.bdbiosciences.com/en-at/products/reagents/flow-cytometry-reagents/research-reagents/flow-cytometry-controls-and-lysates/pe-rat-igg1-isotype-control.554685

Antibodies for T cell activation
https://www.thermofisher.com/antibody/product/CD3e-Antibody-clone-145-2C11-Monoclonal/16-0031-82
https://www.thermofisher.com/antibody/product/CD28-Antibody-clone-37-51-Monoclonal/16-0281-82

Antibodies for in vivo treatment
https://bioxcell.com/invivoplus-anti-mouse-pd-1-cd279-bp0146?psafe_param=1&gad=1&gclid=CjwKCAjwrranBhAEEiwAzbhNtaF7E-IyhEPPbzG9YrqPKZ1Erf9_b5hU3drWDF0xA7iwNPdAPrGy1BoCYjAQAvD_BwE
https://bioxcell.com/invivoplus-anti-mouse-ctla-4-cd152-bp0164?gad=1&gclid=CjwKCAjwrranBhAEEiwAzbhNtfZq0JcYygOuO_4HYJRrZQadiKd5ehKt7US_gjQZQ11eC6UR9rhLShoCglkQAvD_BwE
https://bioxcell.com/invivomab-anti-mouse-nk1-1-be0036
https://bioxcell.com/invivomab-anti-mouse-ifnar-1-be0241?psafe_param=1&gad=1&gclid=CjwKCAjwrranBhAEEiwAzbhNtblHOyWh77kCe1_BIoqolfJmLr7FY7mJLFukpKcCXKSxvWJSqpxtLxoC_ZMQAvD_BwE
https://bioxcell.com/invivomab-anti-mouse-ifn-gamma-be0055#tab_pdetails
https://bioxcell.com/invivomab-recombinant-flt-3l-ig-hum-hum
https://shop.bio-connect.nl/antibodies/invivomab-rat-igg2b-isotype-control/be0090_25mg/sfid/5189041

https://www.biozol.de/de/product/bxc-be0089-100mg
https://bioxcell.com/invivomab-mouse-igg1-isotype-control-unknown-specificity-be0083?
gad=1&gclid=CjwKCAjwrranBhAEEiwAzbhNtY8dTzxEJQ6To_BBBEXR2fCjoEhd3keXbgLEDPy5Tzxj_EIIQUTgKRoCYYAQAvD_BwE

Antibodies for western blot
https://www.sigmaaldrich.com/AT/de/product/sigma/v9131
https://www.cellsignal.com/products/primary-antibodies/cox2-d5h5-xp-rabbit-mab/12282?_requestid=140729
https://www.cellsignal.com/products/secondary-antibodies/anti-rabbit-igg-hrp-linked-antibody/7074?_requestid=699267
https://www.cellsignal.com/products/secondary-antibodies/anti-mouse-igg-hrp-linked-antibody/7076

Antibodies for in vitro treatment
https://bioxcell.com/invivomab-anti-mouse-ifnar-1-be0241?
psafe_param=1&gad=1&gclid=CjwKCAjwrranBhAEEiwAzbhNtblHOyWh77kCe1_BIoqolfJmLr7FY7mJLFukpKcCXKSxvWJSqpxtLxoC_ZM
QAvD_BwE
https://bioxcell.com/invivomab-mouse-igg1-isotype-control-unknown-specificity-be0083?
gad=1&gclid=CjwKCAjwrranBhAEEiwAzbhNtY8dTzxEJQ6To_BBBEXR2fCjoEhd3keXbgLEDPy5Tzxj_EIIQUTgKRoCYYAQAvD_BwE

Antibodies for IF staining
https://www.biolegend.com/en-us/products/brilliant-violet-421-anti-mouse-cd3-antibody-7326
https://www.biolegend.com/en-us/products/brilliant-violet-510-anti-mouse-i-a-i-e-antibody-7997
https://www.rndsystems.com/products/mouse-integrin-alphae-cd103-antibody_af1990
https://www.scbt.com/p/fascin-1-antibody-55k-2
https://www.biolegend.com/en-us/products/alexa-fluor-647-anti-mouse-ly-6c-antibody-4897

# Eukaryotic cell lines

Policy information about cell lines and Sex and Gender in Research

| Cell line source(s) | Original YUMM1.7 and YUMM3.3 cell-lines were obtained from the Bosenberg Laboratory. All derivatives of these cell-lines were previously generated and described in Haas et al., 2021, Nature Cancer.<br>A375, M249 and LOX cell-lines and RTT derivatives were obtained from the Massague Laboratory/MSKCC. All derivatives of these cell-lines were previously generated and described in Obenauf et al., 2015, Nature.<br>NCI-H358 cells were obtained from ATCC and RTT derivatives were generated in house.<br>CT-26 were purchased from ATCC. All derivatives were previously generated and described in Haas et al., 2021, Nature Cancer.<br>EPP2 were obtained from the Zuber Laboratory/IMP.<br>KPAR cell-line was obtained from the Downward Laboratory/CRICK Institute.<br>HEK-293T cells were purchased from Takara (Lenti-X 293T, 632180).<br>BLaER-1 cell-line was obtained from the Gaidt Laboratory/IMP<br>MONO-MAC-1 cell-line was obtained from the Zuber Laboratory/IMP |
|---|---|
| Authentication | STR Profiling was performed in-house for the YUMM 1.7, YUMM 3.3, EPP2 and KPAR cell lines. Moreover sensitivity to MAPK inhibitors was confirmed for A375, M249 and LOX (BRAFi), CT-26 (MEKi) and for H358 (KRASi). |
| Mycoplasma contamination | Cells were routinely tested for mycoplasma contamination using our in-house PCR-system. Cells tested negative for mycoplasma contamination. |
| Commonly misidentified lines (See ICLAC register) | No commonly misidentified cell lines were used in this study. |

# Animals and other research organisms

Policy information about studies involving animals; ARRIVE guidelines recommended for reporting animal research, and Sex and Gender in Research

| Laboratory animals | All mice were bred and housed in pathogen-free conditions with a housing temperature of 22±1°C, 55 ± 5% humidity, and a photoperiod of 14 hours light and 10 hours dark. Within each experiment, age and sex-matched groups were used. B6.129S(C)-Batf3tm1Kmm/J (Batf3-/-) mice, B6(Cg)-Zbtb46tm1(HBEGF)Mnz/J (zDC-DTR) mice, B6.Cg-Tg(Itgax-cre)1-1Reiz/J (CD11c-Cre) mice and NOD.Cg-Prkdcscid Il2rgtm1Wjl/SzJ (NSG) mice, mice were purchased from Jackson Laboratories. B6.Cg-Rag2tm1.1Cgn/J Ly5.2 (Rag2-/-) mice, BALB/c and C57BL/6J were obtained from the Vienna Biocenter in-house breeding facility.  ItgaxCrePtger2−/−Ptger4fl/fl mice were kindly provided by Dr. Jan Boettcher (TUM, Munich). For Rag2-/- Batf3-/- strain generation, Batf3-/- were crossed to Rag2-/- mice and homozygous offspring (Rag2-/- x Batf3-/-) were confirmed by genotyping and used in subsequent experiments to evaluate the lack of cDC1s in the context of adoptive T cell transfer. For Rag2-/-  zDC-DTR strain generation, zDC-DTR mice were crossed to Rag2-/- mice and homozygous offspring were confirmed by genotyping and used in subsequent experiments to evaluate the effects of DC depletion. For adoptive T cell transfer experiments and injection of YUMM1.7OVA cell-lines,  Rag2-/- mice were used. For the injection of YUMM3.3, KPAR and EPP2 cell-lines C57BL/6 mice were used. For the injection of the CT-26 cell line BALB/c mice were used. For the generation of bone marrow derived dendritic cells (BM-DCs) and Ly6C+ monocytes, bones (femurs and tibias) were collected from in-house-bred C57BL/6 mice. For all strains listed above,  mice were used between 6-12 weeks old. For OT-1Luc CD8+ T cell isolations, 6-24 week-old OT-1Luc Thy1.1 mice were used. |
|---|---|
| Wild animals | Study did not involve wild animals. |

| | |
|---|---|
| Reporting on sex | Within each experiment, sex-matched mice were used whenever possible. Phenotypes and results were confirmed both in male and female mice |
| Field-collected samples | Study did not involved field-collected samples. |
| Ethics oversight | All mouse experiments were performed according to our license approved by the Austrian Ministry (GZ: MA58-2260492-2022-22, GZ: 340118/2017/25, BMBWF-66.015/0009-V/3b/2019, GZ: 801161/2018/17 and GZ: 2021-0.524.218 and their amendments. |

Note that full information on the approval of the study protocol must also be provided in the manuscript.

# Flow Cytometry

## Plots

Confirm that:

☒ The axis labels state the marker and fluorochrome used (e.g. CD4-FITC).

☒ The axis scales are clearly visible. Include numbers along axes only for bottom left plot of group (a 'group' is an analysis of identical markers).

☒ All plots are contour plots with outliers or pseudocolor plots.

☒ A numerical value for number of cells or percentage (with statistics) is provided.

## Methodology

| | |
|---|---|
| Sample preparation | For flow cytometry-based characterization of the TME, tumors were isolated between day 7-11 post-injection, cut into pieces, and digested for 1.5 h at 37°C with collagenase A (1 mg/ml, Roche) and DNAse (20μg/ml, Worthington) in unsupplemented RPMI-1640 media. Digested tumors were strained through a 70μm filter and resuspended in FACS buffer (0.5% BSA, 2mM EDTA). FC-block was performed with anti-CD16/32 (clone 2.4G2, BD Pharmingen) for 10 min at 4°C to avoid nonspecific antibody binding, and staining for cell-surface markers was performed for 30 min at 4°C. For intracellular staining the Foxp3 Transcription factor staining kit was used (eBioscience). For flow cytometry of cultured cancer cells, cells were detached using 0.05% Trypsin and inhibited in full media. For bone marrow derived dendritic cells supernatants was collected. |
| Instrument | LSR Fortessa (BD)<br>FACS Aria (BD) |
| Software | FACS Diva (version 9.0.1), FlowJo (version 10.8.0) |
| Cell population abundance | For sorting CD45+ immune cells, cells were reanalyzed after sorting and a purity of >90% was confirmed. |
| Gating strategy | For all gating strategies, singlets were identified based on FSC-A/FSC-H. Live dead exclusion was performed by staining with the fixable viability dye eF780 (1:1000, eBioscience). DCs were defined in most experiments as MHCII+ CD11c+ CD24+ out of alive CD45+ cells. cDC1s were identified as CD103+ CD11b- out of the total DCs, cDC2s as CD103- CD11b+ and inflammatory cDC2 as CD103- CD11b+ AXL+. AXL was previously described to identify inflammatory cDC2s. Monocytes were defined as Ly6C+ CD11b+ F4/80-, and inflammatory monocytes were identified as monocytes that were Ly6A+. Ly6A was previously described to identify monocytes expressing high levels of interferon-stimulated genes (ISGs). Macrophages were defined as Ly6C- F4/80+ Cd11b+.For the identification of adoptively transferred CD8+ T cells, CD3+ and CD8+ was used. On this populations we assessed PD-1, TIM-3 and TCF1 levels |

☒ Tick this box to confirm that a figure exemplifying the gating strategy is provided in the Supplementary Information.

