## [Peer Review File · Nature]

Cancer cells impair monocyte-mediated T cell stimulation to evade immunity

Corresponding Author: Dr Anna Obenauf

Parts of this Peer Review File have been redacted as indicated to maintain patient confidentiality.

Version 1:

Reviewer comments:

Referee #1

(Remarks to the Author)

This study applies somewhat limited patient data and mainly mouse models to test how the TME influences myeloid reprogramming, focusing on the PGE2/Type I IFN axis. The authors show loss of tumor intrinsic PGE2 and Type I IFN IRF3/7; and that restoration of both axes can restore DC function. They also demonstrate PGE2 impairs acquisition of a type I induced inflammatory state in DC, and that MAPK mediates this suppressive axis in the tumor. Given extensive prior work in PGE2 and cDC1 suppression in the field (see citations below) some of what is seen is somewhat descriptive and may be seen as an extension of the prior work, unless further mechanistic interrogation of the MAPK mediated suppressive axis, the impact of these pathways on DC differentiation, and regulation governance is documented. Despite this the work is superbly executed, controlled and the experiments are very thoughtfully and rigorously designed. The writing also reflects an outstanding grasp of the field but perhaps fails to highlight the truly innovative or surprising findings, which be corrected easily. While Figure 4 attempts to meet this gap, an improved study may map how restoration of these pathways impacts DC tumor antigen trafficking, or local T cell recall by DC in vivo. Direct ex vivo tumor antigen presentation should be simple, or peptide MHC bound on the cell surface, DC T cell interactions in these hubs etc could also be shown.

Eg. Prior work by Böttcher and Bayerl (<https://pubmed.ncbi.nlm.nih.gov/37315536/>) showed PGE2 through EP2/4 and IRF8. Böttcher, J. P. et al. NK cells stimulate recruitment of cDC1 into the tumor microenvironment promoting cancer immune control. *Cell* 172, 1022–1028.e14 (2018). (PGE2 mechanism); Zeleny et al 2015 *Cell*: Reis Y Sousa lab <https://pubmed.ncbi.nlm.nih.gov/26343581/>

The Reis Y Sousa work already showed loss of Cox2 in BrafV600E melanoma models conferred immune sensitivity; so the main difference here is the use of MEKi/Rafi resistant line specific effect and the Type I pathway. This paper should be cited earlier on, in framing the question. In addition the type I effect is also not entirely new in DC (e.g. Fuertes MB, Kacha AK, Kline J, Woo SR, Kranz DM, et al. 2011. Host type I IFN signals are required for antitumor CD8+ T cell responses through CD8α+ dendritic cells. *J. Exp. Med* 208:2005–16). Therefore the most important new feature of this very well executed study is the hyperactivated MAPK pathway as a source for the regulation of COX2 levels and IFN-I signaling. Though again this is also not entirely new as the authors also note (ref 36/39). Therefore some time should be spent delineating the novelty of these findings above others in the field. The shared regulation by MAPK is important as is the observation PGE2 restoration is insufficient to restore ISGs.

- The most obvious interpretation of Fig1c is the TME of RTT is conferring maturation of monocytes to TAMs
- Multiparameter IF is a particularly nice aspect of this manuscript with respect to quantifying and phenotyping multicellular clusters.
- Single cell and imaging: annotation CCR7+ DC is appropriate for the high mig/nat phenotype
- Experiments injecting the T cells directly into tumor and examining differentiation are particularly helpful and thoughtful Fig11, preactivation of OT1 and FTY720 to show local tumor specific differentiation cues etc
- Authors show restoration of T cell infiltration after IRF3/7 expression in RTT tumors, with this is ICB responsiveness also restored? T cell differentiation?
- The pharmacological testing in Fig 5 is a strength and the authors should relate this to use of Cox2 inhibitors in patients and ICB responsivity or decreased MAPK resistance- are there any other studies than ref 51/52 on this?
- Could the authors kindly improve the manuscript beyond the descriptive aspects of Fig4 to better document in vivo or ex vivo functional differences and differentiation trajectories?
- Flt3L alone controls are needed e.g. extended data 7.

• Please delineate ISGs as Type I unique and demonstrate no change in type II responses given common Stat1 /IRF 1 activity. Many shown are Type II targets as well. Or control experiments with anti IFN γ R1 blocking antibodies in at least a couple instances.

Other:

Omission of citation to Edelson and Murphy first showing BATF3 cDC1 confer antitumoral immunity. Omission of citation to Nirschl et al which first showed identification of DC migration/maturation program in melanoma (227hi) when referencing what has now been coopted as DC3/mreg etc and review Pittet JEM as well reconciling these. Key review to cite Sancho lab Mat Rev immuno Wculek et al 2020 also not cited.

Overall this is a beautifully executed study which may not quite meet the level of conceptual impact needed in its current form.

Referee #2

(Remarks to the Author)

In the manuscript by Elewaut et al entitled "Counter-regulated PGE2 and IFN-I create an immune evasive tumor microenvironment", the authors use well-established melanoma models of immunotherapy resistance to investigate the mechanisms involved in the establishment of an immune-evasive tumor microenvironment (TME) and propose therapeutic strategies to reverse these mechanisms. The authors demonstrated a loss of monocytes and cDC populations and an increase in immunosuppressive TAM in IT-resistant tumors. It was further shown that in IT-sensitive tumors, T cells interact with monocyte and cDC1 populations and that the TME supported T cell expansion. When the T cell egress from lymph nodes was blocked, T cell recruitment and expansion was still maintained, suggesting that the TME drives this process. To determine the mechanisms by which TME dictates T cell anti-tumor activity, the authors conducted RNAseq and demonstrated an upregulation of the PGE2 and a downregulation of IFN-I in IT-resistant tumors. Blocking PGE2 and inducing IFN production could increase T cell infiltration and tumor killing in R tumors. The investigators then demonstrate that MAPK hyperactivity induces this immune evasion through PGE2 and IFN. PGE2 signaling was shown to disrupt the acquisition of an inflammatory state in myeloid cells and blocking PGE2 could restore these functions. In vitro dose escalation experiments demonstrated that increasing doses of PGE2 can inhibit the stimulatory effect of IFN on myeloid cells. Together, these findings suggested MAPK-induced downregulation of IFN-I and increase in PGE2 leads to tumor immune evasion through regulating the inflammatory status in myeloid cells. Lastly, the authors leveraged these findings to identify opportunities for translating these mechanisms into treatment strategies to enhance tumor immunity. Specifically, treatment with COX inhibitors such as Celecoxib or 5-Azacytidine that induces IFN-I was shown to transform the tumor immune microenvironment by increasing DCs, inflammatory monocytes, and infiltration of T cells. The effect of Celecoxib was further shown to be synergized by treatment with Flt3L, which induces DC expansion. Together, these studies demonstrate novel mechanisms underlying tumor immune evasion and offer new opportunities for development of therapeutic strategies.

Overall, this is an important body of work from an outstanding group of investigators that build on extensive prior work, and they are to be commended for these studies. This manuscript adds substantially to the current literature with a clear path to translation – however a few limitations exist with the manuscript in its current form and it could be significantly improved by addressing several key points.

Major points to address:

1. Inclusion of human data to strengthen the translational studies: In its current iteration, the manuscript focuses mainly on very elegant studies in pre-clinical models. The authors summarize recent studies showing the association of NSAIDs with improved response to immunotherapy, thus providing some translational relevance. However it would strengthen the manuscript substantially if additional human data could be included. Are any human tissues available for query that might help validate these observations in pre-clinical models? Given the ability to interrogate FFPE samples using spatial transcriptomics and other novel platforms, there could be avenues through which such studies could be explored. However, this could admittedly be outside the scope of this work – in which case this limitation should at least be acknowledged and flagged as an area of future investigation for the field.
2. Limitations associated with targeted selection of candidates: It would be very helpful to provide an explanation as to why PGE2 was specifically chosen among the 4 metabolites enriched in R (Figure 2b). Specifically, some of these other candidates such as LTB4 have been shown to affect tumor immunity and the collective effect of these metabolites along with PGE2 should be taken into consideration.
3. Inclusion of rationale for selection of timepoints in animal studies: Throughout the presented experiments, particularly in Figure 1, various timepoints post-ACT have been used for different measurements (some at 48 h, others at 72, 96 or 5 days post-Tx). It would be helpful to the readers if an explanation is provided on the rationale behind selecting these different timepoints.
4. Limitations associated with conclusions made in Figure 1: The statement "Collectively, these findings suggest that cell hubs containing cDC1s, CCR7+ DCs and, unexpectedly, a substantial fraction of monocytes, have an impact on intratumoral T cell expansion and penetration into tumors." should be interpreted with caution and reworded, as imaging demonstrating physical proximity of cells does not indicate interaction and functional impact. I recommend the authors to also evaluate these interactions in the context of intra-tumoral T cell injection experiments (Figure 1i) and evaluate association between

physical proximity and T cell functional status.

5. Evaluation of the baseline TME prior to T cell injection in R and N tumors in opposing flanks (figure 1m): This is an elegant and informative experiment. I wonder if the authors have tried injecting T cells in R tumors to assess distribution to N tumors on the opposing flank. The question is whether the IT-sensitive TME can affect the R tumors through secreted factors, as has been shown in other contexts, and enhance initial intratumoral proliferation of T cells in R and potential distribution to the N tumors. Alternatively, the authors can provide a description of the tumor immune environment (pre T cell injection) in standalone R tumors compared to those with an N tumor on the other flank.

Minor points:

1. The CTRL and ACT colors in Figure 1 show as blue and red but referred to as gray and red in the legend. Please correct accordingly to avoid confusion.
2. In Figure 1b, it would be helpful to demonstrate longitudinal images of T cell infiltration, similar to other experiments.
3. In Figure 1i, presenting numbers relative to total T cells (as opposed to CD45 alive) could be more informative and relevant to the conclusions made.
4. In line 164, please add a citation that shows an immunosuppressive function for PGE2.
5. Please provide tumor volume data for experiment shown in Figure 4f.

Referee #3

(Remarks to the Author)

This paper examines the reprogramming of the myeloid compartment in the tumor microenvironment triggered by cancer cells using an extensive set of genetically modified melanoma disease models. The authors show that cancer cell-induced reprogramming leads to PGE2-secretion and type I interferon down-regulation, creating immune escape. Genetic or pharmacological disruption of the COX-2-PGE2 axis or restoration of IFN-I production facilitates the expansion of T cells and re-sensitizes resistant tumors to immunotherapy. The authors also examine available patient data sets and suggest future combination therapies. While the work in multiple disease models is very elaborate and sophisticated, some aspects of that work and particularly of the work with patient data sets, may require some more attention. Some comments and concerns that should be addressed are:

- 1) The description of the changes in immune cell subsets in the COX1/2 knockout and IRF3/7 overexpressing resistant tumors compared to the resistant controls and sensitive tumors does not match what the data show fully. In line 218-219, the authors says that these genetic manipulations shift the phenotype closer to that of sensitive tumors. And then go on to discuss details. However, looking at Fig 3 there are also very distinct differences between the RTT-IRF3/7 and the RTT Ptgs1/2 models (not commented on), and while I agree that some monocyte/macrophage subsets come back compared to the RTT CTRL there are also subsets that do not come back in the IRF3/7 model and TAM subsets that are retained in the ptgs1/2 model – this should be commented on so that the description aligns better with the data.
- 2) In Fig 4a-e and Ext data Fig 6, the authors look at publicly available transcriptome data sets (?) from melanoma patients using different inflammatory gene expression signatures and correlate with disease outcomes. It is unclear how these correlations were done and how the data was accessed. The methods description is largely lacking for this part.
- 3) For fig 4i the authors should explain to the reader why they used an anti-CD8 antibody (depletion).
- 4) In Extended Data Fig 6h and I, the authors summarize retrospective studies of patients with melanoma and non-small cell lung cancer on immunotherapy and receiving COX inhibitors (from refs 51 and 52). It is not stated which drugs the patients were on (apart from aspirin), and the data are extracted from published literature. I would think it would be better just to use a verbatim description of the findings and refer the reader to the references rather than providing such a summary, which could give the impression that some meta-analysis has been done in the present paper.
- 5) Line 316-18: Mouse models of TB have given somewhat conflicting results with respect to the use of COX2 inhibitors depending on whether the infection was induced by injection or inhalation. Human studies of COX-2 in TB in combination with vaccine boosting have indicated that COX-2 inhibition may not be feasible.
- 6) In the last paragraph of the discussion (lines 400-402), the authors suggest combination therapies of COX2i, Flt3L, and 5-AZA to facilitate durable tumor control while patients are on immune checkpoint inhibitor therapy. These suggestions are made based on studies in mice models combining either COX2i and 5-AZA or COX2i and Flt3L, but the three drugs have not been combined. Furthermore, combinations with immune checkpoint inhibitors have not been tested. Given the considerable differences between the mouse and human immune systems, I think further studies in human immune cells and human tumors would be necessary and could, with benefit also be provided here. Moreover, the toxicities and possible adverse effects of such triple combinations on top of immune checkpoint inhibitors could be prohibitory, and safety concerns would require stepwise testing. As a minimum, I think the authors need to state clearly that further studies into effects in human models and safety assessments would be warranted prior to patient studies.

Referee #4

(Remarks to the Author)

In their manuscript entitled "Counter-regulated PGE2 and IFN-I create an immune evasive tumor microenvironment", Anna

Obenaus and colleagues present data of their most recent analyses addressing the molecular and cellular mechanisms underlying the cross-resistance of BRAF-mutated melanoma cells to targeted MAPK-inhibition and to T cell-directed immunotherapy. They show that cross-resistant BRAF-mutated melanoma cells reprogrammed the intratumoral myeloid compartment towards an immunosuppressive phenotype through MAPK-dependent secretion of prostaglandin E2 (PGE2) and down-regulation of cancer-cell derived type I IFNs, “reminiscent of immune escape strategies exploited by various pathogens.” By abrogating PGE2 production or by restoring type I IFN production in cancer cells, they could re-sensitize immune evasive tumours to destruction by cytotoxic CD8+ T cells.

By investigating the molecular mechanisms how cancer cells create an immunosuppressive microenvironment, the authors address a clinically very important issue in melanoma immunotherapy. The finding that T cells not only spatially organize with dendritic cells but also with large numbers of IFN-activated inflammatory monocytes in “inflammatory hubs” of the tumor microenvironment (where their activity is restimulated) is relatively novel. In a very recent review Mellman and colleagues (Immunity, October 2023) designated this as a “cancer immunity subcycle”. The work also touches upon the dynamic phenotypic plasticity of mononuclear phagocytes (dendritic cells, monocytes, and macrophages) that are recruited into and reprogrammed within the tumor microenvironment. This process is currently only incompletely understood.

The first and last author of the current paper are among the authors of a closely related paper entitled “Tumor-derived prostaglandin E2 programs cDC1 dysfunction to impair intratumoral orchestration of anti-cancer T cell responses” that was recently published in Immunity (reference number 12 in the paper). They should carve out and explain more clearly to what extent their new data now contribute to advance the very interesting concept that prostaglandin signalling inhibits type I IFN responses and promotes immune evasion.

EP2 and EP4 are G α s-coupled GPCRs. Wu et al. recently described that a “GPCR–G α s–PKA signaling axis promotes T cell dysfunction and cancer immunotherapy failure” (Nat. Immunol 2021, <https://doi.org/10.1038/s41590-023-01529-7>). Have the authors considered this molecular connection?

A thorough analysis of PGE2 production by carefully selected human melanoma cell lines and their interaction with dendritic cells would certainly add significance to their work. If such experiments would confirm the clinical relevance of the findings this would be an important step forward.

Specific questions and suggestions for improvement

1. In Figure 1a targeted therapy-naive (N-TT) and resistant (R-TT) BRAF-V600E mouse melanoma cells (YUMM1.7OVA) were injected into syngeneic Rag2^{-/-} mice and treated with adoptively transferred CD8⁺ OT-I T cells. These melanoma cell lines were also originally generated in syngeneic Rag2^{-/-} mice. What was the rationale for using Rag2^{-/-} instead of immunocompetent mice? Adoptive CD8⁺ T cell therapies in Rag2^{-/-} mice fail to recapitulate the contribution of CD4⁺ helper and regulatory T cells (and in particular their interaction with Ag-presenting myeloid cells in the TME). This experimental limitation should be clearly labelled in the figure and mentioned in the main text. The results would be significantly strengthened, if they were shown with another matched pair of N-TT and R-TT melanoma cell lines that originate in and can be transplanted back in immunocompetent syngeneic mice. The bioluminescence images are unlabelled - which image belongs to which group?
2. In Figure 1b/n/p many points do not show error bars. Do the tumours regress so synchronously? Individual growth curves of mice, ideally presented as tumour diameter, would be beneficial to understand the variability of tumour size between mice in the same cohort.
3. Figure 1c (and Fig ED1) show scRNAseq data from one single ACT-treated N-TT and R-TT tumour each. Data from at least 3-4 tumours per group need to be generated and shown to draw valid conclusions. For comparison untreated N-TT and R-TT should also be analysed. Why was the 48 hour post-ACT time-point chosen, since in Figure 1k shows stronger CD8⁺ T cell infiltration after 72 hr?
4. The graph in Fig. 1d is difficult to understand. What does a “normalized fraction of cells per cluster” mean? It would be more informative if relative numbers of the different immune cell subsets per condition were shown (similar to the quantitation in Extended Data 2b).
5. Figure 1f and Extended Data Fig. 2b show that T cells interacted predominantly with monocytes, which present by far the largest population of MHC II+ cells, and much less commonly with cDC1s. The authors conclude that “unexpectedly, a substantial fraction of monocytes, have an impact on intratumoral T cell expansion and penetration into tumors.” This important finding is highlighted again in Fig 3 and should be emphasized much more strongly. The authors could also present more details regarding the different monocyte states in each condition and discuss their importance in comparison to cDCs. This could for example be incorporated in Fig. 1d.
6. In Figure 2 the authors first present evidence that resistant (R-TT) YUMM1.7 BRAF-mutant melanoma cells show upregulated prostaglandin synthesis and downregulated type I IFN signalling. Genetic knock-out of Ptg1/2 restores responsiveness to adoptive T cell transfer therapy (Fig. 2g). Was this experiment also performed in Rag2^{-/-}?
7. From Fig. 2h,i, resistant (R-TT) YUMM3.3 BRAF-mutant melanoma cells are used, this time in wild-type and Rag2^{-/-} mice. Here, genetic knock-out of Ptg1/2 led to tumor rejection in wild-type but not Rag2^{-/-} mice. It seems as if YUMM3.3 and YUMM1.7 BRAF-mutant melanoma cells behave fundamentally different. The authors should explain and comment on this.

Of importance: these cell lines genetically differ with respect to the expression of PTEN, a known regulator of anti-melanoma immunity (Peng et al. *Cancer Discovery* 2016, <https://doi.org/10.1158/2159-8290.CD-15-0283>).

For a recent review of cancer cell-intrinsic mechanisms driving immune evasion see: Ghorani et al. (*Immunity* 2023, <https://doi.org/10.1016/j.immuni.2023.09.004>). In addition, PTEN has been shown to regulate IFN responses via IRF3 (Li et al., 2016 *Nat Immunol*, <https://doi.org/10.1038/ni.3311>). The authors should discuss the role of PTEN expression in their models and the potential impact on the observed results.

8. In Fig. 2j it is not clear which cell lines were used to assess the production of IFN β . Both the production of PGE2 and the secretion of type I IFNs should be shown for all cell lines used.

9. Figure 2p: What does the 6/6 refer to, since one line does not reach a volume of 0 mm³? Why are only four lines shown?

10. Figure 3a (and Extended Data Fig. 4a -d) show scRNAseq data for CD45⁺ cells isolated from YUMM1.7OVA RTT-CTRL, RTT-Ptgs1/2 KO and RTT-IRF3/7 overexpressing tumours 72h after ACT treatment. The data are compared to the results from YUMM1.7OVA RTT 48h after ACT treatment (from Fig.1). What is the immune cell composition of untreated Rtt Ptgs1/2 KO and Rtt IRF3/7 tumours? Data from the same time point and a comparison with untreated tumours should be shown. Moreover, the figure legend says that n=3 tumours were pooled from RTT tumours, however the number of cells is similar to the single NTT tumour. Are the UMAP plots representative of one tumour from each group? Again, the graph in Fig. 3b (now labelled as "percent of cells per cluster across conditions") is difficult to understand. It could be replaced by relative numbers of the different immune cell subsets per condition (similar to Fig. 1d).

11. Figure 3c: Activated CD8⁺ T cells produce IFN γ in the TME. Has the role and contribution of IFN γ on the myeloid compartment been assessed?

12. Interestingly, the growth of melanomas could still be controlled by adoptively transferred T cells in Rag2^{-/-} Batf3^{-/-} mice that lack cDC1s (Figure 3h,i). Are inflammatory monocytes able to compensate the function of cDC1? Individual growth curves should be shown.

13. The data presented in Extended Data Fig. 3h-k suggest that PGE2 upregulation and type I IFN downregulation result from oncogenic MAPK signalling. Is this specific to the two mouse cell lines used or might this be a more general phenomenon? The authors should investigate this also in human cell lines, taking into account that loss of PTEN also impacts on type I IFN regulation.

14. The correlation shown in Fig. 4b-e between the extent of CD8 T cell infiltration in human melanomas and the expression of "TME-COX signature" (containing also Stat1, Isg15, Cxcl9, Cxcl10, and several HLA molecule genes) or the "TMIS signature" (containing also Isg15, Irf7, Ifit3, Ifit1b) in bulk RNAseq data is expected. Numerous studies have shown that key IFN-induced genes correlate with the presence of CD8 T cells in tumors (and with the response to immune checkpoint blockade immunotherapy). This data does not provide any support for the conclusion of the Figure (PGE2 impairs the acquisition of an interferon-induced inflammatory state in myeloid cells).

15. An important experiment was performed with YUMM1.7OVA RTT melanoma cells in mice where CD11c⁺ immune cells lack the prostaglandin receptors EP2 and EP4. Flow cytometry shows increased infiltration with T cells as well as with inflammatory DC and monocytes (Fig. 4f-h). Did this restore tumor growth control?

16. The experiment with YUMM3.3.RTT melanoma cells shown in Fig. 4i does not support any conclusion about the role of prostaglandin receptors on myeloid cells for T cell-mediated anti-tumor immunity, because the group of "CTRL" mice did not receive CD8 T cell depletion.

17. Figure 4f / 5a: It is unclear if this is intravenous or intratumoural CD8⁺ T cell transfer. Please clarify.

18. In Fig. 5b scRNAseq data of ACT-treated YUMM1.7OVA RTT tumors with or without COX inhibitors are shown. Again, the graph in Fig. 5c (labelled as "percent of cells per cluster across conditions") could be replaced by relative numbers of the different immune cell subsets per condition (similar to Fig. 1d and Fig. 3b).

19. The adoptive T cell transfer therapies in Fig. 5 are most likely also performed in Rag2^{-/-} mice. This should be specifically stated. Were T cells injected intravenously in 5a,d,h? These results are impressive. Individual growth curves for the experiment depicted in Fig. 5k should be presented (similar to Fig. 5g).

Version 2:

Reviewer comments:

Referee #1

(Remarks to the Author)

The manuscript previously entitled "Counter-regulated PGE2 and IFN 1 -I create an immune evasive tumor microenvironment" has now been extensively revised and retitled "Cancer cells impair monocyte-mediated intratumoral T cell stimulation to evade immunity."

- In the single cell analysis the inflammatory monocytes seem to have differentiated into inhibitory macrophages therefore burden of proof the monocytes are a/monocytes and not AXL DC2 and b/ have not simply become inhibitory macrophages as we enter the story. The authors have chosen in the revision to expand the monocyte story which has added novelty and a more clear direction to the narrative. The level and extent of revision is impressive and the interrogation of this work is to be commended and relentlessly thorough.
- The differentiation towards inhibitory macs is of course still of interest but beyond the scope of the current work.
- Extensive effort in the revision to really tackle the antigen presenting direct vs cross-dressed presentation is evident.
- The authors are highly responsive to all reviewers.
- Minor: Extended data Fig3f- DC depletion after a single dose of DT lasts 48 hrs therefore the legend should include some mention of length of time of DC depletion
- The authors have done a great and thorough job with the revision.
- However as pointed out in the last round of revision there was a lack of novelty in the original framing around IFN and now the novelty of this manuscript, which is very different from their prior work really rests on the importance of this inflammatory monocyte derived cross-dressing mechanism. It's therefore somewhat unclear why a simple comparison of the level of OT1 priming in vivo in the absence of DC vs these monocytes was not conducted side by side to test the relative contribution as likely both mechanisms are important but many compensatory mechanisms for antigen presentation occur in the absence of DC1. They would implant tumors into the relative CCR2 (or ly6A) ko vs BATf3 ko mice and measure tumor growth side by side +/- addition of CFSE ;labelled OT1 T cells at Day 9, or for acute loss could compare an inflammatory monocyte specific DTR vs DC specific DTR as BM chimera for parity. Very sorry if this was missed as there is a lot of key data in the extended but this is the final experiment really needed to show the relative importance of this mechanism side by side as the push back from some of the DC1 field may be significant.

It is curious the authors who are superbly wielding use of DTRs and IFN antibody depletion etc did not take a more direct approach of loss of the monocytes versus DC to see how this normalizes the RTT and NTT difference. I am concerned a little that this was not directly addressed as the authors themselves state the impact of this revision rests with monocytes as they have published on the IFN PGE axis before.

Otherwise this is truly a wonderful revision and the authors should be commended on their efforts.

Referee #2

(Remarks to the Author)

In the revised manuscript by Elewaut et al entitled "Cancer cells impair monocyte-mediated intratumoral T cell stimulation to evade immunity", the authors build upon their previous work and prior literature demonstrating that acquired resistance to targeted therapy can impair response to subsequent immunotherapy. The authors use elegant animal models and experimental designs complemented with validating clinical studies to explore the mechanisms underlying T cell restimulation in the tumor microenvironment and identify an inflammatory state in monocytes that facilitates T cell expansion in a cDC-independent manner. The authors further demonstrate that cancer cells produce PGE₂, which impacts the myeloid cell response to IFN- γ and their inflammatory state in the tumor microenvironment and impairs T cell stimulation. Lastly, the authors demonstrate an association between NSAID co-medication and response to ICB in melanoma and NSCLC patients and further explore the role of pharmacological modulation of PGE₂ in changing tumor immunity and response to immunotherapy in a number of mouse models.

Overall, this is an outstanding body of work that provides novel insights into the mechanisms underlying tumor immune evasion and as such, holds great promise for advancing the field with clear translational applications. In this revision, the authors have done a tremendous job addressing concerns and strengthening the manuscript. The inclusion of additional pre-clinical and clinical studies in this revision has significantly bolstered the study. The experimental designs and findings appropriately support the conclusions made.

In essence, the authors addressed all reviewer comments in an elegant and thoughtful manner. These findings will be of great interest to the broad readership of Nature.

Referee #3

(Remarks to the Author)

I have read the revised paper and the responses to my comments and those of the other reviewers. In aggregate, this paper expands our understanding of the "triangular interplay" between cancer cells, myeloid cells, and T cells in the TME and brings forward novel insights. It is superbly executed, and a lot of data has been added, making it a very comprehensive analysis and a huge, solid body of work. Some of the insight may have implication for treatment of patients on ICIs and combinations and how we use COX inhibitors.

With respect to my comments 3.1 – 3.6, they have been addressed to my satisfaction.

Referee #4

(Remarks to the Author)

In their revised manuscript now entitled "Cancer cells impair monocyte-mediated intratumoral T cell stimulation to evade

immunity”, Anna Obenauf and colleagues now provide a large amount of new experimental data that allowed them to conceptually advance their insights into the molecular and cellular mechanisms underlying the cross-resistance of BRAF-mutated melanoma cells to targeted MAPK-inhibition and to T cell-directed immunotherapy. I was very impressed by the thorough implementation of all the referee’s suggestions that have greatly improved the manuscript.

By focussing on the very original and important observation that inflammatory monocytes / macrophages play an important role in supporting T cell effector functions in tumour tissues and – specifically - by showing that peptide-MHC-I “cross-dressing” plays an important role, the authors now present a conceptual advance in the field. This clearly demarcates their work from their previous related manuscripts. The new data and analyses that connect their experimental work to the clinical care of patients is impressive and greatly expands the relevance of their work.

The authors have addressed most of my comments very thoroughly. However, one aspect remains. I still did not understand what the graphs that relate to their scRNAseq data (now Figs. 1d, 4b, 5c, ED Figs. 1f, 1i, 10g) are showing. They now call this “relative frequency (across conditions)”. In their former manuscript these graphs were called “normalized fraction of cells per cluster”. The authors need to explain how they calculate and what they intend to convey with these graphs. I could not find any information in the methods.

The authors now explain why they mostly used Rag knockout mice with a focus on effector functions of CD8 T cells and their interaction with inflammatory monocytes / macrophages. The cooperation of T cells with monocytes/macrophages and the role of interferons have also been reported by recent papers. The concept of “monocyte-mediated intratumoral T cell stimulation” has also been described by other recent papers that show how this cooperation expands the mechanisms of tumour cell killing beyond classical cytotoxicity. For example: Kruse et al. Nature 2023 (PMID: 37316667; <https://doi.org/10.1038/s41586-023-06199-x>), van Elsas et al Cancer Cell 2024 (PMID: 38759656; <https://doi.org/10.1016/j.ccell.2024.04.011>). The authors might consider citing these papers in their discussion.

Version 3:

Reviewer comments:

Referee #1

(Remarks to the Author)

Thank you for addressing all outstanding concerns. I have none further and commend the authors on this beautiful and important body of work.

The images or other third party material in this Peer Review File are included in the article’s Creative Commons license,

Point-by-point reply to the reviewer's comments

(*Reviewer comments in blue italics*, our responses in black)

Summary for all four referees

First, we would like to thank all four referees for their thorough evaluation of our work and the positive feedback highlighting the technical quality and scope of our study, as well as the relevance of our findings. We were particularly excited to read that the reviewers found that our work is “*superbly executed (Rev. 1)*” and “*very elaborate and sophisticated (Rev. 3)*”, addressing “*a clinically very important issue in melanoma immunotherapy (Rev. 4)*” and that it “*adds substantially to the current literature with a clear path to translation (Rev. 2)*”. We are equally grateful to all four referees for their thoughtful comments, constructive criticism, and specific suggestions.

One key conclusion we drew from the collective feedback of the reviewers was that our initial manuscript did not sufficiently “*highlight the truly innovative or surprising findings (Rev. 1)*” our study contributes to a very active field of research and, specifically, to our still limited understanding of the triangular interplay between cancer cells, myeloid cells and T cells in the tumor microenvironment (TME). We have taken these and all other comments of the reviewers fully on board, have thoroughly revised our manuscript and added a substantial amount of new experimental data, which corroborate and further expand our novel findings. We provide a detailed point-by-point response to each of the referee's comments below. Here, we would like to briefly summarize additional experiments and new insights we provide in our revised manuscript:

- We have substantially expanded our mechanistic work uncovering inflammatory monocytes as central players in intratumoral T cell restimulation, a function that has so far been mainly attributed to conventional dendritic cells (cDC1s). Notably, as opposed to cDC1s that cross-present antigens, we now show that inflammatory monocytes source and present intact pMHC-I complexes from cancer cells through cross-dressing, which in concert with the expression of key cytokines for T cell function (e.g., *Cxcl9*, *Cxcl10*, *Il15*) enables them to restimulate T cells, even in the absence of cDC1s. This novel and exciting dimension of our study has not been observed before and challenges the current view that DCs are the main class of immune cells that can promote intratumoral T cell restimulation.
- We now elucidated the ontogeny and regulation governance of inflammatory monocytes showing that their functional inflammatory state is instructed by the coordinated action of type-I and type II IFN and dampened by PGE₂. Although these factors have previously been shown to modulate activities of T cells and cDC1s, their coordinated impact on inflammatory monocytes in the TME as a major axis modulating T cell restimulation and antitumor immunity has not been described before.
- To expand on our findings that oncogenic MAPK signaling counter-regulates the production of IFN-I and PGE₂ by cancer cells to instruct an immune-suppressive TME, we validated this regulatory function of the MAPK pathway in clinically relevant human cancer cell-lines. We further show that PGE₂ and IFN-I levels present in conditioned media from human cancer cells converge to dampen the inflammatory state of human monocyte models. To our knowledge, a function of oncogenic MAPK signaling in counter-regulating the production of PGE₂ and IFN-I and its impact on intratumoral T cell restimulation has not been reported previously.

- Beyond these new functional experiments in human models, we have addressed comments and suggestions of the reviewers concerning the human relevance of our findings by adding a substantial amount of additional human data. Specifically, we identify inflammatory macrophages as the putative analogous population to inflammatory monocytes, show in melanoma patient biopsies that such inflammatory macrophages co-localize with activated CD8⁺ T cells in inflammatory hubs, and demonstrate that changes in the TME upon PGE₂ depletion and IFN-I reinstatement are predictive for response to immunotherapy in two patient cohorts. Moreover, we provide a new meta-analysis of patients receiving NSAIDs and immunotherapy, which support a conserved immune-modulatory function of PGE₂ in humans, and further implicate COX1/2 inhibition with mechanism-based combinations, such as those we propose, as a promising concept for boosting T cell mediated anti-tumor immunity.
- In addition, we further expanded on our surprising finding that T cells, upon re-stimulation in inflammatory hubs of N^{TT} tumors, can infiltrate and eradicate distant immune-evasive R^{TT} tumors. This raises the potential translational avenue of restoring a functional “cancer immunity subcycle” in a single metastatic lesion to serve as a reservoir for T cell expansion, impacting distant metastatic disease. While this concept remains speculative at this point and will require further investigation, we have discussed this potential implication of our findings in our revised manuscript.

Taken together, our work reveals comprehensive cellular and mechanisms in intratumoral T cell re-activation, uncovers inflammatory monocytes as a central player in the recently proposed “cancer immunity subcycle”, and delineates a cascade of events through which cancer cells shape an immune-evasive TME. Based on the collective feedback of the reviewers, we have decided to change the title of our manuscript to “**Cancer cells impair monocyte-mediated intratumoral T cell stimulation to evade immunity**”, which in our view much better conveys the scope and novel findings of our study.

Once more, we would like to thank all four reviewers for their time and their truly thoughtful comments and suggestions, which were extremely helpful to better highlight and further enrich the novel insights provided by our study and tighten the overall concept. We hope the referees share our enthusiasm about the new insights we have gained and are satisfied with our revision.

Reviewer #1:

R1.1: This study applies somewhat limited patient data and mainly mouse models to test how the TME influences myeloid reprogramming, focusing on the PGE/Type I IFN axis. The authors show loss of tumor intrinsic PGE2 and Type I IFN IRF3/7; and that restoration of both axes can restore DC function. They also demonstrate PGE2 impairs acquisition of a type I induced inflammatory state in DC, and that MAPK mediates this suppressive axis in the tumor. Given extensive prior work in PGE2 and cDC1 suppression in the field (see citations below) some of what is seen is somewhat descriptive and may be seen as an extension of the prior work, unless further mechanistic interrogation of the MAPK mediated suppressive axis, the impact of these pathways on DC differentiation, and regulation governance is documented. Despite this the work is superbly executed, controlled and the experiments are very thoughtfully and rigorously designed. The writing also reflects an outstanding grasp of the field but perhaps fails to highlight the truly innovative or surprising findings, which be corrected easily.

We really appreciate that the reviewer considers our study to be superbly executed and rigorous. In terms of novelty, we believe that our original manuscript did not place sufficient emphasis on novel aspects of our work (hinted at by Rev. 1 themselves). As summed up above and detailed further in R1.2, in our revised manuscript, inflammatory monocytes take the center stage, which we contrast and compared to cDCs, well-established mediators of immunity. This is the first-time inflammatory monocytes have been implicated in restimulation of tumor infiltrating CD8⁺ T cells, and the first time the link between T cell restimulation and cancer cell-mediated regulation of PGE₂/IFN-I has been established. To place more emphasis on this novel and exciting dimension of our study, we have restructured our manuscript. We have now dedicated **New Fig. 2** to inflammatory monocytes, providing new insights into their function, specifically focusing on their role in T cell restimulation, their ontogeny and regulation governance, and how PGE₂ impairs the acquisition of a functional inflammatory monocyte state (**New Fig. 4**). Furthermore, we have identified cellular counterparts of murine inflammatory monocytes in human cancers and demonstrated their co-localization with T cells in inflammatory hubs (**New Fig. 2l, m**). Altogether, we believe that these new findings set our manuscript apart from previous studies on the immune-evasive properties of PGE₂.

*R1.2: While Figure 4 attempts to meet this gap, an improved study may map **how restoration of these pathways impacts DC tumor antigen trafficking, or local T cell recall by DC in vivo**. Direct **ex vivo tumor antigen presentation** should be simple, or peptide MHC bound on the cell surface, DC T cell interactions in these hubs etc could also be shown. Eg. Prior work by Bottcher and Bayerl (<https://pubmed.ncbi.nlm.nih.gov/37315536/>) showed PGE2 through EP2/4 and IRF8.*

We thank the reviewer for these suggestions. We now have assessed antigen presentation on DCs upon PGE₂ depletion or IFN-I reinstatement across our conditions (N^{TT}, R^{TT}, R^{TT} *Ptgs2* KO and R^{TT} IRF3/7). Notably, lowering PGE₂ levels or reactivating IFN-I signaling in R^{TT} tumors not only increased the abundance of cDC1s, cDC2s, and inflammatory cDC2s (**Fig. 4a, b**) and enhanced their functionality (i.e., *Il12b*, *Il15*, *Cd40/80/86*, *Cxcl9/10*) (**Extended Data Fig. 6e**), but also reinstated antigen-presentation capacity, as measured by H2-K^b-SIINFEKL complex formation (**New Extended Data Fig. 6f**). However, we would like to point out that the main focus of our work was on characterizing the previously unknown function of monocytes as key regulators of CD8⁺ T cell restimulation, so we focused on further analysis of inflammatory monocytes rather than DCs.

Remarkably, we found that immune-permissive tumors are controlled by antigen-specific CD8⁺ T cells in *Batf3*^{-/-} and *Zbtb46*-DTR mice, where all DC subsets are depleted (**New Fig. 2a, b, 3m and**

New Extended Data Fig. 3f-i). This, together with the expression of Nur77 (a reporter of TCR signaling) on T cells that co-localized with monocytes, suggested that monocytes engage TCRs via MHC-I to induce restimulation (**New Fig. 1g and Extended Data Fig. 2e, f**). Notably, inflammatory monocytes lack expression of genes associated with cross-presentation (*Clec9a, Wdfy4*) (**New Fig. 2c**), yet we found that they can display the SIINFEKL peptide from OVA-expressing tumors on MHC-I (**New Fig. 2f**). This led us to propose that inflammatory monocytes, similar to inflammatory cDC2s¹, may acquire intact pMHC-I complexes directly from cancer cells, a phenomenon known as “MHC-I cross-dressing”^{2,3}. To assess cross-dressing, we implanted N^{TT} tumors (C57BL/6 origin, H2-K^b) into H2-K^d-exclusive BALB/c mice and monitored H2-K^b expression on monocytes. Only inflammatory monocytes, and not their non-inflammatory counterparts, were H2-K^b positive in the H2-K^d-exclusive BALB/c environment and promoted OT-1 T cell stimulation *in vitro* (**New Fig. 2h-j**). Importantly, when we depleted MHC-I through knockout of *B2m* in N^{TT} cancer cells, we did not detect H2-K^b on inflammatory monocytes, highlighting that pMHC-I complexes were sourced from N^{TT} cancer cells (**New Fig. 2i and Extended Data Fig. 3n**). Whether cross-dressing through cancer-cell derived pMHC-I complexes determines the stimulatory capacity of inflammatory monocytes is best discerned in the absence of cross-presenting cDC1s. Thus, we engrafted N^{TT} and N^{TT} *B2m* KO tumors into *Rag2*^{-/-} and *Rag2*^{-/-} *Batf3*^{-/-} mice and performed ACT with OT-1^{Luc} T cells. Indeed, in the absence of cDC1s, T cells expanded only in N^{TT} CTRL, but not in N^{TT} *B2m* KO tumors, whereas in the presence of cDC1s both N^{TT} CTRL and N^{TT} *B2m* KO tumors accumulated T cells (**New Fig. 2k**). This indicates that the process of 'cross-dressing' by inflammatory monocytes is crucial for T cell expansion. We also show that the inflammatory phenotype and subsequent ability to cross-dress are significantly hindered by high levels of PGE₂ and low levels of IFN-I (**Fig. 3a, b, f, h-n**).

As requested by the reviewer, we addressed T cell memory formation and local recall by using mice that previously harbored an R^{TT} tumor and fully responded to COX2i in combination with ACT. At day 130, 9 mice remained tumor free (n=3 COX2i, n=4 COX2i + Flt3L and n=2 COX2i + 5-AZA) and these were rechallenged (in the absence of any treatment) by injecting R^{TT} cells subcutaneously, together with a cohort of naive mice as controls. While R^{TT} tumors efficiently formed in naive mice after 10 days, in 100% of the rechallenged mice, tumors were rapidly rejected (**New Extended Data Fig. 11a**). This indicates that our proposed therapy combinations aimed at remodeling the TME facilitate a cytotoxic T cell response that can eliminate primary tumors and foster memory T cell formation and tumor control upon rechallenge.

[REDACTED]

[REDACTED]

R1.4: The most obvious interpretation of Fig1c is the TME of RTT is conferring maturation of monocytes to TAMs

We fully agree with the reviewer's perspective on monocyte maturation, which we have further expanded upon in our revised manuscript. The new data includes RNA velocity analyses of our scRNA-seq dataset predicting that the Monocyte 1 cluster is a common precursor of TAMs and inflammatory monocytes, and SCENIC analysis that identified IRFs and STATs as specific regulators of inflammatory monocytes (**New Fig. 2e, 4g and Extended Data Fig. 7h**). We hypothesized that the lack of IFN-I and high levels of PGE₂ in R^{TT} tumors preclude the transition from monocytes to inflammatory monocytes by impairing the IRF/STAT transcriptional program, fostering their maturation into TAMs. Indeed, exposing murine bone marrow-derived dendritic cells (BM-DCs), Ly6C⁺ monocytes, or human monocyte models to conditioned media (CM) from R^{TT} mouse or human cells, respectively, led to a decreased expression of ISGs compared to CM from N^{TT}. This could be reverted by exposing R^{TT} cells to COX1/2i during media conditioning, and the effect was abolished upon IFNAR1 blockade (**New Fig. 4k-m and New Extended Data Fig. 8c-g**). This suggests that when PGE₂ levels are significantly reduced, very low amounts of type I IFN are still capable of inducing an inflammatory state. To further probe monocyte fates within the different TMEs, we transferred Ly6C⁺ monocytes intratumorally into N^{TT} and R^{TT} tumors. Only in N^{TT} tumors monocytes were able to acquire an inflammatory state (**New Fig. 4h and Extended Data Fig. 7j**). Moreover, *in vitro* differentiation of primary monocytes into macrophages was enhanced by the addition of PGE₂, highlighting that PGE₂ not only impairs the differentiation of monocytes into inflammatory monocytes, but also favors the differentiation of monocytes into TAMs with immunosuppressive features (**New Extended Data Fig. 7i**).

R1.5: Multiparameter IF is a particularly nice aspect of this manuscript with respect to quantifying and phenotyping multicellular clusters. Single cell and imaging: annotation CCR7+ DC is appropriate for the high mig/nat phenotype.

We thank the reviewer for appreciating the multiparameter IF in our study. We have extended these analyses by incorporating additional stainings and quantifications to show that OT-1 antigen-specific T cells associate with inflammatory monocytes in inflammatory hubs within immune-responsive N^{TT} tumors established in *Batf3*^{-/-} mice (**New Fig. 2b and Extended Data Fig. 3d, e**). We also used multiparameter IF to show that CD8⁺ T cells located in close proximity to monocytes were frequently Nur77^{GFP}-positive, which suggested that monocytes actively engage with CD8⁺ T cell TCRs via MHC-I, i.e. by direct immunological interaction rather than mere physical association (**New Fig. 1g and Extended Data Fig. 2e, f and 3l**). We further validated key findings in human myeloid scRNAseq datasets from melanoma and lung cancer patients^{5,6} and our spatial transcriptomics (CosMx) on biopsies of melanoma patients (**New Fig. 2l, m and Extended Data Fig. 4a-c and 7e**). Analogous to our findings in mouse models, CXCL9/10⁺ expressing macrophages, the cell type we identified as

human counterparts of inflammatory monocytes, spatially organized with CD8⁺ T cells in human melanoma samples (**New Fig. 2m**).

R1.6: Experiments injecting the T cells directly into tumor and examining differentiation are particularly helpful and thoughtful Fig1l, preactivation of OT1 and FTY720 to show local tumor specific differentiation cues etc

We appreciate the reviewer's enthusiasm for the insights provided by this experimental setup. In our revised manuscript, we have included a scRNA-seq dataset from T cells to assess differentiation (**Fig. 1l and Extended Data Fig. 2j-l**) and expanded on the contralateral injection experiments to assess T cell functionality (**Extended Data Fig. 2t, u**).

R1.7: Authors show restoration of T cell infiltration after IRF3/7 expression in R^{TT} tumors, with this is ICB responsiveness also restored? T cell differentiation?

Indeed, in immune-evasive R^{TT} tumors of the YUMM3.3 model, IRF3/7 overexpression was sufficient to facilitate T cell differentiation (i.e. a stronger expansion of the CD8⁺ T cell pool and higher proportion of differentiated TIM3⁺ TCF1⁻ T cells from *in vivo* tumors; (**Rebuttal Fig. 2**) and mediate tumor rejection, even without the need for ICB. Thus, the YUMM3.3 model complements data from the YUMM1.7^{OVA} model, in which IRF3/7 expression in R^{TT} tumors also leads to the re-establishment of an immune-permissive TME that is dominated by inflammatory monocytes and re-sensitized to immunotherapy via adoptive T cell transfer (ACT).

Rebuttal Fig. 2: Scheme depicting the injection of YUMM3.3 R^{TT} CTRL and R^{TT} IRF3/7 cells into C57BL/6 mice followed by flow cytometry analysis of TCF1 and TIM-3 (left) and stacked bar graph depicting the percentage of CD8⁺ T cells of CD45⁺ cells in YUMM3.3 R^{TT} CTRL and R^{TT} IRF3/7 with the corresponding fraction of TCF1⁻ TIM-3⁺ and TCF1⁺ TIM-3⁻ within CD8⁺ T cells. Bar graphs depict the mean ± s.e.m.

R1.8: The pharmacological testing in Fig 5 is a strength and the authors should relate this to use of Cox2 inhibitors in patients and ICB responsivity or decreased MAPK resistance- are there any other studies than ref 51/52 on this?

We thank the reviewer for highlighting the importance of our pharmacological studies. Beyond references 51 and 52, we identified additional datasets in which NSAID use was documented in melanoma and NSCLC patients receiving ICB, which we used to conduct an independent meta-analysis. Our analysis revealed a significant association between the use of NSAIDs and improved overall response rate to immunotherapy, as well as enhanced progression-free survival (**New Fig. 5a, New Extended Data Fig. 9a-d and New Supplementary Table 5 and 6**). We also found that cessation of NSAID treatment led to rapid tumor relapse, suggesting that constant T cell restimulation

is required for maintaining T cell function (**Fig. 5d**). This could provide insights into why widely used NSAIDs ultimately fail to elicit durable benefit in patients. To further strengthen clinical implications of our work, we used COX2i in combination with Flt3L and 5-AZA in additional murine MAPK-driven models that show poor responses to ICB, i.e. the pancreatic cancer model EPP2 (KRAS^{G12D}, TP53^{-/-}, MYC); the colorectal cancer model CT26 (KRAS^{G12D}); and the lung cancer model KPAR (KRAS^{G12D}, TP53^{-/-}). Our data demonstrate that COX2i combinations significantly enhance the efficacy of ICB treatment in all these models (**New Extended Data Fig. 11b-o**). This addition broadens the relevance of our study beyond melanoma and underscores the promising therapeutic prospect of COX2i combinations, providing a solid foundation for future clinical studies.

R1.9: Could the authors kindly improve the manuscript beyond the descriptive aspects of Fig4 to better document *in vivo* or *ex vivo* functional differences and differentiation trajectories?

We thank the reviewer for the constructive feedback and the suggestion to better document *in vivo* or *ex vivo* functional differences and differentiation trajectories of myeloid cells. As further detailed in R1.4 and R1.11, we have delineated monocyte trajectories and antigen presentation functions on monocytes and cDCs. In our revised manuscript, we now dedicate a major part of **New Fig. 2 and New Fig. 4** to inflammatory monocytes, providing new insights into their function, specifically focusing on their role in T cell restimulation, their ontogeny, and how PGE₂ impairs the acquisition of the functional inflammatory state.

R1.10: Flt3L alone controls are needed e.g. extended data 7.

We have now performed the suggested control experiments (**New Extended Data Fig. 10j, 11d, f**).

R1.11: Please delineate ISGs as Type I unique and demonstrate no change in type II responses given common Stat1/IRF 1 activity. Many shown are Type II targets as well. Or control experiments with anti IFN γ R1 blocking antibodies in at least a couple instances.

We thank the reviewer for this insightful suggestion, which together with our finding that inflammatory monocytes further expand upon ACT (**Fig. 1c, d and New Extended Fig. 1e, 6b**) prompted us to specifically assess the differential contributions of IFN-I and IFN- γ on the myeloid compartment.

We leveraged single-cell scRNA-seq data from the TME and conducted a pseudo bulk analysis to computationally assess the expression differences between N^{TT} and R^{TT} before and after ACT. Focusing on the top 300 variable genes, this revealed three defined expression clusters (**New Extended Data Fig. 7f**): **Cluster #3** contains genes consistently upregulated in the TME of R^{TT} compared to N^{TT}, independently of ACT. **Cluster #2** contains genes that are consistently downregulated in the TME of R^{TT} compared to N^{TT} tumors, regardless of ACT. These genes are predominantly type I IFN-driven ISGs including *Irf7*, *Isg15*, *Cxcl10*, or *Oas2*, in line with the diminished inflammatory state of R^{TT} tumors. Moreover, as brought up by the reviewer, some of these genes are integral to both type I and type II IFN programs, and therefore influenced by IFN- α/β and IFN- γ . One example is *Stat1*, which plays a central role on both type I and type II IFN signal transduction. **Cluster #1** consists of genes that are upregulated following ACT treatment, particularly in N^{TT} tumors. These genes are primarily IFN γ -driven ISGs including *Gbp2* or *Cxcl9*. Altogether, these data strongly indicate that IFN- γ also plays a role in establishing the inflammatory state of the TME, as astutely raised by the reviewer.

Next, we specifically blocked IFN-I and IFN-II before and after ACT. In N^{TT} tumors, inflammatory monocytes are present before ACT (~25% of CD45⁺ cells) and fully rely on IFN-I, as blocking the type I IFN receptor (α IFNAR1 antibody administration), but not blocking IFN γ (α IFN γ

neutralizing antibody administration), depleted all Ly6A⁺ inflammatory monocytes (**New Fig. 4f**). This confirms that, pre-ACT, cancer cell-derived IFN-I is the main driver of the inflammatory state. Upon T cell transfer, inflammatory monocytes expand dramatically making up to ~60% of the CD45⁺ cells, further suggesting a contribution of T cell-derived IFN γ (**New Fig. 1c, d and Extended Data Fig. 1e**). After ACT, individual treatments with α IFNAR1 or α IFN γ reduced the number of inflammatory monocytes considerably, but only the combined blocking of IFN-I and IFN-II was able to deplete all inflammatory monocytes (**New Fig. 4f**). This suggests that, in N^{TT} tumors, monocytes initially rely on cancer-derived IFN-I, which shifts towards both IFN-I and IFN γ after ACT.

In R^{TT} *Ptgs1/2* KO tumors, inflammatory monocytes make up ~10% of CD45⁺ cells before ACT. Similar to N^{TT} tumors, blocking IFNAR1 resulted in a complete depletion of inflammatory monocytes prior to ACT. Upon ACT we observed a striking expansion of inflammatory monocytes and blocking IFN-I reduced the abundance of inflammatory monocytes, which was even more striking with depletion of IFN γ (**New Extended Data Fig. 7g**). This indicates that the generation of inflammatory monocytes is initially dictated by the low levels of cancer cell-derived IFN-I in *Ptgs1/2* KO tumors. In absence of PGE₂, the maintenance of inflammatory monocytes upon ACT more strongly depends on T cell-derived IFN γ .

R1.12: Overall this is a beautifully executed study which may not quite meet the level of conceptual impact needed in its current form.

We would like to thank the reviewer once more for highlighting the quality of our work and the many thoughtful and extremely helpful comments and suggestions, which guided us during our revision. Most importantly, in our revised manuscript we have tried to better emphasize and further expand “*the truly innovative or surprising findings*” of our study by restructuring our manuscript and providing additional experimental data in support for our main conclusions. In our view, these major revisions and additions have further strengthened our manuscript and much more clearly convey the conceptual novelty and impact of our work, and we hope the reviewer shares this view.

R1.13: Other: Omission of citation to edelson and murphy first showing BATF3 cDC1 confer antitumoral immunity. Omission of citation to Nirschl et al which first showed identification of DC migration/maturation program in melanoma (227hi) when referencing what has now been coopted as DC3/mreg etc and review Pittet JEM as well reconciling these. Key review to cite Sancho lab *Mat Rev immuno* Wculek et al 2020 also not cited.

We thank the referee for pointing out these highly relevant and impactful papers on DC biology and have included these citations in our revised manuscript.

Reviewer #2:

In the manuscript by Elewaut et al entitled “Counter-regulated PGE2 and IFN-I create and immune evasive tumor microenvironment”, the authors use well-established melanoma models of immunotherapy resistance to investigate the mechanisms involved in the establishment of an immune-evasive tumor microenvironment (TME) and propose therapeutic strategies to reverse these mechanisms. The authors demonstrated a loss of monocytes and cDC populations and an increase in immunosuppressive TAM in IT-resistant tumors. It was further shown that in IT-sensitive tumors, T cells interact with monocyte and cDC1 populations and that the TME supported T cell expansion. When the T cell egress from lymph nodes was blocked, T cell recruitment and expansion was still maintained, suggesting that the TME drives this process. To determine the mechanisms by which TME dictates T cell anti-tumor activity, the authors conducted RNAseq and demonstrated an upregulation of the PGE2 and a downregulation of IFN-I in IT-resistant tumors. Blocking PGE2 and inducing IFN production could increase T cell infiltration and tumor killing in R tumors. The investigators then demonstrate the MAPK hyperactivity induces this immune evasion through PGE2 and IFN. PGE2 signaling was shown to disrupt the acquisition of an inflammatory state in myeloid cells and blocking PGE2 could restore these functions. In vitro dose escalation experiments demonstrated that increasing doses of PGE2 can inhibit the stimulatory effect of IFN on myeloid cells. Together, these findings suggested MAPK-induced downregulation of IFN-I and increase in PGE2 leads to tumor immune evasion through regulating the inflammatory status in myeloid cells. Lastly, the authors leveraged these findings to identify opportunities for translating these mechanisms into treatment strategies to enhance tumor immunity. Specifically, treatment with COX inhibitors such as Celecoxib or 5-Azacitidine that induces IFN-I was shown to transform the tumor immune microenvironment by increasing DCs, inflammatory monocytes, and infiltration of T cells. The effect of Celcoxib was further shown to be synergized by treatment with Flt3L, which induces DC expansion. Together, these studies demonstrate novel mechanisms underlying tumor immune evasion and offer new opportunities for development of therapeutic strategies.

Overall, this is an important body of work from an outstanding group of investigators that build on extensive prior work, and they are to be commended for these studies. This manuscript adds substantially to the current literature with a clear path to translation – however a few limitations exist with the manuscript in its current form and it could be significantly improved by addressing several key points.

We are extremely grateful for this excellent summary of our study and the enthusiastic comments about the quality, importance, and translational potential of our work.

Major points to address:

R2.1: *Inclusion of human data to strengthen the translational studies: In its current iteration, the manuscript focuses mainly on very elegant studies in pre-clinical models. The authors summarize recent studies showing the association of NSAIDs with improved response to immunotherapy, thus providing some translational relevance. However it would strengthen the manuscript substantially if additional human data could be included. Are any human tissues available for query that might help validate these observations in pre-clinical models? Given the ability to interrogate FFPE samples using spatial transcriptomics and other novel platforms, there could be avenues through which such studies could be explored. However, this could admittedly be outside the scope of this work – in which*

case this limitation should at least be acknowledged and flagged as an area of future investigation for the field.

We thank the referee for these excellent suggestions and fully agree that additional human data would complement our extensive studies in pre-clinical models and further enhance the impact of our work. As one of our main priorities, in our revised manuscript we have added a substantial amount of human data analyses to explore the relevance of major conclusions from pre-clinical models in human patients. Specifically,

1. We now include analyses of several new human myeloid scRNAseq datasets from melanoma and lung cancer patients, as well as spatial transcriptomics (CosMx) analyses in biopsies from melanoma patients. In scRNA-seq analyses, we show that mouse inflammatory monocytes correspond to human inflammatory macrophages with high expression of interferon stimulated genes (ISGs), which are annotated as ISG⁺ macrophages and CXCL9/10⁺ macrophages (**New Fig. 2l and Extended Data Fig. 4a-c**). Spatial transcriptomics on human melanoma samples further revealed that human inflammatory macrophages characterized by CXCL9/10 expression spatially organize with CD8⁺ T cells, analogous to inflammatory monocytes in our pre-clinical models (**New Fig. 2m**).
2. In our initial manuscript, we had re-analyzed one publicly available RNA-seq dataset with clinical response information from melanoma patients treated with ICB⁷, where we had scored the TME-COX and TMIS signature. In our revised manuscript, we have included the TME-COX and TME-IRF3/7 signatures, which capture TME changes upon PGE₂ depletion and IRF3/7 overexpression. Remarkably, both signatures correlated with CD8⁺ T cell presence in the tumor and were able to predict ICB response and survival, which further supports the clinical relevance of our models (**New Extended Data Fig. 7a-d**). In addition, we took advantage of a recently published study with scRNA-seq data from 13 patients with metastatic melanoma, who received tumor infiltrating lymphocyte (TIL)-ACT therapy after progressing on ICB treatment⁶. We found that our TME-COX and TME-IRF3/7 signatures scored higher in responding compared to non-responding patients, further indicating that TME changes we observe upon PGE₂ depletion and IFN restoration are predictive for response to immunotherapy (**New Fig. 4c and New Extended Data Fig. 7e**).
3. We also assessed PGE₂ and IFN-I production in three clinically relevant human BRAF^{V600E}-driven melanoma cell-lines (A375, M249, LOX) and a KRAS^{G12C}-driven NSCLC cell-line (NCI-H358). For each of these cell lines, we established matched pairs of targeted-therapy naive (N^{TT}) and a targeted-therapy resistant (R^{TT}) sublines. Across all these models, we detect elevated levels of PGE₂ and reduced levels of IFN-I in the R^{TT} lines, compared to their N^{TT} counterparts, highlighting the conserved counter-regulation of PGE₂ and IFN-I upon MAPK pathway reactivation in targeted therapy-resistant human tumors (**New Fig. 3n and New Extended Data Fig. 5r**).
4. To explore how differences in PGE₂ and IFN-I production in human cancer cells affect human myeloid cells, we utilized two human monocyte models: the MONO-MAC-1 and the BLaER1 cell-line⁸. We exposed these models to conditioned media (CM) from the matched human N^{TT} and R^{TT} derivatives (A375, M249, LOX and H358). Consistent across all these models, we found that CM from N^{TT} cells induced an inflammatory state in human monocytes, characterized by upregulation of ISGs. In contrast, CM from R^{TT} cells did not induce ISG expression. Notably, CM of R^{TT} cells treated with COX1/2 inhibitor during media conditioning led to a recovery of ISG expression in monocytes (**New Fig. 4m and New Extended Data Fig. 8g**). Together, these findings confirm that the counter-regulation of PGE₂ and IFN-I in targeted therapy-resistant human cancer cells consistently impairs the inflammatory activation of human monocytes across different models.

5. Finally, we have further investigated associations between NSAID treatment and the response to immunotherapy in available clinical datasets. Besides clinical studies we had cited in our initial manuscript, we have now conducted our own comprehensive meta-analysis, including new cohorts of melanoma and lung cancer patients undergoing immunotherapy that were previously not included (**New Fig. 5a and New Extended Data Fig. 9a-d**). In line with previous meta-analyses, we found a significant correlation between the use of NSAIDs and improved clinical outcomes in patients receiving immunotherapy. While these results in patients are encouraging in itself, our mechanistic studies provide much needed insights into how NSAIDs lead to a better immune response and how this could be improved through mechanism-based therapy combinations. Importantly, our findings strongly indicate that an early discontinuation of COX2i treatment leads to rapid tumor outgrowth, despite initial tumor control, suggesting that continuous CD8⁺ T cell restimulation within the TME is required to maintain a productive T cell response.

R2.2: Limitations associated with targeted selection of candidates: It would be very helpful to provide an explanation as to why PGE₂ was specifically chosen among the 4 metabolites enriched in R (Figure 2b). Specifically, some of these other candidates such as LTB₄ have been shown to affect tumor immunity and the collective effect of these metabolites along with PGE₂ should be taken into consideration.

We thank the reviewer for pointing this out - our focus on PGE₂ was indeed not sufficiently explained. We chose to focus on PGE₂, because our prior analysis identified PGE₂ as the most abundantly produced eicosanoid in R^{TT} cells. However, this finding was not sufficiently represented in former Figure 2b, which depicted relative differences in metabolite levels between N^{TT} and R^{TT} tumors instead of absolute levels. To more accurately quantify the abundance of these metabolites, we have now performed targeted metabolomics using high-performance liquid chromatography coupled to mass spectrometry (HPLC-MS). We analyzed the concentrations of specific eicosanoids (PGE₂, PGD₂, PGF_α, PGI, LTB₄) and normalized the results to the initial tumor weight (**New Extended Data Fig. 5a**). This targeted analysis confirmed our previous rationale for focusing on PGE₂ as the predominant prostaglandin present in these tumor models (>100-fold higher than others).

R2.3: Inclusion of rationale for selection of timepoints in animal studies: Throughout the presented experiments, particularly in Figure 1, various timepoints post-ACT have been used for different measurements (some at 48 h, others at 72, 96 or 5 days post-Tx). It would be helpful to the readers if an explanation is provided on the rationale behind selecting these different timepoints.

We thank the reviewer for pointing out this inconsistency. In our revised manuscript, we have harmonized BLI images to show T cell abundance 96 hours post ACT. Additionally, we now provide detailed information on our rationale for selecting specific time-points in the main text and the Methods section.

R2.4: Limitations associated with conclusions made in Figure 1: The statement “Collectively, these findings suggest that cell hubs containing cDC1s, CCR7+ DCs and, unexpectedly, a substantial fraction of monocytes, have an impact on intratumoral T cell expansion and penetration into tumors.” should be interpreted with caution and reworded, as imaging demonstrating physical proximity of cells does not indicate interaction and functional impact. I recommend the authors to also evaluate these interactions in the context of intra-tumoral T cell injection experiments (Figure 1i) and evaluate association between physical proximity and T cell functional status.

We thank the reviewer for this word of caution. We fully agree and have revised this statement, which now reads (line 142-144): "Collectively, these findings suggest that multicellular hubs containing cDCs and, unexpectedly, a substantial fraction of monocytes, are associated with intratumoral CD8⁺ T cell expansion, penetration into tumors, and tumor control (**Fig. 1e-g**).".

To address the reviewer's comment and determine whether T cells directly interact with monocytes and cDC1s within inflammatory hubs, we analyzed TCR stimulation in adoptively transferred OT-1 T cells equipped with a Nur77^{GFP} reporter, serving as a marker for early TCR signaling activation (**New Fig. 1g**). Our multiparameter immunofluorescence analysis showed that T cells located in close proximity to monocytes and cDC1 were frequently Nur77^{GFP} positive, suggesting that CD8⁺ T cell TCRs are actively engaged via MHC-I in a direct immunological interaction. Conversely, in R^{TT} tumors, T cells rarely co-localized with the few cDC1s and monocytes present and were mostly negative for Nur77^{GFP} (**New Extended Data Fig. 2e, f and 3I**). Flow cytometry of pre-activated OT-1^{Luc} T cells injected intratumorally also revealed a significantly higher fraction of Nur77⁺ T cells in N^{TT} tumors compared to R^{TT} tumors, highlighting TCR engagement (**Rebuttal Fig. 3**). Furthermore, we injected T cells intratumorally into N^{TT} and R^{TT} tumors, isolated them after 5 days, and performed scRNA-seq. This revealed that in N^{TT} tumors T cells acquire gene expression profiles characteristic of effector memory and progenitor exhausted T cells (Tpex), states previously associated with immunotherapy response⁹⁻¹², which was substantially diminished in R^{TT} tumors (**New Fig. 1I and New Extended Data Fig. 2j, k**). T cells in N^{TT} tumors also became primarily CXCR6⁺, which indicates an effector phenotype in CD8⁺ T cells^{13,14} (**New Extended Data Fig. 2I**). Remarkably, even in mice lacking cDCs, T cells were found in clusters with monocytes, and tumors were controlled by T cells (**New Fig. 2a, b, Fig. 3n and New Extended Data Fig. 3f-i**), further highlighting the important role of monocytes in T cell restimulation within the TME, which was previously assigned only to a rare population of cDC1s.

Rebuttal Fig. 3: Quantification of Nur77⁺ T cells after intratumoral injection of CD8⁺ OT-1 T cells in N^{TT} and R^{TT} tumors. Tumors were isolated 5 days post injection and analyzed by flow cytometry. Bar graphs depict the mean \pm s.e.m. Statistical analysis was performed with a two-tailed unpaired student's *t*-test. *****p*<0.0001.

R2.5: Evaluation of the baseline TME prior to T cell injection in R and N tumors in opposing flanks (figure 1m): This is an elegant and informative experiment. I wonder if the authors have tried injecting T cells in R tumors to assess distribution to N tumors on the opposing flank. The question is whether the IT-sensitive TME can affect the R tumors through secreted factors, as has been shown in other contexts, and enhance initial intratumoral proliferation of T cells in R and potential distribution to the N tumors. Alternatively, the authors can provide a description of the tumor immune environment (pre T cell injection) in standalone R tumors compared to those with an N tumor on the other flank.

We thank the reviewer for these excellent suggestions. In our revised manuscript, we added new experiments demonstrating that the presence of an N^{TT} tumor on the opposing flank has no impact on the immune composition of R^{TT} tumors prior to T cell transfer. This strongly suggests that T cells become restimulated in an immune permissive TME and then migrate to R^{TT} tumors without systemic

changes induced by N^{TT} tumors (**Extended Data Fig. 2q**). We also repeated the intratumoral injection experiment presented in the former Fig. 1m and added a group of mice with N^{TT} and R^{TT} tumors, where we inject T cells intratumorally into R^{TT} tumors (**Extended Data Fig. 2t, u**). We found that T cells injected into R^{TT} tumors initially do not expand (over 24-72 hours), but they migrate into N^{TT} tumors and further expand in the local N^{TT} TME (starting from 96 hours post intratumoral injection of T cells). After 120-144h, we observed a significant expansion of the BLI signal in the injected R^{TT} tumors, which points towards T cells trafficking back from N^{TT} to R^{TT} tumors. Altogether these findings indicate the local R^{TT} TME is unable to support T cell expansion even in the presence of an N^{TT} tumor but does not make the T cells irreversibly dysfunctional as they can still traffic to N^{TT} tumors on the opposite flank and there expand upon restimulation. We have included representative images of selected time-points together with growth curves in new **Extended Data Fig. 2t, u**.

R2.6: Minor points:

2.6.1. *The CTRL and ACT colors in Figure 1 show as blue and red but referred to as gray and red in the legend. Please correct accordingly to avoid confusion.*

We thank the reviewer for spotting this and have eliminated the color names (gray/red) to avoid confusion.

2.6.2. *In Figure 1b, it would be helpful to demonstrate longitudinal images of T cell infiltration, similar to other experiments.*

We appreciate this suggestion and have added these longitudinal images to our extended data (**New Extended Data Fig. 1a**).

2.6.3. *In Figure 1i, presenting numbers relative to total T cells (as opposed to CD45 alive) could be more informative and relevant to the conclusions made.*

We have changed the visualization of these data to show a stacked bar graph indicating the T cell states within the total amount of T cells out of CD45⁺ live cells (**New Fig. 1k**). This plot better highlights that the number of CD3⁺CD8⁺ T cells are significantly lower in R^{TT} than in N^{TT} after intratumoral T cell injection and that there are much fewer differentiated T cells (TCF1⁺PD1⁺TIM3⁺) in these tumors. We think that presenting relative numbers out of total T cells would not accurately consider differences in the abundance of T cells, and we hope that the reviewer appreciates this point.

2.6.4. *In line 164, please add a citation that shows an immunosuppressive function for PGE2.*

We thank the reviewer for pointing this out and have added relevant citations.

2.6.5. *Please provide tumor volume data for experiment shown in Figure 4f.*

We have added tumor volume data for former Figure 4f, which is now shown in **New Extended Data Fig. 8a**.

Once more, we would like to thank the referee for their very thoughtful and constructive feedback and the excellent suggestions, which were extremely helpful for further strengthening our manuscript.

Reviewer #3:

This paper examines the reprogramming of the myeloid compartment in the tumor microenvironment triggered by cancer cells using an extensive set of genetically modified melanoma disease models. The authors show that cancer cell-induced reprogramming leads to PGE2-secretion and type I interferon down-regulation, creating immune escape. Genetic or pharmacological disruption of the COX-2-PGE2 axis or restoration of IFN-I production facilitates the expansion of T cells and re-sensitizes resistant tumors to immunotherapy. The authors also examine available patient data sets and suggest future combination therapies. While the work in multiple disease models is very elaborate and sophisticated, some aspects of that work and particularly of the work with patient data sets, may require some more attention. Some comments and concerns that should be addressed are:

We thank the reviewer for highlighting the scope and quality of our experimental work and we are also grateful for the constructive criticism and excellent suggestions, which we have addressed by revising and substantially expanding our study, including analyses of patient datasets.

***R3.1:** The description of the changes in immune cell subsets in the COX1/2 knockout and IRF3/7 overexpressing resistant tumors compared to the resistant controls and sensitive tumors does not match what the data show fully. In line 218-219, the authors say that these genetic manipulations shift the phenotype closer to that of sensitive tumors. And then go on to discuss details. However, looking at Fig 3 there are also very distinct differences between the RTT-IRF3/7 and the RTT Ptgs1/2 models (not commented on), and while I agree that some monocyte/macrophage subsets come back compared to the RTT CTRL there are also subsets that do not come back in the IRF3/7 model and TAM subsets that are retained in the ptgs1/2 model – this should be commented on so that the description aligns better with the data.*

We thank the reviewer for raising this important point. We fully agree that a more nuanced description of the immune cellular changes within the TME between different conditions is helpful, and we have revised these descriptions in our revised manuscript accordingly. The TME of R^{TT} tumors contains marginal numbers of inflammatory monocytes and is instead dominated by a large TAM compartment. In the TME of R^{TT} Ptgs2 KO tumors, we observe a prominent recovery of monocytes and inflammatory monocytes, as well as cDC1s and CCR7⁺ DCs. TAM subsets are slightly reduced and in a more immune-stimulatory state (**New Fig. 4a, b**). In R^{TT} IRF3/7 tumors, we observe a very large monocyte and inflammatory monocyte cluster, increased cDC subsets and strong reduction in TAMs (**New Fig. 4a, b**). In our revised manuscript we dissect these differences and find that both PGE₂ and IFN act together to modulate the myeloid compartment (**New Fig. 4g-n and Extended Data Fig. 7h-j**).

RNA velocity analyses on our scRNA-seq dataset reveal Monocyte 1 cluster as the common precursor of TAMs and inflammatory monocytes (**New Fig. 4g and Extended Data Fig. 7h**). Further computational analyses together with extensive FACS-based characterizations and functional experiments suggest the following cellular trajectories and interactions: (i) Prior to adoptive transfer of activated CD8⁺ T cells (pre-ACT), the generation of inflammatory monocytes is dictated by cancer cell-derived IFN-I, which in turn facilitates the primary expansion of T cells (**Fig. 1c, New Fig. 4 and Extended Data Fig. 1e, 6b, 7g**); (ii) Upon ACT, T cell-derived IFN- γ further suppresses monocyte differentiation into TAMs and bolsters their inflammatory state (**New Fig. 4f and Extended Data Fig. 6b, 7g**); (iii) R^{TT} tumors produce very low levels of IFN-I and are therefore not conducive to T cell expansion. Consequently, their TME is not exposed to sufficient IFN- γ from T cells, maintaining an immune-evasive TME with low abundance of inflammatory monocytes (**Fig. 1c and Extended Data**

Fig. 6b, 7g). In our revised manuscript, we have described these insights and differences between conditions in greater detail and full alignment with the presented data.

R3.2: In Fig 4a-e and Ext data Fig 6, the authors look at publicly available transcriptome data sets (?) from melanoma patients using different inflammatory gene expression signatures and correlate with disease outcomes. It is unclear how these correlations were done and how the data was accessed. The methods description is largely lacking for this part.

We apologize for omitting this - we have updated our Methods section to provide detailed information about how these analyses were conducted. In addition, we have expanded these comparative analyses and have now also evaluated our TME-COX and TME-IRF3/7 signatures using a recently published scRNAseq dataset from 13 patients with metastatic melanoma, who received tumor infiltrating lymphocyte (TIL)-ACT therapy after progressing on ICB treatment⁶. We found that our TME-COX and TME-IRF3/7 derived signatures scored higher in responder patients (n=6) compared to non-responders (n=7), providing further support for the clinical relevance of TME changes observed upon PGE₂ depletion or IFN-I reinstatement in our models, also for TIL-based immunotherapies (**New Fig. 4c and New Extended Data Fig. 7e**).

R3.3: For fig 4i the authors should explain to the reader why they used an anti-CD8 antibody (depletion).

We thank the reviewer for raising this point and apologize for the lack of clarity. We used an anti-CD8 antibody to deplete CD8⁺ T cells and verify that the observed tumor rejection in CD11c-Cre Ep2/Ep4 KO mice was primarily mediated by CD8⁺ T cell antitumor immunity. In our revised manuscript, we have explained our reasoning and experimental design in the main text and figure legends.

R3.4: In Extended Data Fig 6h and I, the authors summarize retrospective studies of patients with melanoma and non-small cell lung cancer on immunotherapy and receiving COX inhibitors (from refs 51 and 52). It is not stated which drugs the patients were on (apart from aspirin), and the data are extracted from published literature. I would think it would be better just to use a verbatim description of the findings and refer the reader to the references rather than providing such a summary, which could give the impression that some meta-analysis has been done in the present paper.

We thank the reviewer for this point – our initial manuscript did not contain an own meta-analysis, and we did not intend to give this impression. Seeking to gain more confidence in the effect of COX inhibitors on tumor immunity, we now have indeed conducted our own comprehensive meta-analysis, focusing on cohorts of melanoma and lung cancer patients undergoing immunotherapy, where aspirin/NSAID use was reported (**New Fig. 5a and New Extended Data Fig. 9a-d**). In this analysis, we included several studies not covered in previous references 51 and 52^{15,16} and found a significant correlation between the use of aspirin/NSAIDs and improved clinical outcomes in patients receiving immunotherapy, adding to the increasing body of evidence that COX inhibition can enhance the response to immunotherapies. Details of these cohorts, including the identification of the new studies, specific COX inhibitors used, number of patients in each study etc. are presented in **New Extended Data Fig. 9d, New Supplementary Table 5 and 6** and further elaborated in our Methods section.

R3.5: Line 316-18: Mouse models of TB have given somewhat conflicting results with respect to the use of COX2 inhibitors depending on whether the infection was induced by injection or inhalation. Human studies of COX-2 in TB in combination with vaccine boosting have indicated that COX-2 inhibition may not be feasible.

We appreciate these learnings from TB and acknowledge that there are likely differences between the role of PGE₂ in the context of a TB infection and cancer. However, we had to trim down some parts of our original text to make room for the description of new experimental data, and now briefly mention this potential parallel in our discussion.

R3.6: In the last paragraph of the discussion (lines 400-402), the authors suggest combination therapies of COX2i, Flt3L, and 5-AZA to facilitate durable tumor control while patients are on immune checkpoint inhibitor therapy. These suggestions are made based on studies in mice models combining either COX2i and 5-AZA or COX2i and Flt3L, but the three drugs have not been combined. Furthermore, combinations with immune checkpoint inhibitors have not been tested. Given the considerable differences between the mouse and human immune systems, I think further studies in human immune cells and human tumors would be necessary and could, with benefit also be provided here. Moreover, the toxicities and possible adverse effects of such triple combinations on top of immune checkpoint inhibitors could be prohibitory, and safety concerns would require stepwise testing. As a minimum, I think the authors need to state clearly that further studies into effects in human models and safety assessments would be warranted prior to patient studies.

We thank the referee for raising these important points. We have now expanded our discussion to address some of the limitations of our work and now clearly state that further studies into safety assessments would be warranted prior to patient studies (**line 554-556**).

Regarding therapy combinations, we did not intend to propose a triple combination of COX2i, Flt3L, and 5-AZA, and fully agree with the reviewer that this might lead to toxicities and adverse events. To directly address the reviewer's point, we have tested the triple combination of COX2i, Flt3L and 5-AZA with ICB or ACT in two melanoma mouse models (YUMM 3.3 and YUMM 1.7). The triple combination only slightly improved tumor control in the YUMM 3.3 model and led to no improvement in YUMM 1.7, likely due to redundant changes in the TME (**Rebuttal Fig. 4**). Based on our findings, we propose the use of COX2i in combination with FLT3L, or *alternatively* with 5-AZA, which is already in clinical use for the treatment of leukemias and, in our preclinical models, enhances tumor control over COX2i alone (**New Fig. 5h and Extended Data Fig. 11d, e**). To substantiate these findings, we now have evaluated the combination of COX2i with either Flt3L or 5-AZA together with immune checkpoint inhibitors in three additional mouse models (colorectal cancer: CT-26, lung cancer: KPAR, and pancreatic cancer: EPP2). Remarkably, in all three models, the combination of COX2i with either Flt3L or 5-AZA improves the response to immune checkpoint inhibition (**New Extended Data Fig. 11g-o**). We believe, these mechanism-based interventions warrant further exploration to identify patient groups that can benefit from the addition of COX2i + Flt3L/5-AZA to current immunotherapies (e.g., through screening of biomarkers informative of the TME status, such as IFN/PGE₂ levels or inflammatory markers in tumor biopsies). We speculate that these combinations may be most beneficial in tumors where pro-immunogenic inflammatory markers are low.

We also agree with the referee that additional studies on human immune cells and tumors would further strengthen our findings. In response, we have extended our validation efforts beyond mouse models to include human cancer cell-lines (**Fig. 3n and Extended Data Fig. 5r, s**), human immune cells (**Fig. 4m and Extended Data Fig. 8g**) and patient data (**Fig. 2m, 4c and Extended Data Fig. 7a-e**) focusing particularly on inflammatory monocytes, which now take the center stage in our revised manuscript.

Once more, we would like to thank the reviewer for their thoughtful comments, constructive criticism, and excellent suggestions, which were extremely helpful to revise and further improve our manuscript.

Rebuttal Fig. 4: a, Growth of YUMM1.7^{OVA} R^{TT} tumors injected into *Rag2*^{-/-} mice treated with the triple combination of COX2i + Fit3L + 5-AZA followed by ACT ($n=7$ mice). Arrow depicts day of ACT (left) and survival plot of YUMM1.7^{OVA} R^{TT} tumors injected into *Rag2*^{-/-} mice treated with ACT and the triple combination of COX2i + Fit3L + 5-AZA ($n=7$ mice) compared to double combinations of COX2i + Fit3L + ACT ($n=8$ mice) or COX2i + 5-AZA + ACT ($n=7$ mice). **b**, Growth of YUMM3.3 R^{TT} tumors injected into *C57BL/6* mice treated with the triple combination of COX2i + Fit3L + 5-AZA + ICB (left) and survival plot of YUMM3.3 R^{TT} tumors injected into *C57BL/6* mice treated with the triple combination of COX2i + Fit3L + 5-AZA + ICB compared to double combinations of COX2i + Fit3L + ICB or COX2i + 5-AZA + ICB. ICB=anti-PD1/CTLA-4.

Reviewer #4:

In their manuscript entitled “Counter-regulated PGE2 and IFN-I create an immune evasive tumor microenvironment”, Anna Obenauf and colleagues present data of their most recent analyses addressing the molecular and cellular mechanisms underlying the cross-resistance of BRAF-mutated melanoma cells to targeted MAPK-inhibition and to T cell-directed immunotherapy. They show that cross-resistant BRAF-mutated melanoma cells reprogrammed the intratumoral myeloid compartment towards an immunosuppressive phenotype through MAPK-dependent secretion of prostaglandin E2 (PGE2) and down-regulation of cancer-cell derived type I IFNs, “reminiscent of immune escape strategies exploited by various pathogens.” By abrogating PGE2 production or by restoring type I IFN production in cancer cells, they could re-sensitize immune evasive tumours to destruction by cytotoxic CD8+ T cells.

By investigating the molecular mechanisms of how cancer cells create an immunosuppressive microenvironment, the authors address a clinically very important issue in melanoma immunotherapy. The finding that T cells not only spatially organize with dendritic cells but also with large numbers of IFN-activated inflammatory monocytes in “inflammatory hubs” of the tumor microenvironment (where their activity is restimulated) is relatively novel. In a very recent review Mellman and colleagues (Immunity, October 2023) designated this as a “cancer immunity subcycle”. The work also touches upon the dynamic phenotypic plasticity of mononuclear phagocytes (dendritic cells, monocytes, and macrophages) that are recruited into and reprogrammed within the tumor microenvironment. This process is currently only incompletely understood.

We sincerely thank the reviewer for the thorough and thoughtful evaluation of our manuscript, for emphasizing the high clinical relevance of our work, and for providing such insightful suggestions and constructive feedback.

R4.1: *The first and last author of the current paper are among the authors of a closely related paper entitled “Tumor-derived prostaglandin E2 programs cDC1 dysfunction to impair intratumoral orchestration of anti-cancer T cell responses” that was recently published in Immunity (reference number 12 in the paper). They should carve out and explain more clearly to what extent their new data now contribute to advance the very interesting concept that prostaglandin signaling inhibits type I IFN responses and promotes immune evasion.*

We thank the reviewer for raising this point. Together with comments from Reviewer #1, this prompted us to take a critical look at our original manuscript as it may have fallen short from clearly explaining the novelty of our findings and why they represent a major contribution to the field. To address this issue, we have thoroughly revised our text to better emphasize the novelty and significance of our findings in a complex and very active field of research. Specifically, to better highlight key differences and advancements over previous studies focusing on the immune-evasive properties of PGE₂ (such as Bayerl *et al.*, previous reference 12), we emphasize the following points more clearly:

- 1. Inflammatory monocytes mediate intratumoral CD8⁺ T cell restimulation:** While cDC1s have long been implicated as regulators of T cell responses, the focus of our manuscript is on inflammatory monocytes. We show that inflammatory monocytes are capable of restimulating CD8⁺ T cells, even in the absence of cDC1s (e.g. in *Batf3*^{-/-} mice lacking cDC1s, or *Zbtb46*-DTR mice where all DC subsets are depleted, **New Fig. 2a, b, 3m and New Extended Data Fig. 3f-i**). To the best of our knowledge, the pivotal role of inflammatory monocytes in the cancer immunity cycle, particularly through intratumoral CD8⁺ T cell restimulation, has not been previously described and is a key novelty of our manuscript (please see also R4.8).

2. **Inflammatory monocytes employ MHC-I cross-dressing:** Surprisingly, we find that inflammatory monocytes isolated from tumors harbor tumor antigen (OVA)-MHC-I complexes and can restimulate OT-1 T cells *in vitro*, although they are not known to cross-present antigens (**New Fig. 2c, f, g**). We find that monocytes can obtain intact pMHC-I complexes from cancer cells through a mechanism of MHC-I cross-dressing, which is unique to the inflammatory monocyte state (**New Fig. 2f-j**). Interfering with the ability of monocytes to cross-dress prevents restimulation of T cells (**New Fig. 2k**). Cancer cell-derived PGE₂ and low levels of IFN-I attenuate the inflammatory state in monocytes and consequently interfere with MHC-I cross-dressing and T cell restimulation. This is a major new finding of our revision and was previously unknown.
3. **High PGE₂ and low type I IFN impair CD8⁺ T cell restimulation by disrupting inflammatory hubs.** While Bayerl *et al.* focused solely on the suppressive effect of PGE₂ on cDC1s, we find that global reprogramming of myeloid populations is driven by a coordinated interplay between high PGE₂ and low IFN-I, and that this shapes the immune-evasive TME. PGE₂ reduces the abundance of monocytes, facilitates the differentiation into immune-suppressive TAMs and prevents the acquisition of their functional inflammatory state (**New Fig. 4a, b, g-n and New Extended Data Fig. 8a, b**) by raising the threshold at which monocytes respond to cancer cell-derived IFN-I (**New Extended Data Fig. 8c-i**). These data provide significant new insights into the pleiotropic effects of PGE₂ and IFN-I in shaping the phenotypic plasticity of mononuclear phagocytes.
4. **Oncogenic MAPK signaling counter-regulates PGE₂ and IFN-I production in cancer cells.** We find that oncogenic MAPK signaling regulates the PGE₂/IFN-I axis, and that genetic or pharmacological targeting of the PGE₂/IFN-I axis restores an immune-permissive TME (**New Fig. 4a, b, and 5b-c**). While oncogenic MAPK signaling is recognized as a major driver of proliferation, and inhibition of the MAPK pathway leads to an influx of CD8⁺ T cells in tumors that contributes to responses of targeted therapies^{17,18}, its central role in simultaneously regulating PGE₂/IFN-I to disrupt CD8⁺ T cell restimulation and impair anti-tumor immunity has not been previously described.
5. **Implications for rational combination therapies.** Our preclinical models suggest that NSAIDs expand inflammatory monocytes and cDC1s within the TME, effectively reinstating T cell function and leading to tumor control. We also found that cessation of NSAID treatment led to rapid tumor relapse, suggesting that constant T cell restimulation and modulation of the immune-evasive TME is required for maintaining T cell function. This could provide new insights into why widely used NSAIDs ultimately fail to elicit durable benefit in patients. Finally, based on our mechanistic insights, we put forward rational therapy combinations, including clinically approved drugs, and schedules that can be exploited to overcome an immune-evasive TME and elicit durable tumor control in patients (**Fig. 5 and Extended Data Fig. 10 and 11**).

We hope that the reviewer agrees that this sets our work apart from previous studies, thus representing a major advancement over the existing literature.

R4.2: *EP2 and EP4 are Gas-coupled GPCRs. Wu et al. recently described that a “GPCR–Gas–PKA signaling axis promotes T cell dysfunction and cancer immunotherapy failure” (Nat. Immunol 2021, <https://doi.org/10.1038/s41590-023-01529-7>). Have the authors considered this molecular connection?*

We thank the reviewer for making us aware of this interesting study. To investigate this potential molecular connection, we have performed several additional experiments: First, we examined the effects of the EP2/EP4 knockout in antigen-specific CD8⁺ T cells in our models and observed that they were not able to effectively control R^{TT} tumors (**New Extended Data Fig. 7k**). In contrast, deletion of EP2/EP4 receptors in CD11c⁺ myeloid cells elicit a robust CD8⁺ T cell response in R^{TT} tumors (**Fig. 4i, j, and Extended Data Fig. 8a, b**). To investigate whether the GPCR–Gas–PKA signaling axis operates in monocytes and dendritic cells, we treated bone marrow derived DCs (BM-DCs) and monocytes with PGE₂ in the presence or absence of EP2/EP4 inhibitors, or with forskolin and assessed pCREB levels and transcriptional changes. In different myeloid models, two hours of treatment with PGE₂ and forskolin induced CREB phosphorylation, which was blunted by the addition of EP2/4 antagonists (**Rebuttal Fig. 5a**). These data indicate that, in myeloid cells, PGE₂ signals through EP2/4 receptors and activates CREB (in line with PKA activation), similar to what has recently been shown by *Bayerl et al.*,¹⁹. Interestingly, transcriptome profiling 24 hours after incubation with PGE₂ revealed a down-regulation of several ISGs (e.g., *Stat1*, *Irf7*, *Cxcl10* and *Oas2*), which was partially rescued by pre-treating cells with EP2/EP4 antagonists (**Rebuttal Fig. 5b**). Further elucidating this signaling axis downstream of PGE₂ in myeloid cells is an interesting avenue, and we have gained first insights. However, these experiments will require further optimization and mechanistic work on how pCREB regulates ISGs, which we feel lies beyond the scope of this study. Nevertheless, we have briefly reflected on open questions in myeloid PGE₂ signaling in our discussion (line 537-539).

Rebuttal Fig. 5: **a**, Median fluorescence intensity (MFI) of pCREB analyzed by flow cytometry in bone-marrow derived dendritic cells (BM-DCs) or Ly6C⁺ monocytes, treated for 2 hours with PGE₂, Forskolin or PGE₂ + EP2/4 inhibitors. **b**, Heatmap depicting differential gene expression of BM-DCs treated with PGE₂ and EP2/EP4 inhibitors (left). Heatmap depicting the differential gene expression of the human MONO-MAC-1 cell-line treated with PGE₂ for 24 hours (center). Heatmap depicting the differential gene expression of human derived blood monocytes treated with PGE₂ for 24 hours (right). Bar graphs depict the mean ± s.e.m. Statistical analysis was performed in **a** with a one-way ANOVA with Dunnett's multiple comparisons test. **p*<0.05, ****p*<0.001. ns=not significant.

R4.3: A thorough analysis of PGE₂ production by carefully selected human melanoma cell lines and their interaction with dendritic cells would certainly add significance to their work. If such experiments would confirm the clinical relevance of the findings this would be an important step forward.

We are grateful for the insightful suggestion. In response, we now have assessed PGE₂ and IFN-I production in three clinically relevant human BRAF^{V600E}-driven melanoma cell lines (A375, M249, LOX) and a KRAS^{G12C}-driven NSCLC cell-line (NCI-H358). For each of these cell lines, we derived a targeted-therapy naive (N^{TT}) and a targeted-therapy resistant (R^{TT}) counterpart. Across all models, we found higher levels of PGE₂ and lower levels of IFN-I in the R^{TT} cell models, compared to their N^{TT} counterparts and overexpression of NRAS in A375 cells led to higher levels of COX2 protein levels (**New Fig. 3n and New Extended Data Fig. 5r, s**). We also explored how differences in PGE₂ and

IFN-I production in human cancer cells affect human myeloid cells, using two human monocyte models (MONO-MAC-1 and BLaER1 cells). We exposed these models to conditioned media (CM) from the matched human N^{TT} and R^{TT} derivatives (A375, M249, LOX and H358). Consistent across all these models, we found that CM from N^{TT} cells induced an inflammatory state in human monocytes, characterized by upregulation of ISGs (**New Fig. 4m, New Extended Data Fig. 8g, and Rebuttal Fig. 6**). In contrast, CM from R^{TT} cells did not induce ISG expression. Notably, CM of R^{TT} cells treated with COX1/2 inhibitor during media conditioning led to a recovery of ISG expression in monocytes. Collectively, these new data demonstrate that the counter-regulation of PGE₂ and IFN-I is a common feature in targeted-therapy resistant melanoma and other MAPK-driven cancers that impairs the inflammatory activation of human monocytes.

Rebuttal Fig. 6: Scheme outlining the treatment of human BLaER-1 monocytes with conditioned media (CM) from human melanoma A375 cell-line and A375 NRAS cell-line (left). Heatmap depicting differential expression of interferon-stimulated genes (ISGs) by RT-qPCR (right).

Specific questions and suggestions for improvement

R4.4: (1). In Figure 1a targeted therapy-naive (N-TT) and resistant (R-TT) BRAF-V600E mouse melanoma cells (YUMM1.7OVA) were injected into syngeneic Rag2^{-/-} mice and treated with adoptively transferred CD8⁺ OT-1 T cells. These melanoma cell lines were also originally generated in syngeneic Rag2^{-/-} mice. What was the rationale for using Rag2^{-/-} instead of immunocompetent mice? Adoptive CD8⁺ T cell therapies in Rag2^{-/-} mice fail to recapitulate the contribution of CD4⁺ helper and regulatory T cells (and in particular their interaction with Ag-presenting myeloid cells in the TME). This experimental limitation should be clearly labelled in the figure and mentioned in the main text. The results would be significantly strengthened, if they were shown with another matched pair of N-TT and R-TT melanoma cell lines that originate in and can be transplanted back in immunocompetent syngeneic mice.

We thank the reviewer for raising these important questions about our experimental setup and the origin of the YUMM1.7 model, which has been described in detail in a previous study⁴. Importantly, the YUMM1.7 N^{TT} and the matching R^{TT} model were generated in fully immunocompetent C57BL/6 mice and not in Rag2^{-/-} mice. Because the YUMM1.7 model (neither N^{TT} nor R^{TT}) did not respond to immune checkpoint inhibitors (anti-PD1/CTLA-4), likely due to a lack of sufficiently strong (neo)antigens, we engineered the YUMM1.7 N^{TT} or R^{TT} cell lines with the model antigen OVA and used adoptively transferred CD3/CD28-activated luciferase-expressing OT-1 T cells (OT-1^{Luc}), which allowed us to dynamically assess CD8⁺ T cell trafficking and expansion *in vivo* through bioluminescent imaging (BLI). We utilized Rag2^{-/-} mice for these experiments to specifically focus on the effect of adoptively transferred antigen-specific CD8⁺ T cells by intravenous injection, and we now highlight this experimental setup more strongly in our revised figures, legends, and text.

To substantiate our findings in the YUMM 1.7^{OVA} model, we utilized the YUMM3.3 model, with N^{TT} and R^{TT} matched pairs, which can be transplanted into fully immunocompetent recipient mice, as the reviewer suggested. N^{TT} tumors respond to anti-PD1/CTLA-4, whereas the R^{TT} tumors are resistant and grow unperturbed. Analogous to the YUMM 1.7^{OVA} model, YUMM3.3 R^{TT} tumors harbor a repolarized TME, and T cells do not expand or induce tumor control, and this can be modulated through genetic/pharmacological interventions targeting PGE₂/IFN-I. To further corroborate our key findings in immunocompetent models, we investigated the effects of PGE₂ depletion in an R^{TT} variant of the KRAS-driven colorectal cancer model CT-26⁴ and the immune-evasive lung cancer model KPC3. In both models, *Ptgs2* KO re-sensitized the tumors to ICB in fully immunocompetent mice, emphasizing the role of PGE₂ as a key player involved in immune evasion across different models (**Rebuttal Fig. 7a, b**). Of note, we so far have not included these new data in our revised manuscript due to space limitations but are happy to do so if the reviewer would find this important.

Rebuttal Fig. 7: **a**, Scheme outlining the genetic ablation of *Ptgs2* in CT-26 R^{TT} tumors followed by injection into BALB/c mice and treatment with anti-PD1 (ICB) every 3 days for at least 3 weeks (left). Spider plots depicting representative response to anti-PD1 ($n=5$ mice per group) (right). **b**, Scheme outlining the genetic ablation of *Ptgs2* in KPC3 tumors followed by injection into C57BL/6 mice and treatment with anti-PD1/CTLA-4 (ICB) (left). Spider plots depicting representative response to ICB ($n=7$ mice per group) (center). Survival analysis (right). Statistical analysis was performed with a Log-rank Mantel Cox test in **b**. * $p<0.05$

The bioluminescence images are unlabelled - which image belongs to which group?

We apologize for the missing image labels and have now added these to **Fig. 1b**.

R4.5: (2.) *In Figure 1b/n/p many points do not show error bars. Do the tumours regress so synchronously? Individual growth curves of mice, ideally presented as tumour diameter, would be beneficial to understand the variability of tumour size between mice in the same cohort.*

Indeed, the tumors in this model do regress very synchronously upon ACT. To make this clearer, we have replaced the average curves by spider plots depicting the individual growth curves in our revised **Fig. 1b**. Some of the error bars in former Fig. 1n, p (revised Fig. 1n, p) are indeed not clearly visible due to similar tumor sizes, which we now have clarified in the figure legend. We also made spider plots for data shown in the revised Fig. 1n, p (**Rebuttal Fig. 8**) - however, due to space restrictions, we were not planning to include these plots in our revised manuscript. Nonetheless, wherever

possible we have provided individual growth curves as spider-plots in the revised manuscript. Finally, raw data for all mouse experiments, including tumor volumes across time-points, are provided as source data.

Rebuttal Fig. 8: Individual spider plots depicting the response to the intratumoral (i.tu) injection of OT-1^{Luc} T cells in N^{TT} tumors (N^{TT} itu) and N^{TT} CL tumors ($n=6$ mice) (left), average growth curve is depicted in Fig. 1n. Individual spider plots depicting the response to the i.tu injection of T cells in R^{TT} tumors (R^{TT} itu) and R^{TT} CL tumors ($n=6$ mice) (center), average growth curves depicted in Fig. 1n. Growth curves depicting response to the i.tu injection of T cells in N^{TT} tumors and R^{TT} CL tumors ($n=6$ mice) (right), average is depicted in Fig. 1p. Arrow indicates day of ACT.

R4.6: (3.) *Figure 1c (and Fig ED1) show scRNAseq data from one single ACT-treated N-TT and R-TT tumour each. Data from at least 3-4 tumours per group need to be generated and shown to draw valid conclusions. For comparison untreated N-TT and R-TT should also be analysed. Why was the 48 hour post-ACT time-point chosen, since in Figure 1k shows stronger CD8+ T cell infiltration after 72 hr?*

We thank the reviewer for this suggestion on how to enhance the robustness of our findings. As requested, we have performed additional scRNA-seq of YUMM1.7^{OVA} N^{TT} and R^{TT} tumors after ACT (3 tumors pooled for each condition). The amended UMAP plots are shown in **New Fig. 1c**, containing the pooled cells from the first scRNA-seq analysis ($n=1$ tumor per condition) together with the second scRNA-seq analysis ($n=3$ tumors per condition), corresponding to a total of $n=16,370$ cells for N^{TT} and $n=17,733$ cells for R^{TT}. Comparisons of the first and second scRNA-seq experiments revealed an essentially identical immune landscape within the TME, providing additional robustness to this TME analysis. All scRNA-seq experiments in the manuscript involving ACT were performed 72 hours post ACT, which we have now stated more clearly in the figure legends and the text.

R4.7: (4.) *The graph in Fig. 1d is difficult to understand. What does a “normalized fraction of cells per cluster” mean? It would be more informative if relative numbers of the different immune cell subsets per condition were shown (similar to the quantitation in Extended Data 2b).*

Response: We have now replaced the stacked graphs by individual plots per cell type, showing relative numbers of cells across conditions (**revised Fig. 1d** and following quantifications).

R4.8: 5. *Figure 1f and Extended Data Fig. 2b show that T cells interacted predominantly with monocytes, which present by far the largest population of MHC II+ cells, and much less commonly with cDC1s. The authors conclude that “unexpectedly, a substantial fraction of monocytes, have an impact on intratumoral T cell expansion and penetration into tumors.” This important finding is highlighted again in Fig 3 and **should be emphasized much more strongly**. The authors could also present more details regarding the different monocyte states in each condition and discuss their importance in comparison to cDCs. This could for example be incorporated in Fig. 1d.*

We are grateful for the reviewer's enthusiasm for these previously unknown functions of inflammatory monocytes and for the encouragement to emphasize this novel and important finding of our study much more strongly. We have done so by revising and restructuring our manuscript and by adding a substantial amount of new experimental data. In our revised manuscript, we have dedicated major parts of **Figure 2** and **Figure 4** to inflammatory monocytes, providing new insights into their function, focusing on their role in CD8⁺ T cell restimulation, their ontogeny and regulation. Specifically, our revised manuscript emphasizes the following points: (1) inflammatory monocytes functionally interact with CD8⁺ T cells within the TME (**New Fig. 1g, New Fig. 2b and New Extended Data Fig. 2e, f and 3l**); (2) inflammatory monocytes display tumor antigen-derived peptides on MHC-I (**New Fig. 2f, g and Extended Data Fig. 3m**); (3) inflammatory monocytes cross-dress with cancer-derived pMHC-I complexes to restimulate CD8⁺ T cells (**New Fig. 2h-k, and New Extended Data Fig. 3n**); (4) High PGE₂ and low IFN-I in the TME impair the formation inflammatory monocytes (**New Fig. 4g-n, Extended Data Fig. 7h-j and New Extended Data Fig. 8a-i**); and (5) inflammatory macrophages are present in inflammatory hubs across human cancers (**New Fig. 2l, m and Extended Data Fig. 4a-c**). We also decided to highlight this newly discovered function of monocytes in our revised title. Altogether, we believe that our revised manuscript highlights this major finding of our study much more effectively – and we would like to thank the reviewer once more for prompting us to do so.

R4.9: 6. *In Figure 2 the authors first present evidence that resistant (R-TT) YUMM1.7 BRAF-mutant melanoma cells show upregulated prostaglandin synthesis and downregulated type I IFN signaling. Genetic knock-out of Ptgs1/2 restores responsiveness to adoptive T cell transfer therapy (Fig. 2g). Was this experiment also performed in Rag2-/-?*

Yes, for all ACT experiments performed with the YUMM1.7^{OVA} model we used *Rag2*^{-/-} recipient mice. We have clarified this in the text and in figure legends and have expanded on it in our response to **R4.4**. Beyond the YUMM1.7^{OVA} model, we have corroborated all key findings using the YUMM3.3 murine melanoma model, which is engrafted into fully immunocompetent C57BL/6 mice. Both genetic perturbations (*Ptgs1/2* KO and *IRF3/7* overexpression) and corresponding pharmacological treatments led to comparable tumor regression in both YUMM1.7 and YUMM3.3 models (**Fig. 3d-m, New Fig. 5e-i, and Extended Data Fig. 11b-f**). To expand our findings into more diverse immunocompetent models, we introduced genetic perturbations (*Ptgs2* KO and *IRF3/7* overexpression) in the KRAS-driven colorectal carcinoma model CT-26 and tested pharmacological therapy combinations in the CT-26 model, the NSCLC model KPAR²⁰, and the PDAC model EPP²¹, which independently confirmed our results from YUMM1.7^{OVA} and YUMM3.3 models (**See previous Rebuttal Fig. 7 and New Extended Data Fig. 11g-o**).

R4.10: 7. *From Fig. 2h,i, resistant (R-TT) YUMM3.3 BRAF-mutant melanoma cells are used, this time in wild-type and Rag2-/- mice. Here, genetic knock-out of Ptgs1/2 led to tumor rejection in wild-type but not Rag2-/- mice. It seems as if YUMM3.3 and YUMM1.7 BRAF-mutant melanoma cells behave fundamentally different. The authors should explain and comment on this. Of importance: these cell lines genetically differ with respect to the expression of PTEN, a known regulator of anti-melanoma immunity (Peng et al. Cancer Discovery 2016, <https://doi.org/10.1158/2159-8290.CD-15-0283>). For a recent review of cancer cell-intrinsic mechanisms driving immune evasion see: Ghorani et al. (Immunity 2023, <https://doi.org/10.1016/j.immuni.2023.09.004>). In addition, PTEN has been shown to regulate IFN responses via IRF3 (Li et al., 2016 Nat Immunol, <https://doi.org/10.1038/ni.3311>). The authors should discuss the role of PTEN expression in their models and the potential impact on the observed results.*

We thank the reviewer for bringing up these points, which gave us the impression that we did not sufficiently convey the complementarity of the YUMM1.7 and YUMM3.3 models, as well as important differences in their biology and their experimental use. We apologize for this, and first would like to briefly summarize key features of these models that are relevant to our study: The YUMM1.7 and YUMM3.3 models indeed differ in their genetic makeup and their baseline immunogenicity, as is the case in human cancers. The YUMM1.7 model is much less immunogenic in C57BL/6 recipient mice, which is illustrated by its unresponsiveness to ICB therapy. Therefore, to utilize this model for investigating cancer-immune cell interactions, we engineered N^{TT} and R^{TT} variants expressing an Ova antigen (YUMM1.7^{OVA}) and modelled T-cell interactions through ACT of Ova-specific OT-1 T cells into Rag2^{-/-} mice harboring YUMM1.7^{OVA} tumors. By contrast, the YUMM3.3 N^{TT} model is more immunogenic in C57BL/6 mice and responsive to ICB therapy, while the R^{TT} variant has lost this trait. As already discussed in our response to R4.4, we took advantage of this baseline immunogenicity of the YUMM3.3 model throughout our study to conduct experiments in immunocompetent mice. In the experiments presented in former Fig. 2h,i, *Ptgs2* KO led to spontaneous rejection of YUMM3.3 R^{TT} tumors in immunocompetent C57BL/6 recipient mice, which suggested an enhanced T-cell mediated tumor control. To further test this, we injected YUMM3.3 R^{TT} *Ptgs2* KO cells into Rag2^{-/-} mice, where these tumors grew out. These results complement findings in the YUMM1.7^{OVA} model, where *Ptgs1/2* KO YUMM1.7^{OVA} R^{TT} tumors are re-sensitized to ACT therapy while growing unperturbed without ACT (Fig. 3e and Extended Data Fig. 5b). Thus, independent experimental models and approaches lead to analogous findings. To clarify this point, we have replaced former Fig. 2i with a **New Fig. 3g**, in which we only show the tumor growth in C57BL/6 mice, while moving the control experiments using Rag2^{-/-} mice to **Extended Data Fig. 5g**.

The presence of a PTEN deletion in the YUMM1.7 model may indeed underly differences in the baseline immunogenicity between both models. We thank the reviewer for pointing us to the interesting study by Li et al. showing that PTEN loss, besides promoting tumor growth, also dampens IFN-I signaling by diminishing IRF3 nuclear translocation. If we extrapolate this to our YUMM1.7 model, which harbors a PTEN deletion, we would expect lower *Irf1* transcription and, thus, lower IFN- β production compared to YUMM3.3. However, this is not the case in our models (**Extended Data Fig. 5h and k**), indicating that the low immunogenicity of YUMM1.7 cannot be explained by this PTEN-dependent mechanism.

Finally, although YUMM1.7 and YUMM3.3 indeed differ in their genetic makeup and their immunogenicity, we would like to point out that we did not base any conclusions on comparisons between these models. Instead, all our experiments and derived conclusions are based on comparisons between N^{TT} and R^{TT} phenotypes within the *same* model. Moreover, we would like to emphasize that N^{TT} and R^{TT} derivatives of the YUMM1.7^{OVA} and YUMM3.3 models share critical features such as their sensitivity to immunotherapy (ACT or ICB) and targeted therapy (BRAFi) in N^{TT} variants, and their cross-resistance between targeted and immunotherapy in the R^{TT} variants. Furthermore, the N^{TT} and R^{TT} variants of both models share characteristics in their TME, for example, the absence of inflammatory monocytes in R^{TT} tumors, as well as shared molecular programs, including the counterregulation of PGE₂ and IFN-I in both R^{TT} variants compared to their matching N^{TT} counterparts (Fig. 1c and Extended Data Fig. 1k). In line with these observations, genetic and pharmacological depletion of PGE₂ or reinstatement of an IFN-I program led to immune-mediated clearance of both R^{TT} models, further highlighting similarities and the complementarity of both models.

R4.11: 8. In Fig. 2j it is not clear which cell lines were used to assess the production of IFN β . Both the production of PGE₂ and the secretion of type I IFNs should be shown for all cell lines used.

We apologize for this missing information. We now show the PGE₂ and IFN- γ levels for all mouse and human cell-lines in the **New Fig. 3c, h, n** and **Extended Data Fig. 5a, d, f, h, k, p-r**.

R4.12: Figure 2p: What does the 6/6 refer to, since one line does not reach a volume of 0 mm³? Why are only four lines shown?

We thank the reviewer for pointing out this apparent discrepancy, which stems from the fact that some tumors are very efficiently controlled by CD8⁺ T cells. To enhance clarity, we indicated the number of rejected (below 20mm³) over the number of injected tumors in our figures. In former Fig. 2p / **New Fig. 3i**, six out of six (6/6) tumors were controlled upon IRF3/7 overexpression, and even though all six lines are shown, two are poorly resolved due to very efficient tumor control. In addition to the figure, we provide all tumor volumes for each tumor and timepoint in our source data.

R4.13: 10. Figure 3a (and Extended Data Fig. 4a -d) show scRNAseq data for CD45+ cells isolated from YUMM1.7OVA RTT-CTRL, RTT-Ptgs1/2 KO and RTT-IRF3/7 overexpressing tumours 72h after ACT treatment. The data are compared to the results from YUMM1.7OVA RTT 48h after ACT treatment (from Fig.1). What is the immune cell composition of untreated Rtt Ptgs1/2 KO and Rtt IRF3/7 tumours? Data from the same time point and a comparison with untreated tumours should be shown. Moreover, the figure legend says that n=3 tumours were pooled from RTT tumours, however the number of cells is like the single NTT tumour. Are the UMAP plots representative of one tumour from each group? Again, the graph in Fig. 3b (now labelled as “percent of cells per cluster across conditions”) is difficult to understand. It could be replaced by relative numbers of the different immune cell subsets per condition (similar to Fig. 1d).

We thank the reviewer for raising these questions and excellent suggestions, which allowed us to clarify the following points (for information on chosen timepoints and tumor numbers, please also see R4.6):

1. **Time points:** This seems to be a misunderstanding – we always chose 72 hours post ACT for our scRNA-seq analysis from N^{TT} and R^{TT} YUMM1.7^{OVA} tumors shown in the **new Fig. 1c** and **Fig. 4a**. We have highlighted this in the revised figures, figure legends, and methods.
2. **Number of tumors and cells analyzed:** In the revised manuscript, we performed additional scRNA-seq of N^{TT} tumors to now depict the results of 3-4 tumors per condition (**New Fig. 1c** and **New Fig. 4a**). In brief, we pooled 3 tumors per condition prior to the isolation of CD45⁺ cells from the tumors. This pooled cell suspension ($n=3-4$ tumors) was submitted for scRNA-seq library preparation. In all conditions, up to 20,000 cells were sequenced (the exact numbers are depicted above the UMAPs). The single cell landscape, depicted in the UMAPs, is thus representative of CD45⁺ immune cells from 3-4 independent tumors per condition. We have described this in the figure legends and methods.
3. **Data visualization:** We thank the reviewer and agree that this depiction of cellular changes in the TME was not very intuitive. We have now replaced the stacked graphs by individual plots per cell type, showing relative numbers of cells across conditions (**New Fig. 1d** and **subsequent plots alike**).
4. **Flow cytometry validations and TME status prior to ACT:** To investigate the immune cell composition of untreated R^{TT} Ptgs2 KO and R^{TT} IRF3/7 tumors, we harvested tumors at day 10 post-injection and analyzed the TME through multicolor flow cytometry. In the same experiment, we included pre- and post-ACT tumors. Tumors of both groups were harvested at day 10 post tumor injection (i.e., 72 hours post ACT), in order to allow a direct cross-

comparison to the previous scRNA-seq results. The flow cytometry-based characterization pre and post ACT across all conditions is now depicted in **New Extended Data Fig. 6b, c**, together with the new scRNA-seq data of untreated and ACT-treated N^{TT} and R^{TT} tumors (**New Fig. 1c**, **New Extended Data Fig. 1e**). We provide an overview of the immune cell composition of tumors pre and post ACT in our response below (R4.14).

R4.14: 11. Figure 3c: Activated CD8+ T cells produce IFN γ in the TME. Has the role and contribution of IFN γ on the myeloid compartment been assessed?

We thank the reviewer for this thoughtful comment (also relating to point R4.13). To address this question, we now have assessed the differential contributions of IFN-I and IFN- γ on the myeloid compartment by leveraging new scRNAseq data from the TME before and after ACT (**New Extended Data Fig. 7f**) and selective antibody-mediated inhibition of IFN-I or IFN- γ signaling (**New Fig. 4f and New Extended Data Fig. 7f, g**). We found that before an active CD8⁺ T cell response (pre-ACT), the generation of inflammatory monocytes is dictated by cancer cell-derived IFN-I, which is sufficient to induce an inflammatory state in monocytes that facilitates the primary expansion of CD8⁺ T cells in the tumor. Upon ACT, CD8⁺ T cell-derived IFN- γ further diminishes monocyte differentiation into TAMs and bolsters the inflammatory state of monocytes. By contrast, the TME of R^{TT} tumors is not conducive to CD8⁺ T cell expansion and, therefore, never exposed to sufficient IFN- γ levels from T cells, maintaining an immune-evasive TME with low abundance of inflammatory monocytes. These new data are shown in **New Fig. 4f and New Extended Data Fig. 7f, g** and described in the Results section “Type I and type II IFN instruct the inflammatory state”.

R4.15: 12. Interestingly, the growth of melanomas could still be controlled by adoptively transferred T cells in Rag2-/- Batf3-/- mice that lack cDC1s (Figure 3h,i). Are inflammatory monocytes able to compensate the function of cDC1? Individual growth curves should be shown.

Indeed, tumor growth could still be controlled by ACT in *Batf3*^{-/-} mice lacking cDC1s and T cell expansion still occurred even in *Zbtb46*-DTR mice, in which all DC subsets are depleted (**New Fig. 2a, b, 3m and New Extended Data Fig. 3f-i**). This is a very surprising and important finding of our study that helps to establish inflammatory monocytes as important players in intratumoral T-cell restimulation. While we do not dispute well-established functions of cDC1s in this process, our finding that inflammatory monocytes can fully compensate for the loss of DCs challenges the current model that intratumoral T-cell re-stimulation is mainly executed by cDC1s. Instead, we favor a model where cDC1s and inflammatory monocytes play redundant roles in the cancer immunity subcycle. To enhance clarity, we now display individual growth curves in **New Figure 3m**.

R4.16: 13. The data presented in Extended Data Fig. 3h-k suggest that PGE₂ upregulation and type I IFN downregulation result from oncogenic MAPK signaling. Is this specific to the two mouse cell lines used or might this be a more general phenomenon? The authors should investigate this also in human cell lines, taking into account that loss of PTEN also impacts on type I IFN regulation.

We thank the reviewer for this excellent suggestion. As requested, we now have analyzed PGE₂ production in three different BRAF^{V600E}-driven human driven melanoma cell lines (A375, M249, LOX) and one lung KRAS^{G12C}-driven NSCLC cell-line (NCI-H358), of which only one cell line (M249) carries a PTEN deletion. For all four cell lines, we derived matched pairs of parental (N^{TT}) and resistant (R^{TT}) sublines, which were made resistant to BRAFi or KRASi²². In all four models, we consistently observed an elevated production of PGE₂ and lower levels of IFN- β in the R^{TT} derivatives (**New Extended Data Fig. 5r**). In accordance with the increased PGE₂ production, A375 R^{TT} exhibited increased COX2

protein levels compared to N^{TT}, and upon re-exposure to BRAFi, COX2 protein levels were downregulated, indicating that even in a resistant cell line, the MAPK pathway still controls some of its targets. In addition, overexpression of NRAS in A375 cells also increased COX2 protein levels (**New Extended Data Fig. 5s**). Collectively, these data suggest that counter-regulation of the COX2/IFN-I axis is a conserved function of oncogenic MAPK signaling in human cancers.

R4.17: 14. The correlation shown in Fig. 4b-e between the extent of CD8 T cell infiltration in human melanomas and the expression of “TME-COX signature” (containing also Stat1, Isg15, Cxcl9, Cxcl10, and several HLA molecule genes) or the “TMIS signature” (containing also Isg15, Irf7, Ifit3, Ifit1b) in bulk RNAseq data is expected. Numerous studies have shown that key IFN-induced genes correlate with the presence of CD8 T cells in tumors (and with the response to immune checkpoint blockade immunotherapy). This data does not provide any support for the conclusion of the Figure (PGE₂ impairs the acquisition of an interferon-induced inflammatory state in myeloid cells).

We thank the reviewer for bringing up this very thoughtful justified concern. In our revised manuscript, we have removed the TMIS signature and instead directly correlated the TME changes in our preclinical models with response to immunotherapy in a clinical data set. We show that TME-COX and TME-IRF3/7 signatures, derived from differentially expressed genes in the TME of R^{TT} *Ptgs1/2* KO and R^{TT} IRF3/7 compared to R^{TT} CTRL, are associated with the presence of CD8⁺ T cell in human tumors and identify patients who respond to ICB (**New Extended Data Fig. 7a-d**). In addition, in our revised manuscript, we also took advantage of a recently published scRNA-seq dataset of 13 patients with metastatic melanoma who received tumor infiltrating lymphocyte (TIL)-ACT therapy after progressing on ICB treatment⁶. We found that our TME-COX and TME-IRF3/7 signatures scored higher in responder compared to non-responder patients, indicating that changes we observe in the TME immune composition upon PGE₂ depletion and IFN restoration are also predictive for the response to TIL-based immunotherapy (**New Fig. 4c and Extended Data Fig. 7e**).

To improve the presentation of these data and avoid over-statements, we have substantially revised this figure and changed its title to: “Cancer-derived PGE₂ and IFN-I determine myeloid cell abundance and their functional state in the TME” (**New Fig. 4**). We also conducted additional experiments to further support our conclusion that PGE₂ impairs the acquisition of an interferon-induced inflammatory state in myeloid cells. In our first submission we showed that the addition of increasing amounts of PGE₂ in presence of IFN-I decreases the induction of ISGs, indicative of the inflammatory state, in myeloid cells (**New Fig. 4n**). In our revised manuscript, we now further demonstrate that conditioned media (CM) of R^{TT} cells fails to induce ISGs in DCs and monocytes, both human and mouse, which can be reverted by conditioning the media in the presence of a COX1/2i (**New Fig. 4k-m, New Extended Data Fig. 8g**). Furthermore, we have also shown that in R^{TT} *Ptgs1/2* KO tumors inflammatory myeloid populations are recovered without increasing IFN-I levels in the TME (**Extended Data Fig. 5o, p**), highlighting the suppressive effect of PGE₂. We hope that these additional experimental data, re-analyses, and revisions address the referee's concern.

R4.18: 15. An important experiment was performed with YUMM1.7OVA RTT melanoma cells in mice where CD11c+ immune cells lack the prostaglandin receptors EP2 and EP4. Flow cytometry shows increased infiltration with T cells as well as with inflammatory DC and monocytes (Fig. 4f-h). Did this restore tumor growth control?

Indeed, increased CD8⁺ T cell infiltration in this model resulted in tumor growth control (**former Fig. 4f-h, now Fig. 4i, and Extended Data Fig. 8a**).

R4.19: 16. *The experiment with YUMM3.3.RTT melanoma cells shown in Fig. 4i does not support any conclusion about the role of prostaglandin receptors on myeloid cells for T cell-mediated anti-tumor immunity, because the group of “CTRL” mice did not receive CD8 T cell depletion.*

This concern might be based on a misunderstanding - and we apologize for not clearly describing our reasoning and setup for these experiments. In this experiment (former Fig. 4i, now **Extended Data Fig. 8a**), CTRL mice refer to CD11c^{Cre} (*Itgax-Cre*) mice, in which cross-resistant YUMM3.3 R^{TT} tumors grow out comparably to WT mice (as shown e.g. in **Extended Data Fig. 1j**). CD8⁺ T cell depletion in this context would, if anything, promote even faster tumor growth. In stark contrast, in CD11c^{Cre}/*Itgax-Cre*)/*Ptger2*^{-/-}/*Ptger4*^{fl/fl} recipient mice, YUMM3.3 R^{TT} tumors were rejected, even without additional ICB, illustrating the potency of PGE₂-driven immune evasion. To demonstrate that these effects were T cell mediated, we depleted CD8⁺ T cells in these mice via anti-CD8 antibody treatment, which reversed the phenotype and led to tumor outgrowth. Altogether, these experiments confirmed that also in the YUMM3.3 R^{TT} model, the depletion of prostaglandin receptors on myeloid cells is sufficient to induce tumor control mediated by CD8⁺ T cells. We hope this explanation clarifies the reviewers point, and we have more clearly described this experimental setup in our revised manuscript.

R4.20: 17. *Figure 4f / 5a: It is unclear if this is intravenous or intratumoural CD8+ T cell transfer. Please clarify.*

We apologize for the confusion. To clarify, our standard procedure for ACT involves the intravenous injection of pre-activated antigen-specific CD8⁺ T cells. The only exceptions are **Fig. 1i-p and Extended Data Fig. 2t, u**, where we injected the same amount of CD8⁺ T cells intratumorally to investigate the expansion of CD8⁺ T cell within the TME of N^{TT} and R^{TT} tumors. We have updated the text and figure legends accordingly and amended our schemes to more clearly illustrate this experimental setup.

R4.21: *Again, the graph in Fig. 5c (labelled as “percent of cells per cluster across conditions”) could be replaced by relative numbers of the different immune cell subsets per condition (similar to Fig. 1d and Fig. 3b).*

We thank for reviewer for this suggestion, which we have implemented.

R4.22. *The adoptive T cell transfer therapies in Fig. 5 are most likely also performed in Rag2^{-/-} mice. This should be specifically stated. Were T cells injected intravenously in 5a,d,h? These results are impressive. Individual growth curves for the experiment depicted in Fig. 5k should be presented (similar to Fig. 5g).*

We are glad that the referee shares our excitement about these results and apologize for the missing information. Indeed, these experiments were performed using intravenous ACT in *Rag2*^{-/-} mice, and we have now added this information in the text and our figure schematics. We also fully agree with the reviewer's comment that displaying individual growth curves adds value to the findings and we have now added them in **New Fig. 5g, h and New Extended Data Fig. 10e, j and 11c-o**.

Once more, we would like to thank the reviewer for this very thorough and thoughtful evaluation of our manuscript. The raised comments, concerns and excellent suggestions were extremely helpful to better highlight and further expand our key findings, and thereby improve our manuscript. We hope the reviewer will be satisfied with our revision.

References

1. Duong, E. *et al.* Type I interferon activates MHC class I-dressed CD11b+ conventional dendritic cells to promote protective anti-tumor CD8+ T cell immunity. *Immunity* **55**, 308–323.e9 (2022).
2. MacNabb, B. W. *et al.* Dendritic cells can prime anti-tumor CD8+ T cell responses through major histocompatibility complex cross-dressing. *Immunity* **55**, 982–997.e8 (2022).
3. Wakim, L. M. & Bevan, M. J. Cross-dressed dendritic cells drive memory CD8+ T-cell activation after viral infection. *Nature* **471**, 629–632 (2011).
4. Haas, L. *et al.* Acquired resistance to anti-MAPK targeted therapy confers an immune-evasive tumor microenvironment and cross-resistance to immunotherapy in melanoma. *Nature Cancer* **2021 2:7 2**, 693–708 (2021).
5. Cheng, S. *et al.* A pan-cancer single-cell transcriptional atlas of tumor infiltrating myeloid cells. *Cell* **184**, 792–809.e23 (2021).
6. Barras, D. *et al.* Response to tumor-infiltrating lymphocyte adoptive therapy is associated with preexisting CD8+ T-myeloid cell networks in melanoma. *Sci Immunol* **9**, eadg7995 (2024).
7. Gide, T. N. *et al.* Distinct Immune Cell Populations Define Response to Anti-PD-1 Monotherapy and Anti-PD-1/Anti-CTLA-4 Combined Therapy. *Cancer Cell* **35**, 238–255.e6 (2019).
8. Gaidt, M. M., Rapino, F., Graf, T. & Hornung, V. Modeling Primary Human Monocytes with the Trans-Differentiation Cell Line BLaER1. *Methods Mol. Biol.* **1714**, 57–66 (2018).
9. Siddiqui, I. *et al.* Intratumoral Tcf1+PD-1+CD8+ T Cells with Stem-like Properties Promote Tumor Control in Response to Vaccination and Checkpoint Blockade Immunotherapy. *Immunity* **50**, 195–211.e10 (2019).
10. Kurtulus, S. *et al.* Checkpoint Blockade Immunotherapy Induces Dynamic Changes in PD-1-CD8+ Tumor-Infiltrating T Cells. *Immunity* **50**, 181–194.e6 (2019).
11. Miller, B. C. *et al.* Subsets of exhausted CD8+ T cells differentially mediate tumor control and respond to checkpoint blockade. *Nat. Immunol.* **20**, 326–336 (2019).
12. Krishna, S. *et al.* Stem-like CD8 T cells mediate response of adoptive cell immunotherapy against human cancer. *Science* **370**, 1328–1334 (2020).
13. Di Pilato, M. *et al.* CXCR6 positions cytotoxic T cells to receive critical survival signals in the tumor microenvironment. *Cell* **184**, 4512–4530.e22 (2021).
14. Lacher, S. B. *et al.* PGE2 limits effector expansion of tumour-infiltrating stem-like CD8+ T cells. *Nature* 1–9 (2024).
15. Wang, D. Y. *et al.* The Impact of Nonsteroidal Anti-Inflammatory Drugs, Beta Blockers, and Metformin on the Efficacy of Anti-PD-1 Therapy in Advanced Melanoma. *Oncologist* **25**, e602–e605 (2020).
16. Wang, S.-J. *et al.* Effect of cyclo-oxygenase inhibitor use during checkpoint blockade immunotherapy in patients with metastatic melanoma and non-small cell lung cancer. *J Immunother Cancer* **8**, (2020).
17. Ebert, P. J. R. *et al.* MAP Kinase Inhibition Promotes T Cell and Anti-tumor Activity in Combination with PD-L1 Checkpoint Blockade. *Immunity* **44**, 609–621 (2016).
18. Norgard, R. J. *et al.* Reshaping the Tumor Microenvironment of KRASG12D Pancreatic Ductal Adenocarcinoma with combined SOS1 and MEK Inhibition for Improved Immunotherapy Response. *Cancer Res Commun* (2024) doi:10.1158/2767-9764.CRC-24-0172.
19. Bayerl, F. *et al.* Tumor-derived prostaglandin E2 programs cDC1 dysfunction to impair intratumoral orchestration of anti-cancer T cell responses. *Immunity* **56**, 1341–1358.e11 (2023).
20. Boumelha, J. *et al.* An Immunogenic Model of KRAS-Mutant Lung Cancer Enables Evaluation of Targeted Therapy and Immunotherapy Combinations. *Cancer Res.* **82**, 3435–3448 (2022).
21. Rathert, P. *et al.* Transcriptional plasticity promotes primary and acquired resistance to BET inhibition. *Nature* **525**, 543–547 (2015).
22. Obenauf, A. C. *et al.* Therapy-induced tumour secretomes promote resistance and tumour progression. *Nature* **520**, 368–372 (2015).

Point-by-point reply to the reviewer's comments

(Reviewer comments in *blue italics*, **emphasis in bold our own**, our responses in black)

Summary

We would like to thank referees once more for their thorough evaluation of our work and their thoughtful comments. We are pleased to receive such unanimously positive and enthusiastic evaluations of our revised manuscript and gratified to see that our efforts have been so greatly appreciated.

We are excited to resubmit a revised version of our manuscript addressing the remaining points brought up by the reviewers and provide a detailed point by point response below. To align with the formatting guidelines of *Nature*, we have reduced the main figures to the key data, moved additional data to the Extended Data Figures, and shortened the manuscript and figure legends, while maintaining clarity and providing sufficient context.

Reviewer #1:

The manuscript previously entitled "Counter-regulated PGE2 and IFN-I create an immune evasive tumor microenvironment" has now been extensively revised and retitled "Cancer cells impair monocyte-mediated intratumoral T cell stimulation to evade immunity."

*• In the single cell analysis the inflammatory monocytes seem to have differentiated into inhibitory macrophages therefore burden of proof the monocytes are a/monocytes and not AXL DC2 and b/ have not simply become inhibitory macrophages as we enter the story. The authors have chosen in the revision to expand the monocyte story which has added **novelty and a more clear direction to the narrative. The level and extent of revision is impressive and the interrogation of this work is to be commended and relentlessly thorough.***

We are delighted that the reviewer acknowledges the comprehensive nature of the revisions and recognizes the positive impact in enhancing the novelty and clarity of the narrative.

We agree that that the point about the identity and differentiation of inflammatory monocytes is important, and have addressed it by showing that (a) inflammatory monocytes and AXL⁺ DCs are two distinct subpopulations, which differ in their expression of monocyte (*Fcgr1*, *Cd14*, *Ly6c*, *Ly6a*) and dendritic cell (*Cd24*, *Zbtb46*, *Flt3*) markers (**Extended Data Fig. 1e**), and (b) monocytes that are exposed to the R^{TT} or R^{TT} TME, differentiate to TAMs or inflammatory monocytes with immune-stimulatory capabilities, respectively (**Fig. 4f and Extended Data Fig. 7f**). These and our other data let us therefore to conclude that inflammatory monocytes represent a distinct subpopulation of immune cells with a previously unrecognized function in anti-tumor immunity.

• The differentiation towards inhibitory macs is of course still of interest but beyond the scope of the current work.

We agree that the differentiation process towards inhibitory macrophages is highly interesting, and we agree that further investigation extends beyond the scope of this study.

• Extensive effort in the revision to really tackle the antigen presenting direct vs cross-dressed presentation is evident.

and

• The authors are highly responsive to all reviewers.

Thank you!

• Minor: Extended data Fig3f- DC depletion after a single dose of DT lasts 48 hrs therefore the legend should include some mention of length of time of DC depletion

We thank the reviewer for spotting this and have added this information to the legend of Extended Data Figure 3 and in the methods section (lines 3749 and 2736 in the manuscript version with tracked changes).

• The authors have done a great and thorough job with the revision.

We sincerely thank the reviewer for the positive evaluation.

• However as pointed out in the last round of revision there was a lack of novelty in the original framing around IFN and now the novelty of this manuscript, which is very different from their prior work really rests on the importance of this inflammatory monocyte derived cross-dressing mechanism. It's therefore somewhat unclear why a simple comparison of the level of OT1 priming in vivo in the absence of DC vs these monocytes was not conducted side by side to test the relative contribution as likely both mechanisms are important but many compensatory mechanisms for antigen presentation occur in the absence of DC1. They would implant tumors into the relative CCR2 (or ly6A) ko vs BATf3 ko mice and measure tumor growth side by side +/- addition of CFSE ;labelled OT1 T cells at Day 9, or for acute loss could compare an inflammatory monocyte specific DTR vs DC specific DTR as BM chimera for parity. Very sorry if this was missed as there is a lot of key data in the extended but this is the final experiment really needed to show the relative importance of this mechanism side by side as the push back from some of the DC1 field may be significant.

It is curious the authors who are superbly wielding use of DTRs and IFN antibody depletion etc did not take a more direct approach of loss of the monocytes versus DC to see how this normalizes the RTT and NTT difference. I am concerned a little that this was not directly addressed as the authors themselves state the impact of this revision rests with monocytes as they have published on the IFN PGE axis before.

We agree that the relative contribution of cDC1s vs. inflammatory monocytes to different components of anti-tumor immunity is interesting, and we would like to clarify that our study does not aim to challenge or downplay the well-established role of cDC1s in intratumoral T cell stimulation. We fully agree with the reviewer that both populations are important. Instead, our work reveals that intratumoral T cell restimulation is not an exclusive function of cDC1s but is also mediated by tumor-associated inflammatory monocytes, which (1) act as a second, previously unknown key player in intratumoral T cell restimulation, adding robustness to this critical step of the cancer immunity cycle; (2) execute this function through an intriguing mechanism of antigen presentation (cross-dressing); and (3) are a key target of counter-regulated PGE₂ and IFN-I signaling through which cancer cells shape an immune-evasive environment.

We agree with the reviewer that a direct functional side-by-side comparison of the contribution of inflammatory monocyte versus dendritic cell-mediated restimulation of CD8⁺ T cells would

be of interest. However, regarding the genetic depletion experiment suggested by the reviewer, to our knowledge, a selective genetic ablation of inflammatory monocytes is currently not feasible. Existing conditional KO/DTR mice (such as CCR2 KO/DTR mice) would broadly deplete monocytes/macrophages and can affect other myeloid cells, including the DC lineage (e.g., PMID: 37523265, PMID: 19917501, PMID: 26452628), likely misrepresenting the role of inflammatory monocytes in T cell restimulation. Therefore, these established models are not suitable for dissecting the role of inflammatory monocytes. We have conceived possible ways of how to genetically selectively ablate inflammatory monocytes (e.g., by using Ly6c-Cre/Stat1^{fl/fl} Rag2^{-/-} mice, or by generating a Ly6a-DTR mouse as proposed by the reviewer), but this would require the generation of new transgenic mouse lines that would have to be characterized prior to their experimental use, which would take many months without a guarantee for success. We are actively working to overcome these challenges and hope to report on these efforts in the future.

In our study, we have investigated the relative contributions of cDC1s and inflammatory monocytes side-by-side in several ways:

(1) Through selective ablation of cDC1s or all cDCs (using *Batf3* KO mice or *Zbtb46*-DTR mice, respectively), we demonstrate that inflammatory monocytes in N^{TT} tumors can compensate for the absence of cDC1s/cDCs, leading to expansion of primed CD8⁺ T cells and tumor control (**Fig. 2a, b, g, 3h and Extended Data Figure 3d-i**), which is a surprising key finding of our study that establishes important functions of inflammatory monocytes in intratumoral T cell stimulation.

(2) Conversely, we find that CD8⁺ T cell restimulation and expansion also remain intact upon knocking out β 2-microglobulin (B2m KO) in N^{TT} cancer cells, which ablates the ability of inflammatory monocytes to present tumor antigens through MHC-I cross-dressing (**Fig. 2e-g and Extended Data Fig. 3n**). This retained T cell activity reflects the capacity of cDC1s to restimulate CD8⁺ T cells through classical antigen presentation.

(3) Accordingly, only if we ablate both, cDC1s and cross-dressed inflammatory monocytes, by engrafting N^{TT} *B2m* KO tumors in *Batf3* KO mice, intratumoral T cell restimulation and expansion is lost (**Fig. 2g**).

(4) In complementary *in vitro* studies, we further demonstrate that FACS-purified inflammatory monocytes isolated from N^{TT} tumors activate naive CD8 T cells (measured by CFSE dilution) (**Fig. 2e and Extended Data Fig. 3o**), further indicating that T cell restimulation by inflammatory monocytes is not only a compensatory mechanism that arises in the absence of cDC1s.

While these experiments were mainly conducted to investigate cross-dressing, they also demonstrate that both classical antigen-presenting cDC1s and cross-dressed inflammatory monocytes have important contributions to intratumoral T cell restimulation. These conclusions are now more clearly stated in the second revision of our manuscript and thoroughly discussed (lines 101, 443-454, 929, 969-973).

We believe that our discovery of this complementary mechanism of intratumoral T cell stimulation will raise a lot of interest in the cDC1 field and beyond. We are grateful to the reviewer for bringing up this point, as it gave us an opportunity to better highlight our data addressing the roles of cDC1s and inflammatory monocytes in anti-tumor immunity.

Otherwise this is truly a wonderful revision and the authors should be commended on their efforts.

Thank you very much!

Reviewer #2:

In the revised manuscript by Elewaut et al entitled “Cancer cells impair monocyte-mediated intratumoral T cell stimulation to evade immunity”, the authors build upon their previous work and prior literature demonstrating that acquired resistance to targeted therapy can impair response to subsequent immunotherapy. The authors use elegant animal models and experimental designs complemented with validating clinical studies to explore the mechanisms underlying T cell restimulation in the tumor microenvironment and identify an inflammatory state in monocytes that facilitates T cell expansion in a cDC-independent manner. The authors further demonstrate that cancer cells produce PGE2, which impacts the myeloid cell response to IFN- γ and their inflammatory state in the tumor microenvironment and impairs T cell stimulation. Lastly, the authors demonstrate an association between NSAID co-medication and response to ICB in melanoma and NSCLC patients and further explore the role of pharmacological modulation of PGE2 in changing tumor immunity and response to immunotherapy in a number of mouse models.

Overall, this is an outstanding body of work that provides novel insights into the mechanisms underlying tumor immune evasion and as such, holds great promise for advancing the field with clear translational applications. In this revision, the authors have done a tremendous job addressing concerns and strengthening the manuscript. The inclusion of additional pre-clinical and clinical studies in this revision has significantly bolstered the study. The experimental designs and findings appropriately support the conclusions made.

In essence, the authors addressed all reviewer comments in an elegant and thoughtful manner. These findings will be of great interest to the broad readership of Nature.

We are truly excited about the enthusiastic reception of our findings and the potential impact on the understanding of tumor immune evasion and the advancement of cancer immunotherapy. We are grateful for the reviewer’s insightful comments and suggestions, which have greatly improved our manuscript.

Reviewer #3:

*I have read the revised paper and the responses to my comments and those of the other reviewers. In aggregate, this paper expands our understanding of the “triangular interplay” between cancer cells, myeloid cells, and T cells in the TME and brings forward novel insights. **It is superbly executed, and a lot of data has been added, making it a very comprehensive analysis and a huge, solid body of work.** Some of the insight may have implication for treatment of patients on ICIs and combinations and how we use COX inhibitors.*

With respect to my comments 3.1 – 3.6, they have been addressed to my satisfaction.

We appreciate the referee’s thoughtful review and enthusiastic words regarding our revised manuscript. We are pleased to have addressed the reviewers' comments to their satisfaction.

Reviewer #4:

*In their revised manuscript now entitled “Cancer cells impair monocyte-mediated intratumoral T cell stimulation to evade immunity”, Anna Obenauf and colleagues now provide a large amount of new experimental data that allowed them to conceptually advance their insights into the molecular and cellular mechanisms underlying the cross-resistance of BRAF-mutated melanoma cells to targeted MAPK-inhibition and to T cell-directed immunotherapy. **I was very impressed by the thorough implementation of all the referee’s suggestions that have greatly improved the manuscript.***

*By focussing on the very original and important observation that inflammatory monocytes / macrophages play an important role in supporting T cell effector functions in tumour tissues and – specifically - by showing that peptide-MHC-I “cross-dressing” plays an important role, **the authors now present a conceptual advance in the field. This clearly demarcates their work from their previous related manuscripts. The new data and analyses that connect their experimental work to the clinical care of patients is impressive and greatly expands the relevance of their work.***

Thank you very much! We truly appreciate your feedback and comments.

The authors have addressed most of my comments very thoroughly. However, one aspect remains. I still did not understand what the graphs that relate to their scRNAseq data (now Figs. 1d, 4b, 5c, ED Figs. 1f, 1l, 10g) are showing. They now call this “relative frequency (across conditions)”. In their former manuscript these graphs were called “normalized fraction of cells per cluster”. The authors need to explain how they calculate and what they intend to convey with these graphs. I could not find any information in the methods.

We apologize for not sufficiently explaining the relative frequency plots of the scRNA-seq data. Please find below a detailed explanation of how these plots were generated and what they intend to convey.

Compositional analyses address conditional changes not in the gene expression profile of a cell but instead in the relative abundance of different cell types in the form of compositional data. The relative frequency bar plots depict the changes in the relative abundance of a cell type across different experimental conditions. The calculation process involves the following steps: Normalized Abundance Calculation (1): For each condition, we calculated the normalized abundance of a specific cell type by comparing the absolute number of the cell type to the absolute number of all cells in the same condition. This normalization accounts for differences in total number of cells captured between conditions. Relative Cell Abundance calculation (2): We then calculated the relative cell abundance of the cell type in all conditions of the experiment. This was done by comparing the normalized abundance of the cell type to the sum of normalized abundances of the same cell type across conditions of the experiment. This step produces values between 0 and 1 for each condition for each cell type, with the sum of these values across all conditions of the experiment equaling 1 for each cell type (3).

As a specific example, the following equations show how the relative abundance of inflammatory monocytes (Infl.mono) was calculated across the CD45⁺ compartment of an experiment that had four conditions: N^{TT}, R^{TT}, R^{TT}-PTGSKO and R^{TT}-IRF3/7.

$$(1) \frac{\text{Infl.mono}_{[NTT]}}{\text{Total cells}_{[NTT]}} = \text{Norm. Infl. mono abundance in NTT} (N\text{Inflmono}_{NTT})$$

$$(2) \frac{N\text{Inflmono}_{NTT}}{N\text{Inflmono}_{NTT} + N\text{Inflmono}_{RTT} + N\text{Inflmono}_{PTGSKO} + N\text{Inflmono}_{IRF}} = \text{Relative abundance InflMono in NTT} (R\text{Inflmono}_{NTT})$$

$$(3) R\text{Inflmono}_{NTT} + R\text{Inflmono}_{RTT} + R\text{Inflmono}_{PTGSKO} + R\text{Inflmono}_{IRF} = 1$$

These steps ensure that the bar plots convey the proportionate representation of each cell type (i.e. cluster in the scRNA-seq UMAP) relative to its total across the different experimental conditions. We hope that these explanations clarify how the plots were generated and what they intend to convey. We have modified our methods section accordingly (line 2987-2996 in the manuscript version with tracked changes).

The authors now explain why they mostly used Rag knockout mice with a focus on effector functions of CD8 T cells and their interaction with inflammatory monocytes / macrophages. The cooperation of T cells with monocytes/macrophages and the role of interferons have also been reported by recent papers. The concept of “monocyte-mediated intratumoral T cell stimulation” has also been described by other recent papers that show how this cooperation expands the mechanisms of tumour cell killing beyond classical cytotoxicity. For example: Kruse et al. Nature 2023 (PMID: 37316667; <https://doi.org/10.1038/s41586-023-06199-x>), van Elsas et al Cancer Cell 2024 (PMID: 38759656; <https://doi.org/10.1016/j.ccell.2024.04.011>). The authors might consider citing these papers in their discussion.

We thank the referee for bringing up these relevant and highly interesting publications, which we have now cited in our revised manuscript.